# Uni-NTFM: A Unified Foundation Model for EEG Signal Representation Learning

**Zhisheng Chen[1,2], Yingwei Zhang[1,2] [†‡], Qizhen Lan[3], Tianyu Liu[4], Huacan Wang[2], Yi Ding[5],
Ziyu Jia[6], Ronghao Chen[7] [†], Kun Wang[5] & Xinliang Zhou[5] [†]**
[1]Beijing Key Laboratory for Multimodal Collaboration and Advanced Application, Institute of
Computing Technology, Chinese Academy of Sciences  [2]University of the Chinese Academy of
Sciences  [3]University of Alabama at Birmingham  [4]National University of Singapore  [5]Nanyang
Technological University  [6]Institute of Automation, Chinese Academy of Sciences  [7]Peking University
`chenzhisheng25@mails.ucas.ac.cn`, `zhangyingwei@ict.ac.cn`,
`chenronghao@alumni.pku.edu.cn`, `xinliang001@e.ntu.edu.sg`.

## Abstract

Current foundation models for electroencephalography (EEG) rely on architectures adapted from computer vision or natural language processing, typically treating neural signals as pixel grids or token sequences. This approach overlooks that the neural activity is activated by diverse sparse coding across a complex geometric topological cortex. Inspired by biological neural mechanisms, we propose the Unified Neural Topological Foundation Model (Uni-NTFM), an architecture rooted in three core neuroscience principles. In detail, to align with the brain's decoupled coding mechanism, we design the Heterogeneous Feature Projection Module. This module simultaneously encodes both time-domain nonstationary transients and frequency-domain steady-state rhythms, ensuring high quality in both waveform morphology and spectral rhythms. Moreover, we introduce a Topological Embedding mechanism to inject structured spatial priors and align different sensor configurations onto a unified latent functional topography, effectively reconstructing the geometry of brain regions. Furthermore, we achieve functional modularization and sparse coding efficiency of biological networks by constructing the Mixture-of-Experts Transformer network. This dynamic routing mechanism assigns different signal patterns and tasks to specialized neural subnetworks, and effectively preventing task interference while increasing the model capacity to record-breaking 1.9 billion parameters. Uni-NTFM is pre-trained on a diverse corpus comprising 28,000 hours of EEG data, and outperforms existing models across nine distinct downstream tasks under both linear probing and fine-tuning settings, demonstrating that aligning model architecture with neural mechanisms is significant to learn universal representations and achieve generalizable brain decoding. Our code is available at `https://github.com/czs-ict/Uni-NTFM`.

## 1 Introduction

Electroencephalography (EEG), as an effective observational window with high temporal resolution, provides a critical technological means for real-time monitoring of brain activity, has indispensable value in fields such as clinical diagnosis, neuroscience research, and Brain-Computer Interfaces (BCIs) (Pfurtscheller & Neuper, 2001; Flesher et al., 2021; Ieracitano et al., 2019). However, with the increasing ability for data acquisition, researchers are confronted with complex and massively scaled EEG data. In this context, traditional task-specific modeling approaches, limited by their generalization capabilities, are no longer sufficient to fully uncover the universal neural encoding principles embedded within (Banville et al., 2021).

---

† Corresponding Authors
‡ Project Leader

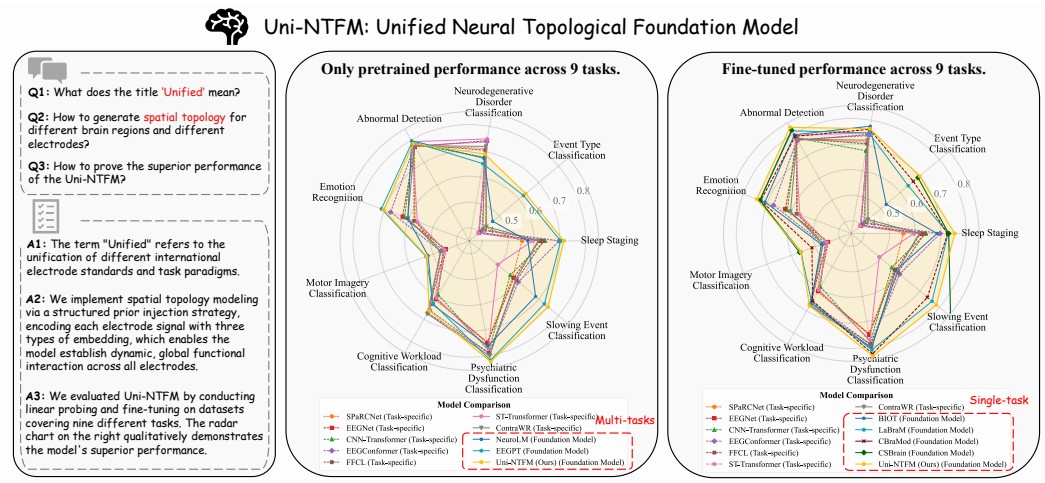

Figure 1: The overview of Uni-NTFM's core concepts and superior performance. The left panel summarizes the model's "Unified" principle, spatial topology strategy, and method of performance validation in a Q&A format. The two radar charts on the right demonstrate that Uni-NTFM comprehensively outperforms baseline models across nine distinct downstream tasks in both linear probing and fine-tuning settings.

Concurrently, the field of artificial intelligence has witnessed the success of the foundation model paradigm, which acquires general-purpose data representations through large-scale self-supervised learning. And this paradigm has achieved breakthrough progress in natural language processing and computer vision (Devlin et al., 2019; Radford et al., 2021; Kirillov et al., 2023; Achiam et al., 2023). Inspired by this, the researchers are actively exploring the migration of this paradigm to the EEG domain (Jiang et al., 2024a; Wang et al., 2024b). The goal is to fully unlock the potential of large-scale EEG data through self-supervised learning, thereby discovering universal neural representations and fundamentally elevating our understanding of the brain's complex dynamics (Zhou et al., 2025a). The work related to brain foundation models (BFMs) is described in detail in Appendix A. However, current BFMs largely inherit the general-purpose paradigm designed for language or vision: segmenting continuous signals into discrete, fixed-scale units (patches or tokens) and then employing a dense attention network for global information interaction. While this approach formally unifies the processing pipeline by directly transferring a model paradigm based on symbols and pixels to the physiological signal domain, its basic design philosophy contradicts the fundamental physical properties of EEG signals, thereby imposing limitations on the model's representation ability. We argue that this mismatch of existing architectures and neural activities creates three critical barriers to learning universal brain representations:

**1) Inability to Capture Decoupled Neural Coding:** Standard architectures process information as a single, homogeneous stream, which contradicts the brain's decoupled coding mechanism. Forcing time-domain and frequency-domain modalities into a unified processing channel conflates waveform morphology with spectral structure.

**2) Failure to Reconstruct Unified Functional Topography:** Existing models typically treat EEG electrodes as invariant sequences, ignoring that they are discrete spatial samplers of a continuous, complex geometric topological cortex. This neglect of the functional topography prevents models from aligning different montage configurations into a consistent semantic space.

**3) Lack of Functional Modularity and Sparse Efficiency:** Biological networks achieve specialized corresponding for specific stimuli through functional modularization and sparse coding. In contrast, standard dense Transformers activate all parameters for each input, which is prone to task interference when processing the highly heterogeneous patterns of EEG signals.

In summary, existing models fundamentally lack an analysis of the intrinsic structure of EEG signals. Therefore, constructing a new architecture capable of synergistically modeling the brain's decoupled coding mechanisms, geometric functional topology, and sparse modularity has become

an imperative for the advancement of the field. As Figure 1, to address these challenges, we propose the **Unified Neural Topological Foundation Model (Uni-NTFM)**, a foundation model designed for generalized EEG decoding that is deeply aligned with the brain's principles. We design a multi-domain and structure-aware information processing framework that computationally emulates three fundamental neural processes: 1) dual-stream neural perception, 2) spatial topology construction, 3) modular cognitive specialization. Our core contributions are as follows:

**1) Biologically-Grounded Transient-Sustained Dual Coding:** We emulate the brain's parallel processing architecture by designing a *Heterogeneous Feature Projection Module (HFPM)*. This module physically decouples neural information into complementary streams, which aims to capture non-stationary transients by the time path and steady-state rhythms by the frequency path. Furthermore, a *Dual-domain Cross-attention Module (DCM)* is established to facilitate a cooperative interaction between these representations, generating a deeply fused representation through bidirectional information enhancement.

**2) Montage Alignment via Topological Embedding:** To reconstruct the brain's geometry from different sensors, we introduce a hierarchical *Topological Embedding (TE)* scheme. By encoding spatial semantics at the intra-region level, region level, and absolute sequence level, this mechanism injects a precise neural coordinate system into the latent space and aligns heterogeneous electrode montages onto a unified functional topography.

**3) Functional Modularity with Sparse MoE:** We simulate the functional specialization and task decoupling of biological networks by replacing the dense FFN with a *Mixture-of-Experts (MoE)* architecture. This design enables dynamic functional routing, where specific signal patterns are processed by specialized expert subnetworks to relieve interferences between different tasks.

**4) Multi-view Self-Supervised Learning:** We propose a pre-training objective that forces the model to master the generative rules of neural activity. By simultaneously reconstructing the masked time-domain waveform patterns, frequency-domain spectral rhythms, and MoE auxiliary loss, this multi-view paradigm ensures that the model learns generalizable and robust representations of brain dynamics, rather than merely superficial features.

We pre-trained Uni-NTFM on a massive corpus containing 28,000 hours of EEG recordings using our specifically designed dual-domain reconstruction self-supervised objective. Extensive experiments on downstream tasks robustly demonstrate the superiority of our new paradigm. Under a linear probing setting without any fine-tuning, the model already exhibits powerful general-purpose representation capabilities. After fine-tuned, Uni-NTFM's performance on public datasets across more than nine different BCI tasks not only comprehensively surpasses existing task-specific models but also significantly outperforms other mainstream foundation models.

## 2 METHODOLOGY

This section explains the paradigm of Uni-NTFM, a self-supervised foundation model proposed to address the challenges of large-scale EEG representation learning. We believe that it is essential to synergistically analyze three intrinsic properties of EEG signals: waveform morphology, spectral rhythms, and spatial topology. Furthermore, to efficiently process these complex multi-domain representations, a sparsely-activated MoE pipeline is required. Thus, we design a hierarchical information processing framework, as shown in Figure 2, which simulates the neural processing: from multi-domain perception and cross-domain fusion to high-level cognitive specialization.

### 2.1 INPUT DATA PREPROCESSING

The raw EEG data is processsd into tensor shapes of $X \in \mathbb{R}^{B \times R \times E \times T}$, where $B$ is the batch size, $R$ and $E$ are the number of predefined brain regions and the maximum number of electrodes per region, respectively, and $T$ is the length of the time series. To enhance the model's robustness to common variations in real-world signals, we also simulate noise, channel loss, and temporal drift in signal acquisition through a series of probabilistic data augmentations ($f_{\text{aug}} : X \rightarrow X_{\text{aug}}$) before being entered into the model.

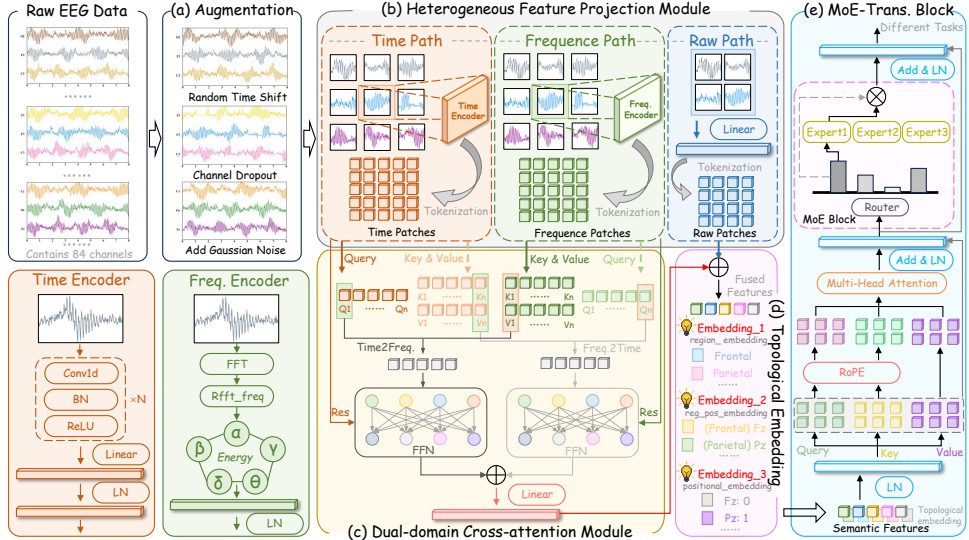

Figure 2: The end-to-end architecture of Uni-NTFM in detail. The data processing flow begins with the data augmentation of raw EEG signals, followed by Heterogeneous Feature Projection Module (HFPM) which parallelly decomposes the input EEG signals into three domain streams: time, frequency, and raw. Next, Dual-domain Cross-attention Module (DCM) performs cross fusion of time domain and frequency domain features, combined with Topological Embedding (TE) to encode the spatial prior information of the electrodes. Finally, the processed representations are sent to the core MoE-Trans. Block to learn universal semantic features.

## 2.2 DECOUPLING AND SYNERGY OF HETEROGENEOUS FEATURES

Unlike images or text, which have natural discrete units (pixels, words), continuous EEG signals can't be tokenized in this way. A common approach that segments the time series into temporal "patches" ignores the continuous characteristic of the signal and fails to capture features at different temporal scales. To address this, we propose a paradigm: instead of segmenting the time axis, we treat the entire time series $x_i \in \mathbb{R}^T$ (where $i$ is the global electrode index) from each individual electrode as a holistic "patch". Our Heterogeneous Feature Projection Module (HFPM) is designed to transform these holistic, channel-wise patches into a set of multi-domain feature embeddings. This module simultaneously projects each channel's signals into three distinct feature domains.

### 2.2.1 DYNAMICS WAVEFORM ENCODER (TIME PATH)

To capture the local waveform structures and non-stationary events of the signal, we employ a one-dimensional convolutional encoder, $\Phi_T(\cdot)$, to map $x_i$ into a temporal feature vector $h_{i,T} \in \mathbb{R}^D$:

$$h_{i,T} = \Phi_T(x_i) \tag{1}$$

This encoder consists of multiple convolutional blocks, where the operation of each block can be formalized as:

$$h^{(l+1)} = \text{ReLU}(\text{BN}(W^{(l)} * h^{(l)} + b^{(l)})) \tag{2}$$

### 2.2.2 FREQUENCY DECOMPOSITION ENCODER (FREQUENCY PATH)

To extract the crucial steady-state rhythmic information from the signal, we first decompose the signal into the frequency domain via the Discrete Fourier Transform to compute the Power Spectral Density. Then parameterize it into a mean power vector $P_b \in \mathbb{R}^{N_b}$ for $N_b$ core frequency bands. This vector is subsequently projected into a frequency feature vector $h_{i,F} \in \mathbb{R}^D$ by an MLP, $\Phi_F(\cdot)$.

$$h_{i,F} = \Phi_F(P_b(x_i)) \tag{3}$$

The mean power $P_{b_j}$ for the $j$-th frequency band is defined as:

$$P_{b_j} = \frac{1}{|\mathcal{K}_j|} \sum_{k \in \mathcal{K}_j} \left| \sum_{t=0}^{T-1} x_i(t) e^{-j2\pi kt/T} \right|^2 \tag{4}$$

where $\mathcal{K}_j$ is the set of discrete frequency indices corresponding to the $j$-th frequency band.

### 2.2.3 STANDARD PROJECTION ENCODER (RAW PATH)

While the time and frequency paths perform non-linear transformations that selectively amplify partial features, the raw path establishes a high-quality and information-lossless reference for the self-supervised reconstruction task. This reference representation serves as the "ground truth" target $H_R$ in our reconstruction loss (Section 2.5), ensuring that the model is trained to recover the full signal, not just a specific domain. This design is critical for maintaining the clarity and integrity of the pre-training objective. The explaination for this process is shown in Appendix E.1

### 2.2.4 DUAL-DOMAIN CROSS-ATTENTION MODULE

Decomposing the signal into parallel temporal and frequency representations, the next critical step is to achieve a unified understanding by modeling their interdependencies. To this end, we introduce the Dual-domain Cross-attention Module (DCM). Distinct from standard self-attention where Query, Key, and Value are from the same source, our module forces a cross-domain dialogue: the temporal feature sequence $H_T$ serves as the Query to probe for relevant Key and Value of frequency features $H_F$, and vice versa.

$$H'_T = \text{LayerNorm}(H_T + \text{CrossAttn}(Q = H_T, K = H_F, V = H_F)) \tag{5}$$

$$H'_F = \text{LayerNorm}(H_F + \text{CrossAttn}(Q = H_F, K = H_T, V = H_T)) \tag{6}$$

The final fused feature $H_{\text{fused}} = \text{FFN}(\text{Concat}(H'_T, H'_F))$ contains deep interaction information between the two domains.

## 2.3 EXPLICIT AND UNIFICATION EMBEDDING OF SPATIAL TOPOLOGY

A critical bottleneck in developing a generalist BFM is the montage heterogeneity problem: unlike the fixed pixel grid in computer vision, EEG datasets exhibit substantial variability in sensor configurations (e.g., clinical 19-channel 10-20 systems and high-density 64-channel 10-10 systems). Standard Transformers, which utilize learnable 1D position encodings, treat electrodes as the simple sequence, thereby discarding the geometric structure of the cortical surface. To overcome this, we introduce a hierarchical Topological Embedding (TE) mechanism that projects diverse sensor layouts onto a unified, biologically grounded standardized neural coordinate system, which deconstructs the spatial identity of each electrode into three levels of neuroanatomical semantics.

**(1) Region Embedding** ($E_{\text{region}}$): To capture the functional modularity of the cortex, we partition the scalp topology into five canonical brain regions based on the international 10-20 system standards. We define a learnable embedding matrix $E_{\text{region}} \in \mathbb{R}^{5 \times D}$ corresponding to: (a) **Frontal**: Encoding executive functions and decision; (b) **Central**: Encoding sensorimotor rhythms; (c) **Temporal**: Encoding auditory processing and memory; (d) **Parietal**: Encoding spatial attention and sensory; (e) **Occipital**: Encoding visual processing features. By explicitly anchoring signals to these functional domains, the model learns to generalize features regardless of the specific channel index used in a particular dataset.

**(2) Intra-Region Embedding** ($E_{\text{intra}}$): Within each region, electrode density varies across standards. To resolve this, we introduce intra-region coordinates that encode the relative spatial orientation. This embedding ensures that the model understands the geometric relationship between neighbors, such as $C3$ and $C1$ are spatially adjacent in the motor cortex. This mechanism provides the structural basis for robustness against missing channels.

**(3) Global Absolute Embedding** ($E_{\text{abs}}$): To preserve precise channel-specific identities for standard clinical montages, we assign a unique global identifier to standard electrodes defined in the International Federation of Clinical Neurophysiology (IFCN) guidelines.

For an input token $x_i$ originating from a specific electrode, its final spatial representation is the summation of these hierarchical priors:

$$H_{in}^{(i)} = H_{\text{fused}}^{(i)} + H_{\text{R}}^{(i)} + E_{\text{region}}[x_{\text{region}}^{(i)}] + E_{\text{intra}}[x_{\text{intra}}^{(i)}] + E_{\text{abs}}[x_{\text{abs}}^{(i)}] \tag{7}$$

where $H_{\text{fused}}^{(i)}$ denotes the dual-domain features integrated by the DCM module, and $H_R^{(i)}$ represents the raw signal reference features. The $E_{\text{region}}$, $E_{\text{intra}}$, and $E_{\text{abs}}$ refer to the learnable embedding matrices for the region, intra-region, and global absolute position, respectively. The indices $x_{\text{region}}^{(i)}$, $x_{\text{intra}}^{(i)}$, and $x_{\text{abs}}^{(i)}$ correspond to the specific neuroanatomical coordinates of the $i$-th electrode within the standardized coordinate system.

This hierarchical design allows Uni-NTFM to function as a geometry-aware encoder. When evaluating on the incomplete data, the model utilizes the $E_{region}$ and $E_{intra}$ embeddings to correctly map these signals to their respective cortical sources in the latent space. This effectively solves the cross-montage transfer challenge without requiring specific re-training.

## 2.4 FUNCTIONALLY MOE-BASED NEURAL TRANSFORMER

The challenge in scaling the Transformer block is that increasing capacity via dense FFN leads to a quadratic increase in computational cost. To overcome this, we replace the dense FFN in each Transformer block with a sparsely-activated MoE mechanism. This architecture is particularly suitable for EEG modeling, as it allows the model to learn specialized subnetworks for distinct signal patterns (e.g., specific neural rhythms, artifacts, or pathological events) through its gating mechanism. This functional specialization not only enhances modeling accuracy but also offers significant advantages in downstream adaptation, where fine-tuning a small subset of relevant experts can lead to highly efficient and robust transfer learning.

$$H_l' = H_{l-1} + \text{MultiHeadSelfAttn}(\text{LayerNorm}(H_{l-1})) \tag{8}$$

$$H_l = H_l' + \text{MoE}(\text{LayerNorm}(H_l')) \tag{9}$$

### 2.4.1 NEURAL MULTI-HEAD SELF-ATTENTION

The self-attention mechanism is designed to learn long-range functional connections in the brain. To enable it to perceive the relative spatial order of electrodes, we introduce Rotary Position Encoding (RoPE). For a $d$-dimensional vector $x_m$ at position $m$, its rotated form $x_m'$ is obtained by grouping the vector into pairs $(x_{m,2i-1}, x_{m,2i})$ and applying a rotation matrix $\mathbf{R}_{\Theta,m,i}$:

$$\begin{pmatrix} x_{m,2i-1}' \\ x_{m,2i}' \end{pmatrix} = \begin{pmatrix} \cos(m\theta_i) & -\sin(m\theta_i) \\ \sin(m\theta_i) & \cos(m\theta_i) \end{pmatrix} \begin{pmatrix} x_{m,2i-1} \\ x_{m,2i} \end{pmatrix} \tag{10}$$

where $\theta_i = 10000^{-2i/d}$. A property of RoPE is that the attention score intrinsically depends only on the relative position $m - n$:

$$\langle q_m', k_n' \rangle = \langle \mathbf{R}_{\Theta,m} q_m, \mathbf{R}_{\Theta,n} k_n \rangle = \langle q_m, \mathbf{R}_{\Theta,n-m} k_n \rangle \tag{11}$$

### 2.4.2 SPARSELY-ACTIVATED MOE MODULE

MoE allows the model to learn function-specific subnetworks. A gating network $g(h_i) = h_i W_g$ computes logits for each input token $h_i$ across $N_e$ expert networks. Through Top-k gating, the final output for a token is the weighted sum of the outputs from the $k$ expert networks $E_j(\cdot)$:

$$\text{MoE}(h_i) = \sum_{j \in \text{TopK}(\text{Softmax}(g(h_i)))} p_j E_j(h_i) \tag{12}$$

To encourage load balancing, we introduce an auxiliary loss $\mathcal{L}_{\text{aux}}$:

$$\mathcal{L}_{\text{aux}} = \alpha \cdot N_e \sum_{j=1}^{N_e} f_j \cdot \bar{p}_j \tag{13}$$

where $f_j$ is the fraction of tokens routed to expert $j$, and $\bar{p}_j$ is the average probability assigned to expert $j$ for those tokens.

## 2.5 Dual-domain Self-supervised Reconstruction Objective

The entire learning process is driven by the masked autoencoding self-supervised task (He et al., 2022). For an input sequence $H_{in}$, we randomly select an index subset $\mathcal{M}$ and replace its tokens with a shared, learnable embedding $e_{[MASK]}$. The model's optimization objective is to reconstruct the original features of the masked tokens. To this end, we designed a Dual-Domain Loss Function, $\mathcal{L}_{total}$, which forces the model to maintain consistency in both the time and frequency domains simultaneously:

$$\mathcal{L}_{total} = \lambda_T \mathcal{L}_{time} + \lambda_F \mathcal{L}_{freq} + \lambda_{aux} \mathcal{L}_{aux} \tag{14}$$

The temporal reconstruction loss $\mathcal{L}_{time}$ and frequency reconstruction loss $\mathcal{L}_{freq}$ are calculated as the Mean Squared Error over the masked positions $\mathcal{M}$:

$$\mathcal{L}_{time} = \frac{1}{|\mathcal{M}|} \sum_{i \in \mathcal{M}} \|\text{Head}_T(H_{out,i}) - h_{i,R}\|_2^2 \tag{15}$$

$$\mathcal{L}_{freq} = \frac{1}{|\mathcal{M}|} \sum_{i \in \mathcal{M}} \|\text{Head}_F(H_{out,i}) - h_{i,F}\|_2^2 \tag{16}$$

where $H_{out}$ is the final output of the Transformer. This learning paradigm enables Uni-NTFM to learn the profound internal structures and regularities of EEG signals without explicit labels.

## 3 Experiments

### 3.1 Pre-training Settings

**1) Data Summary.** We aggregated a large-scale pre-training corpus by amalgamating nine distinct, publicly available EEG datasets to train a foundation model capable of learning truly generalizable neural representations. As detailed in Appendix Table 6, this corpus comprises data from over 17,000 subjects and amounts to approximately 28,000 hours of recordings. It includes recordings from resting-state conditions (e.g., REEG-BACA (Getzmann et al., 2024), Resting State EEG (Trujillo et al., 2017)), emotion induction tasks (e.g., Emobrain (Savran[1] et al., 2006), SEED-series (Zheng et al., 2018; Liu et al., 2021; 2022)), cognitive classification tasks (e.g., Raw EEG Data (Trujillo, 2020)), BCI paradigms (BCI Competition IV-1 (Blankertz et al., 2007)), and extensive clinical recordings from hospital environments (e.g., TUEG (Obeid & Picone, 2016), CAUEEG (Kim et al., 2023), Siena Scalp EEG Database (Detti et al., 2020)).

**2) Data Preprocessing.** Initially, a zero-phase filter was applied with a passband of 0.5-50 Hz, and a notch filter suppressed powerline interference at 50 Hz and its harmonics. To ensure a uniform temporal resolution, all signals were then downsampled to a consistent 200 Hz sampling rate. Before normalization, all channel amplitudes were uniformly scaled to millivolts (mV).

**3) Training Settings.** We designed four versions of Uni-NTFM at different scales, with 57M, 427M, 912M, and 1.9B parameters, respectively. All experiments were conducted on NVIDIA A100-80G GPUs, using Python 3.9.23 and PyTorch 2.3.1 with CUDA 11.8. The specific configurations for each model and other detailed hyperparameter settings are provided in Appendix Table 8.

### 3.2 Downstream Datasets

**1) Data Summary.** To comprehensively evaluate the performance of Uni-NTFM, we collected nine public EEG datasets across various paradigms and tasks, as detailed in Appendix Table 7: 1) TUAB (Harati et al., 2015), 2) TUEV (Harati et al., 2015), 3) SEED (Zheng & Lu, 2015), 4) TDBrain (Van Dijk et al., 2022), 5) ADFTD (Miltiadous et al., 2023), 6) BCIC-IV-2a (Brunner et al., 2008), 7) Workload (Zyma et al., 2019), 8) HMC (Alvarez-Estevez & Rijsman, 2021), and 9) TUSL (von Weltin et al., 2017). We employed two evaluation strategies: first, we used linear probing to directly evaluate the quality of the representations learned by the pre-trained model; second, we conducted full fine-tuning on downstream tasks to check the model's generalization and adaptation abilities.

**2) Evaluation Metrics.** We choose the following metrics to evaluate the model's generalization abilities across various tasks: **A) Balanced Accuracy:** The metric is defined as the arithmetic mean of each class's recall. **B) AUROC:** The metric represents the model's ability to distinguish between

classes, which is independent of the chosen classification threshold. **C) AUC-PR:** The metric is the area under the curve that plots precision against recall for different classification thresholds. **D) Cohen's Kappa ($\kappa$):** The metric is used to measure the agreement between a classifier's predictions and the ground truth. **E) F1-Score:** The metric is the harmonic mean of precision and recall. In the subsequent experiments, we select Balanced Accuracy, AUROC, and AUC-PR as the evaluation metric for binary classification tasks. And we adopt Balanced Accuracy, Cohen's Kappa, and F1-Score to evaluate the performance on multi-class classification tasks. The detailed introduction of the formulas is in Appendix C.

## 3.3 RESULTS ON DOWNSTREAM DATASETS

Table 1: Best performances on TUAB, TUEV, and SEED.

| Method | TUAB (2-class) | | | TUEV (6-class) | | | SEED (3-class) | | |
|---|---|---|---|---|---|---|---|---|---|
| | Balanced Acc. | AUC-PR | AUROC | Balanced Acc. | Cohen's Kappa | Weighted F1 | Balanced Acc. | Cohen's Kappa | Weighted F1 |
| *Traditional Task-specific Methods (Single-task)* | | | | | | | | | |
| SPaRCNet (Jing et al., 2023) | $77.49_{\pm0.91}$ | $83.14_{\pm0.71}$ | $86.31_{\pm0.58}$ | $43.15_{\pm3.17}$ | $43.79_{\pm2.17}$ | $68.14_{\pm1.67}$ | $57.36_{\pm2.92}$ | $35.27_{\pm4.12}$ | $56.35_{\pm2.23}$ |
| EEGNet (Lawhern et al., 2018) | $77.12_{\pm0.56}$ | $82.31_{\pm0.29}$ | $85.01_{\pm0.27}$ | $39.75_{\pm1.07}$ | $34.93_{\pm1.13}$ | $63.84_{\pm2.06}$ | $62.49_{\pm3.52}$ | $44.73_{\pm2.29}$ | $61.77_{\pm2.66}$ |
| CNN-Transformer (Peh et al., 2022) | $78.24_{\pm1.07}$ | $84.86_{\pm0.96}$ | $84.88_{\pm0.76}$ | $41.94_{\pm1.26}$ | $39.32_{\pm1.72}$ | $67.82_{\pm2.17}$ | $61.61_{\pm3.84}$ | $42.62_{\pm6.01}$ | $61.50_{\pm4.63}$ |
| EEGConformer (Song et al., 2022) | $77.46_{\pm0.37}$ | $84.31_{\pm0.65}$ | $85.03_{\pm0.54}$ | $41.34_{\pm2.64}$ | $40.27_{\pm1.82}$ | $69.72_{\pm1.49}$ | $67.42_{\pm2.07}$ | $46.18_{\pm1.99}$ | $64.31_{\pm5.29}$ |
| FFCL (Li et al., 2022) | $77.94_{\pm0.19}$ | $84.33_{\pm0.38}$ | $85.94_{\pm0.66}$ | $40.02_{\pm1.55}$ | $36.97_{\pm2.11}$ | $67.99_{\pm1.37}$ | $57.76_{\pm2.61}$ | $37.66_{\pm3.28}$ | $58.13_{\pm2.33}$ |
| ST-Transformer (Song et al., 2021) | $79.28_{\pm0.17}$ | $85.36_{\pm0.57}$ | $86.89_{\pm0.24}$ | $39.59_{\pm1.76}$ | $38.33_{\pm2.29}$ | $69.11_{\pm2.08}$ | $56.39_{\pm1.77}$ | $37.44_{\pm1.82}$ | $56.27_{\pm1.44}$ |
| ContraWR (Yang et al., 2023b) | $77.52_{\pm0.44}$ | $84.60_{\pm0.68}$ | $84.61_{\pm0.67}$ | $43.58_{\pm2.71}$ | $39.88_{\pm1.68}$ | $68.36_{\pm1.45}$ | $60.34_{\pm0.65}$ | $41.31_{\pm1.32}$ | $60.99_{\pm1.13}$ |
| *Only Pretrained Foundation Models (Multi-tasks)* | | | | | | | | | |
| NeuroLM-XL (Jiang et al., 2024a) | $79.69_{\pm0.91}$ | $72.19_{\pm0.82}$ | $78.84_{\pm1.94}$ | $46.79_{\pm3.56}$ | $45.70_{\pm4.98}$ | $73.59_{\pm2.19}$ | $60.34_{\pm0.10}$ | $40.82_{\pm0.36}$ | $60.63_{\pm0.30}$ |
| EEGPT (Wang et al., 2024a) | $79.22_{\pm0.46}$ | $74.55_{\pm0.50}$ | $86.62_{\pm0.79}$ | $62.32_{\pm1.14}$ | $63.51_{\pm1.34}$ | $81.87_{\pm0.63}$ | $71.22_{\pm0.22}$ | $57.34_{\pm0.49}$ | $70.99_{\pm0.38}$ |
| Uni-NTFM$_{tiny}$ | $71.36_{\pm2.18}$ | $77.07_{\pm1.59}$ | $78.21_{\pm1.42}$ | $56.94_{\pm1.93}$ | $60.11_{\pm1.72}$ | $76.91_{\pm1.91}$ | $67.46_{\pm0.48}$ | $55.11_{\pm0.72}$ | $68.71_{\pm0.55}$ |
| Uni-NTFM$_{small}$ | $74.50_{\pm1.25}$ | $79.31_{\pm1.72}$ | $80.56_{\pm1.14}$ | $59.33_{\pm1.61}$ | $63.12_{\pm2.08}$ | $79.15_{\pm2.58}$ | $68.86_{\pm0.93}$ | $56.23_{\pm0.49}$ | $69.92_{\pm0.47}$ |
| Uni-NTFM$_{middle}$ | $76.71_{\pm1.33}$ | $81.44_{\pm1.55}$ | $81.82_{\pm1.12}$ | $61.01_{\pm1.37}$ | $64.79_{\pm1.22}$ | $81.37_{\pm1.60}$ | $69.51_{\pm0.87}$ | $57.17_{\pm0.59}$ | $70.76_{\pm0.55}$ |
| Uni-NTFM$_{large}$ | $78.44_{\pm0.96}$ | $\underline{82.59}_{\pm1.71}$ | $82.78_{\pm1.36}$ | $62.44_{\pm1.99}$ | $65.48_{\pm1.42}$ | $82.13_{\pm1.39}$ | $70.25_{\pm0.64}$ | $57.81_{\pm0.66}$ | $71.42_{\pm0.39}$ |
| *Pretrained and Fine-tuned Foundation Models (Single-task)* | | | | | | | | | |
| BIOT (Yang et al., 2023a) | $79.59_{\pm0.57}$ | $87.92_{\pm0.23}$ | $88.15_{\pm0.43}$ | $52.81_{\pm2.25}$ | $52.73_{\pm2.49}$ | $74.92_{\pm0.82}$ | $70.97_{\pm0.24}$ | $56.82_{\pm0.51}$ | $71.34_{\pm0.27}$ |
| LaBraM-Base (Jiang et al., 2024b) | $81.40_{\pm0.19}$ | $89.65_{\pm0.16}$ | $\textbf{90.22}_{\pm0.09}$ | $64.09_{\pm0.65}$ | $66.37_{\pm0.93}$ | $83.12_{\pm0.52}$ | $73.18_{\pm0.19}$ | $\textbf{59.94}_{\pm0.31}$ | $73.54_{\pm0.21}$ |
| CBraMod (Wang et al., 2024b) | $78.91_{\pm0.30}$ | $86.36_{\pm0.63}$ | $86.06_{\pm0.57}$ | $66.71_{\pm1.07}$ | $67.72_{\pm0.96}$ | $83.42_{\pm0.64}$ | $72.72_{\pm0.66}$ | $57.64_{\pm0.17}$ | $73.01_{\pm0.41}$ |
| CSBrain * (Zhou et al., 2025b) | $81.72_{\pm0.43}$ | $\textbf{90.05}_{\pm0.66}$ | $89.57_{\pm0.46}$ | $69.03_{\pm0.59}$ | $68.33_{\pm0.47}$ | $83.33_{\pm0.57}$ | $73.02_{\pm0.35}$ | $59.31_{\pm0.26}$ | $72.98_{\pm0.36}$ |
| Uni-NTFM$_{tiny}$ | $76.49_{\pm1.31}$ | $82.30_{\pm1.52}$ | $83.25_{\pm1.26}$ | $64.05_{\pm1.42}$ | $65.82_{\pm1.45}$ | $81.74_{\pm0.97}$ | $71.18_{\pm0.54}$ | $57.62_{\pm1.18}$ | $71.97_{\pm0.23}$ |
| Uni-NTFM$_{small}$ | $78.93_{\pm0.62}$ | $85.53_{\pm1.41}$ | $86.01_{\pm0.87}$ | $66.94_{\pm1.38}$ | $67.96_{\pm1.10}$ | $83.25_{\pm1.37}$ | $72.02_{\pm0.33}$ | $58.59_{\pm0.65}$ | $72.74_{\pm0.49}$ |
| Uni-NTFM$_{middle}$ | $80.81_{\pm0.88}$ | $87.95_{\pm0.91}$ | $88.37_{\pm0.66}$ | $68.86_{\pm1.57}$ | $69.22_{\pm1.33}$ | $84.09_{\pm0.83}$ | $72.90_{\pm0.51}$ | $59.22_{\pm0.82}$ | $73.42_{\pm0.44}$ |
| Uni-NTFM$_{large}$ | $\textbf{81.97}_{\pm0.40}$ | $\underline{89.82}_{\pm0.58}$ | $\underline{89.64}_{\pm1.18}$ | $\textbf{69.91}_{\pm1.70}$ | $\textbf{70.30}_{\pm1.48}$ | $\textbf{84.66}_{\pm1.32}$ | $\textbf{73.37}_{\pm0.45}$ | $59.76_{\pm0.58}$ | $\textbf{73.81}_{\pm0.69}$ |

\* The CSBrain code is not open-source, and the data in the table is all provided from its paper.

To comprehensively evaluate the generalization abilities of Uni-NTFM, we assessed its performance across nine diverse downstream tasks using two distinct strategies: linear probing of the frozen pre-trained model and full fine-tuning. Results are presented as the mean $\pm$ standard deviation, calculated over five independent trials, and a comprehensive table summarizing model size and performance is provided in the Appendix D. As detailed in Table 1, 2, 3, our results consistently demonstrate the superiority of the Uni-NTFM paradigm, establishing a new state-of-the-art (SOTA) across a wide field of EEG analysis application. In addition, we also conducted scaling law experiments on Uni-NTFM, and the detailed results can be found in Appendix H.

**1) Linear Probing Performance:** Under the linear probing setting, which directly measures the intrinsic quality of the learned representations, Uni-NTFM exhibits remarkable transferability and representation quality. Even without fine-tuning, the model consistently outperforms traditional task-specific methods and other pretrained foundation models across the majority of tasks. Specifically, on the TUAB abnormal detection task, the Uni-NTFM$_{large}$ model achieves a Balanced Accuracy of 0.7844, significantly surpassing the task-specific SPARCNet model. This performance across varied tasks, from clinical event detection to cognitive state classification, highlights the universal representations learned through our proposed dual-domain, structure-aware pretraining objective.

**2) Fine-tuned Performance:** By full fine-tuning, Uni-NTFM's performance is further promoted, highlighting its strong ability for task-specific adaptation. On all nine datasets, the fine-tuned Uni-NTFM variants consistently set new performance benchmarks. Especially on complex multi-class tasks such as SEED (Emotion Recognition) and TUEV (Event Type Classification), Uni-NTFM demonstrates substantial benefits over both task-specific models and other foundation models like LaBraM-Base and CBraMod. For example, on the 3-class SEED task, Uni-NTFM$_{large}$ achieves a Balanced Accuracy of 0.7337, indicating its robust ability to decode different cognitive states.

Table 2: Best performances on TDBrain, ADFTD, and BCIC-IV-2a.

| Method | TDBrain (2-class) | | | ADFTD (3-class) | | | BCIC-IV-2a (4-class) | | |
|---|---|---|---|---|---|---|---|---|---|
| | Balanced Acc. | AUC-PR | AUROC | Balanced Acc. | Cohen's Kappa | Weighted F1 | Balanced Acc. | Cohen's Kappa | Weighted F1 |
| *Traditional Task-specific Methods (Single-task)* | | | | | | | | | |
| SPaRCNet (Jing et al., 2023) | $74.67_{\pm1.06}$ | $82.66_{\pm0.92}$ | $81.79_{\pm1.16}$ | $71.32_{\pm1.56}$ | $72.88_{\pm2.12}$ | $73.37_{\pm1.19}$ | $46.97_{\pm1.41}$ | $28.71_{\pm1.29}$ | $44.60_{\pm1.37}$ |
| EEGNet (Lawhern et al., 2018) | $75.16_{\pm1.35}$ | $82.73_{\pm0.88}$ | $83.14_{\pm1.95}$ | $74.32_{\pm1.59}$ | $76.66_{\pm1.99}$ | $74.42_{\pm1.67}$ | $44.62_{\pm1.13}$ | $26.47_{\pm1.09}$ | $43.02_{\pm1.24}$ |
| CNN-Transformer (Peh et al., 2022) | $78.13_{\pm1.98}$ | $82.94_{\pm2.37}$ | $84.62_{\pm1.55}$ | $67.89_{\pm2.04}$ | $70.91_{\pm2.23}$ | $67.29_{\pm2.55}$ | $45.83_{\pm1.61}$ | $28.25_{\pm1.39}$ | $44.73_{\pm1.28}$ |
| EEGConformer (Song et al., 2022) | $79.25_{\pm3.19}$ | $83.81_{\pm1.66}$ | $85.31_{\pm2.01}$ | $73.88_{\pm1.23}$ | $73.39_{\pm1.11}$ | $73.01_{\pm1.59}$ | $46.77_{\pm1.48}$ | $29.17_{\pm1.75}$ | $45.82_{\pm1.39}$ |
| FFCL (Li et al., 2022) | $78.99_{\pm2.10}$ | $81.56_{\pm2.19}$ | $84.15_{\pm0.81}$ | $70.62_{\pm1.31}$ | $68.25_{\pm2.36}$ | $70.60_{\pm2.14}$ | $45.11_{\pm1.05}$ | $27.14_{\pm1.89}$ | $43.23_{\pm1.70}$ |
| ST-Transformer (Song et al., 2021) | $77.37_{\pm1.02}$ | $83.39_{\pm0.96}$ | $83.36_{\pm1.32}$ | $75.04_{\pm1.56}$ | $73.44_{\pm2.31}$ | $74.82_{\pm2.66}$ | $45.21_{\pm1.66}$ | $27.09_{\pm1.55}$ | $44.63_{\pm1.92}$ |
| ContraWR (Yang et al., 2023b) | $80.17_{\pm1.79}$ | $84.22_{\pm1.54}$ | $86.04_{\pm1.77}$ | $72.11_{\pm1.10}$ | $75.22_{\pm2.99}$ | $73.75_{\pm2.79}$ | $46.82_{\pm1.37}$ | $28.94_{\pm1.41}$ | $44.34_{\pm1.53}$ |
| *Only Pretrained Foundation Models (Multi-tasks)* | | | | | | | | | |
| NeuroLM-XL (Jiang et al., 2024a) | $76.29_{\pm1.32}$ | $74.81_{\pm1.67}$ | $75.52_{\pm1.14}$ | $67.46_{\pm1.64}$ | $68.28_{\pm1.04}$ | $72.52_{\pm0.97}$ | $51.95_{\pm0.74}$ | $38.74_{\pm0.91}$ | $49.22_{\pm0.61}$ |
| EEGPT (Wang et al., 2024a) | $81.94_{\pm1.62}$ | $83.77_{\pm0.89}$ | $85.61_{\pm1.03}$ | $65.31_{\pm1.27}$ | $62.92_{\pm0.88}$ | $63.92_{\pm0.74}$ | $\mathbf{52.81}_{\pm0.55}$ | $40.33_{\pm0.62}$ | $51.29_{\pm0.68}$ |
| Uni-NTFM$_{tiny}$ | $81.66_{\pm0.52}$ | $85.19_{\pm0.87}$ | $85.33_{\pm0.74}$ | $66.61_{\pm0.88}$ | $66.25_{\pm0.71}$ | $70.29_{\pm0.94}$ | $51.22_{\pm0.73}$ | $39.68_{\pm0.86}$ | $50.25_{\pm0.65}$ |
| Uni-NTFM$_{small}$ | $82.07_{\pm0.97}$ | $85.62_{\pm1.17}$ | $86.18_{\pm1.02}$ | $67.35_{\pm0.49}$ | $67.44_{\pm1.13}$ | $71.31_{\pm0.72}$ | $52.09_{\pm0.67}$ | $40.44_{\pm1.02}$ | $51.23_{\pm0.46}$ |
| Uni-NTFM$_{middle}$ | $82.12_{\pm0.61}$ | $85.70_{\pm0.81}$ | $86.72_{\pm1.19}$ | $67.80_{\pm0.77}$ | $\mathbf{68.51}_{\pm0.76}$ | $72.08_{\pm0.67}$ | $52.66_{\pm0.36}$ | $41.07_{\pm0.79}$ | $50.77_{\pm1.31}$ |
| Uni-NTFM$_{large}$ | $\mathbf{82.46}_{\pm0.43}$ | $\mathbf{85.84}_{\pm1.36}$ | $\mathbf{86.91}_{\pm0.79}$ | $\mathbf{68.13}_{\pm0.64}$ | $68.22_{\pm1.35}$ | $\mathbf{72.60}_{\pm0.79}$ | $52.58_{\pm0.62}$ | $\mathbf{41.18}_{\pm1.45}$ | $\mathbf{51.44}_{\pm1.05}$ |
| *Pretrained and Fine-tuned Foundation Models (Single-task)* | | | | | | | | | |
| BIOT (Yang et al., 2023a) | $80.66_{\pm0.73}$ | $84.94_{\pm1.35}$ | $85.31_{\pm0.66}$ | $\mathbf{77.63}_{\pm0.91}$ | $74.48_{\pm0.43}$ | $77.21_{\pm0.79}$ | $47.48_{\pm0.93}$ | $29.97_{\pm1.39}$ | $46.07_{\pm1.25}$ |
| LaBraM-Base (Jiang et al., 2024b) | $81.25_{\pm0.91}$ | $84.22_{\pm0.47}$ | $86.48_{\pm0.55}$ | $74.92_{\pm1.33}$ | $73.35_{\pm0.22}$ | $\mathbf{77.47}_{\pm0.83}$ | $55.97_{\pm0.49}$ | $41.66_{\pm1.14}$ | $56.23_{\pm0.45}$ |
| CBraMod (Wang et al., 2024b) | $82.81_{\pm0.64}$ | $85.37_{\pm0.78}$ | $87.44_{\pm0.85}$ | $76.39_{\pm1.12}$ | $75.59_{\pm0.85}$ | $75.33_{\pm0.46}$ | $51.38_{\pm0.66}$ | $35.18_{\pm0.94}$ | $\underline{49.84}_{\pm0.85}$ |
| CSBrain * (Zhou et al., 2025b) | — | — | — | — | — | — | $\mathbf{56.57}_{\pm0.71}$ | $42.09_{\pm0.93}$ | $\mathbf{56.37}_{\pm0.87}$ |
| Uni-NTFM$_{tiny}$ | $82.26_{\pm0.39}$ | $85.56_{\pm0.93}$ | $86.11_{\pm0.78}$ | $74.70_{\pm0.62}$ | $75.17_{\pm0.86}$ | $76.16_{\pm0.51}$ | $54.23_{\pm0.48}$ | $41.43_{\pm0.61}$ | $54.11_{\pm0.80}$ |
| Uni-NTFM$_{small}$ | $83.11_{\pm0.58}$ | $85.95_{\pm0.69}$ | $87.04_{\pm0.60}$ | $75.68_{\pm0.57}$ | $76.32_{\pm0.63}$ | $76.80_{\pm0.42}$ | $55.59_{\pm0.60}$ | $42.01_{\pm0.57}$ | $54.65_{\pm0.95}$ |
| Uni-NTFM$_{middle}$ | $83.37_{\pm0.81}$ | $\underline{85.88}_{\pm0.55}$ | $87.33_{\pm0.36}$ | $76.29_{\pm0.53}$ | $\underline{75.79}_{\pm0.81}$ | $77.11_{\pm0.92}$ | $55.18_{\pm0.67}$ | $41.90_{\pm0.61}$ | $54.72_{\pm0.77}$ |
| Uni-NTFM$_{large}$ | $\mathbf{83.69}_{\pm0.93}$ | $\mathbf{85.97}_{\pm0.75}$ | $\mathbf{87.48}_{\pm0.49}$ | $76.61_{\pm0.55}$ | $\mathbf{76.58}_{\pm0.42}$ | $77.38_{\pm0.88}$ | $56.08_{\pm0.94}$ | $\mathbf{42.66}_{\pm0.87}$ | $55.33_{\pm0.86}$ |

\* The CSBrain code is not open-source, and the data in the table is all provided from its paper.

Table 3: Best performances on Workload, HMC, and TUSL.

| Method | Workload (2-class) | | | HMC (5-class) | | | TUSL (3-class) | | |
|---|---|---|---|---|---|---|---|---|---|
| | Balanced Acc. | AUC-PR | AUROC | Balanced Acc. | Cohen's Kappa | Weighted F1 | Balanced Acc. | Cohen's Kappa | Weighted F1 |
| *Traditional Task-specific Methods (Single-task)* | | | | | | | | | |
| SPaRCNet (Jing et al., 2023) | $61.32_{\pm0.47}$ | $67.26_{\pm2.48}$ | $68.39_{\pm2.24}$ | $55.37_{\pm3.74}$ | $47.88_{\pm2.97}$ | $56.98_{\pm4.34}$ | $56.85_{\pm2.51}$ | $49.16_{\pm4.01}$ | $58.44_{\pm4.86}$ |
| EEGNet (Lawhern et al., 2018) | $60.85_{\pm2.55}$ | $58.17_{\pm2.01}$ | $62.38_{\pm1.51}$ | $62.16_{\pm2.12}$ | $56.14_{\pm1.88}$ | $62.77_{\pm1.42}$ | $57.37_{\pm4.17}$ | $49.83_{\pm4.48}$ | $52.92_{\pm6.88}$ |
| CNN-Transformer (Peh et al., 2022) | $59.11_{\pm1.95}$ | $57.23_{\pm3.37}$ | $59.81_{\pm2.66}$ | $64.19_{\pm2.41}$ | $58.22_{\pm2.66}$ | $67.36_{\pm1.77}$ | $55.62_{\pm1.53}$ | $48.72_{\pm2.06}$ | $56.44_{\pm2.65}$ |
| EEGConformer (Song et al., 2022) | $65.87_{\pm2.19}$ | $69.22_{\pm1.81}$ | $68.77_{\pm3.32}$ | $69.78_{\pm1.44}$ | $62.16_{\pm2.34}$ | $68.87_{\pm1.22}$ | $59.72_{\pm8.12}$ | $51.33_{\pm4.76}$ | $60.02_{\pm4.16}$ |
| FFCL (Li et al., 2022) | $67.28_{\pm3.41}$ | $76.49_{\pm1.15}$ | $75.66_{\pm3.01}$ | $63.68_{\pm3.66}$ | $55.38_{\pm2.13}$ | $66.43_{\pm3.60}$ | $58.76_{\pm1.91}$ | $47.04_{\pm1.57}$ | $51.36_{\pm1.92}$ |
| ST-Transformer (Song et al., 2021) | $60.43_{\pm1.19}$ | $56.81_{\pm1.41}$ | $62.55_{\pm1.67}$ | $59.49_{\pm5.43}$ | $50.03_{\pm2.01}$ | $61.47_{\pm3.31}$ | $58.33_{\pm5.57}$ | $30.16_{\pm10.07}$ | $41.20_{\pm5.88}$ |
| ContraWR (Yang et al., 2023b) | $67.55_{\pm2.42}$ | $75.91_{\pm2.80}$ | $76.33_{\pm2.29}$ | $61.59_{\pm6.23}$ | $56.89_{\pm4.42}$ | $62.31_{\pm1.83}$ | $58.33_{\pm3.99}$ | $42.67_{\pm2.81}$ | $55.08_{\pm4.66}$ |
| *Only Pretrained Foundation Models (Multi-tasks)* | | | | | | | | | |
| NeuroLM-XL (Jiang et al., 2024a) | $63.45_{\pm4.42}$ | $58.89_{\pm4.23}$ | $61.30_{\pm7.64}$ | $57.61_{\pm10.84}$ | $47.95_{\pm14.66}$ | $58.83_{\pm12.86}$ | $68.45_{\pm3.04}$ | $51.94_{\pm4.61}$ | $68.39_{\pm2.97}$ |
| EEGPT (Wang et al., 2024a) | $62.99_{\pm1.78}$ | $67.92_{\pm0.92}$ | $69.28_{\pm1.08}$ | $70.29_{\pm0.82}$ | $65.84_{\pm0.59}$ | $\mathbf{73.23}_{\pm0.41}$ | $72.88_{\pm1.43}$ | $59.72_{\pm2.09}$ | $72.33_{\pm1.53}$ |
| Uni-NTFM$_{tiny}$ | $63.06_{\pm1.44}$ | $66.54_{\pm1.95}$ | $69.55_{\pm1.87}$ | $69.85_{\pm2.45}$ | $63.73_{\pm2.41}$ | $69.34_{\pm0.71}$ | $71.08_{\pm1.57}$ | $62.55_{\pm3.21}$ | $71.38_{\pm2.51}$ |
| Uni-NTFM$_{small}$ | $64.11_{\pm2.29}$ | $67.71_{\pm3.18}$ | $70.26_{\pm2.16}$ | $70.88_{\pm2.39}$ | $65.03_{\pm1.35}$ | $71.55_{\pm1.62}$ | $72.29_{\pm0.90}$ | $63.73_{\pm2.14}$ | $72.20_{\pm3.11}$ |
| Uni-NTFM$_{middle}$ | $63.79_{\pm1.53}$ | $67.86_{\pm1.66}$ | $\mathbf{70.99}_{\pm1.34}$ | $71.34_{\pm1.66}$ | $65.58_{\pm1.16}$ | $71.76_{\pm0.92}$ | $72.56_{\pm1.44}$ | $63.52_{\pm1.41}$ | $72.55_{\pm2.66}$ |
| Uni-NTFM$_{large}$ | $\mathbf{64.16}_{\pm2.68}$ | $\mathbf{68.48}_{\pm1.83}$ | $\underline{70.87}_{\pm1.68}$ | $\mathbf{71.55}_{\pm2.33}$ | $\mathbf{66.11}_{\pm1.45}$ | $72.44_{\pm1.11}$ | $\mathbf{73.14}_{\pm1.15}$ | $\mathbf{64.11}_{\pm2.66}$ | $\mathbf{73.13}_{\pm3.61}$ |
| *Pretrained and Fine-tuned Foundation Models (Single-task)* | | | | | | | | | |
| BIOT (Yang et al., 2023a) | $66.55_{\pm6.65}$ | $71.89_{\pm7.22}$ | $73.42_{\pm5.36}$ | $68.62_{\pm0.41}$ | $62.95_{\pm1.13}$ | $70.91_{\pm1.47}$ | $57.58_{\pm3.03}$ | $20.12_{\pm2.12}$ | $23.94_{\pm0.40}$ |
| LaBraM-Base (Jiang et al., 2024b) | $66.09_{\pm2.04}$ | $71.74_{\pm2.34}$ | $72.72_{\pm1.65}$ | $72.86_{\pm1.01}$ | $68.12_{\pm0.73}$ | $75.54_{\pm0.24}$ | $76.25_{\pm1.31}$ | $64.07_{\pm3.04}$ | $76.14_{\pm2.10}$ |
| CBraMod (Wang et al., 2024b) | $65.37_{\pm2.60}$ | $70.39_{\pm1.33}$ | $70.07_{\pm2.32}$ | $72.69_{\pm0.41}$ | $66.85_{\pm1.04}$ | $\underline{73.95}_{\pm0.89}$ | $73.88_{\pm3.20}$ | $61.49_{\pm5.50}$ | $74.53_{\pm3.60}$ |
| CSBrain * (Zhou et al., 2025b) | — | — | — | $\mathbf{73.45}_{\pm0.47}$ | $68.18_{\pm0.46}$ | $75.06_{\pm0.42}$ | $\mathbf{85.71}_{\pm2.40}$ | $\mathbf{78.28}_{\pm2.70}$ | $\mathbf{85.68}_{\pm1.80}$ |
| Uni-NTFM$_{tiny}$ | $65.60_{\pm2.25}$ | $70.84_{\pm2.11}$ | $72.62_{\pm1.29}$ | $72.44_{\pm1.99}$ | $66.91_{\pm1.22}$ | $74.43_{\pm0.78}$ | $76.69_{\pm2.36}$ | $66.21_{\pm2.11}$ | $76.45_{\pm2.07}$ |
| Uni-NTFM$_{small}$ | $\mathbf{66.72}_{\pm1.71}$ | $71.73_{\pm1.59}$ | $73.88_{\pm1.50}$ | $72.81_{\pm1.86}$ | $67.69_{\pm1.34}$ | $75.00_{\pm2.71}$ | $78.00_{\pm2.71}$ | $67.52_{\pm1.33}$ | $77.10_{\pm1.33}$ |
| Uni-NTFM$_{middle}$ | $66.16_{\pm2.24}$ | $71.66_{\pm1.13}$ | $\underline{73.46}_{\pm1.72}$ | $72.88_{\pm1.00}$ | $67.77_{\pm1.15}$ | $75.12_{\pm1.20}$ | $77.79_{\pm3.06}$ | $67.75_{\pm3.14}$ | $76.91_{\pm2.62}$ |
| Uni-NTFM$_{large}$ | $66.44_{\pm1.47}$ | $\mathbf{72.28}_{\pm1.32}$ | $\mathbf{74.92}_{\pm1.01}$ | $73.11_{\pm0.97}$ | $\mathbf{68.32}_{\pm0.77}$ | $\mathbf{75.72}_{\pm0.54}$ | $78.44_{\pm2.55}$ | $68.53_{\pm1.30}$ | $77.46_{\pm1.45}$ |

\* The CSBrain code is not open-source, and the data in the table is all provided from its paper.

## 3.4 ABLATION STUDY

To systematically validate the effect of each core component of the Uni-NTFM architecture, we conducted a comprehensive ablation study on the TUAB and TUEV downstream tasks, and the results are detailed in Table 4, and the visualization of ablation study on the TUAB and TUEV downstream tasks are shown in Fig 3.

The baseline model (A1, B1), which removes all our proposed modules and represents a standard Vision Transformer, establishes the lowest performance benchmark. The introduction of the HFPM (A2, B2) yields the most substantial single-component performance gain, increasing AUROC from 0.7116 to 0.7805 on TUAB. This highlights the critical importance of decoupling and encoding heterogeneous features. Subsequently, the addition of the DCM (A3, B3) and the TE (A8, B8) further improves performance, confirming their respective functions in fusing multi-domain information and inserting spatial priors. Besides, the MoE module provides a limited improvement when added in alone (A5, B5), its contribution becomes significantly more pronounced when combined with other components (A6, A7, B6, B7). The full model (A10, B10), which integrates all components,

Table 4: Ablation study on TUAB and TUEV datasets.

| No. | Modules | | | | TUAB (2-class) | | | TUEV (6-class) | | |
|-----|------|-----|----|-----|---------------|---------|-------|----------------|---------------|-------------|
| | HFPM | DCM | TE | MoE | Balanced Acc. | AUC-PR | AUROC | Balanced Acc. | Cohen's Kappa | Weighted F1 |
| 1 | ✗ | ✗ | ✗ | ✗ | $62.52_{\pm4.83}$ | $68.71_{\pm3.69}$ | $71.16_{\pm5.37}$ | $58.74_{\pm3.16}$ | $62.29_{\pm2.27}$ | $73.66_{\pm2.82}$ |
| 2 | ✓ | ✗ | ✗ | ✗ | $69.95_{\pm3.24}$ | $77.78_{\pm2.90}$ | $78.05_{\pm2.53}$ | $61.45_{\pm2.56}$ | $63.34_{\pm2.58}$ | $76.69_{\pm3.51}$ |
| 3 | ✓ | ✓ | ✗ | ✗ | $73.64_{\pm2.31}$ | $79.92_{\pm2.11}$ | $79.76_{\pm2.85}$ | $62.88_{\pm1.95}$ | $63.93_{\pm2.15}$ | $78.72_{\pm1.67}$ |
| 4 | ✗ | ✗ | ✓ | ✗ | $67.40_{\pm3.61}$ | $74.05_{\pm2.88}$ | $76.14_{\pm3.39}$ | $60.60_{\pm2.54}$ | $64.11_{\pm1.99}$ | $77.81_{\pm1.71}$ |
| 5 | ✗ | ✗ | ✗ | ✓ | $66.23_{\pm2.08}$ | $73.87_{\pm2.44}$ | $78.62_{\pm2.51}$ | $59.86_{\pm3.44}$ | $62.79_{\pm2.65}$ | $75.52_{\pm2.13}$ |
| 6 | ✗ | ✗ | ✓ | ✓ | $71.81_{\pm1.95}$ | $78.49_{\pm1.46}$ | $80.03_{\pm2.12}$ | $61.50_{\pm2.02}$ | $64.28_{\pm2.37}$ | $78.94_{\pm2.23}$ |
| 7 | ✓ | ✗ | ✓ | ✓ | $73.34_{\pm2.42}$ | $79.33_{\pm1.86}$ | $81.10_{\pm2.55}$ | $63.35_{\pm2.49}$ | $65.39_{\pm1.88}$ | $80.81_{\pm2.41}$ |
| 8 | ✓ | ✓ | ✗ | ✓ | $74.96_{\pm1.92}$ | $81.37_{\pm1.64}$ | $82.39_{\pm2.05}$ | $63.16_{\pm1.73}$ | $64.80_{\pm1.89}$ | $80.19_{\pm1.16}$ |
| 9 | ✓ | ✓ | ✓ | ✗ | $74.17_{\pm1.63}$ | $80.94_{\pm2.28}$ | $81.40_{\pm1.90}$ | $63.53_{\pm2.20}$ | $64.41_{\pm1.75}$ | $79.39_{\pm1.42}$ |
| 10 | ✓ | ✓ | ✓ | ✓ | $76.49_{\pm1.31}$ | $82.30_{\pm1.52}$ | $83.25_{\pm1.26}$ | $64.05_{\pm1.42}$ | $65.82_{\pm1.45}$ | $81.74_{\pm0.97}$ |

**HFPM:** Heterogeneous Feature Projection Module; **DCM:** Dual-domain Cross-attention Module; **TE:** Topological Embedding; and **MoE:** Mixture-of-Experts.

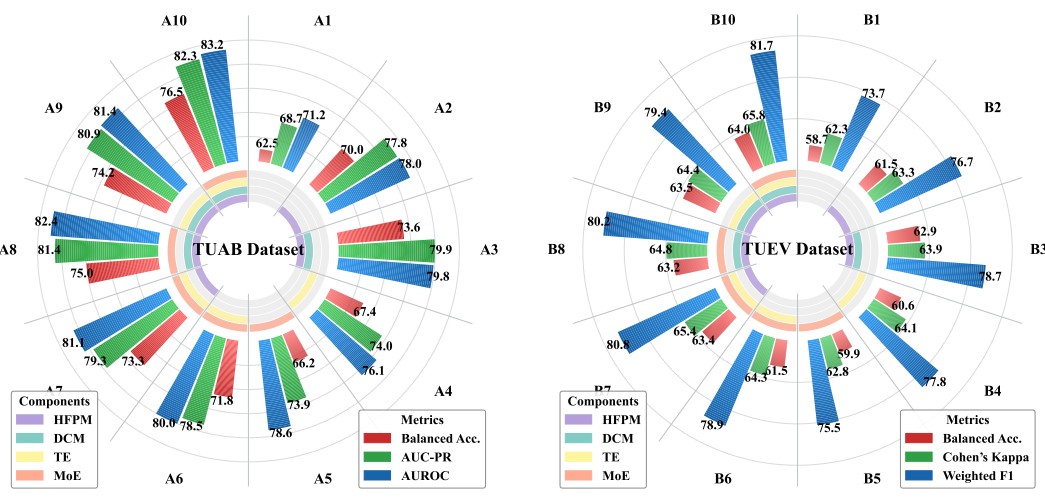

(a) Ablation Study on the TUAB Dataset.

(b) Ablation Study on the TUEV Dataset.

Figure 3: The visualization of ablation study on the TUAB and TUEV downstream tasks.

achieves the highest performance across all metrics on both datasets. Especially on the TUEV, the full model achieves a significant improvement over the baseline and other combination of modules.

## 4 CONCLUSIONS

This paper introduces the Unified Neural Topological Foundation Model (Uni-NTFM), which counters the limitations of traditional models by adopting an architecture designed from the first principles of neuroscience. Specifically, we simultaneously process the temporal, frequency, and standard properties of EEG signals, and insert the spatial priors of multi-domain representations before transfor into the scalable MoE neural Transformer for specialized feature processing. Moreover, pretrained on over 28,000 hours of data, Uni-NTFM sets the standard across nine downstream tasks, demonstrating superior generalization. The success of our model validates the structure-aware modeling theory that respects the characteristics of neural signals is necessary for releasing the potential of large-scale brain foundation models. We hope this work provides a method for the next generation of efficient and interpretable brain-intelligence interfaces in neuroscience applications.

## ACKNOWLEDGEMENT

This work is supported by the Natural Science Foundation of China (No.62302487), Improvement Project of Chinese Academy of Sciences (No.GSZXKYZB2025007), and the Science and Technology Innovation Program of Hunan Province (No.2024JJ9031).

## ETHICS STATEMENT

This research was conducted exclusively using publicly available and fully anonymized EEG datasets. Our model, Uni-NTFM, is intended for foundational research purposes. We acknowledge that potential biases may be inherited from the training data, and addressing fairness is a critical step for future real-world applications. The overarching goal of this work is to contribute positively to the fields of neuroscience and medicine.

## REPRODUCIBILITY

To ensure full reproducibility, we will provide the following:

- **Code and Models:** All source code will be released under an open-source license at `https://anonymous.4open.science/r/Uni-NTFM-0924`.
- **Datasets:** All datasets for pre-training and evaluation are publicly available and are detailed in Appendix Table 6 and Table 7, respectively.
- **Hyperparameters:** Detailed configurations and hyperparameters for all model variants and experiments are provided in the Appendix (Table 8, 9, 10, 11).
- **Environment:** The computing environment is specified in Section 3.1.

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

## A    RELATED WORK

### A.1    MOTIVATION FOR BRAIN FOUNDATION MODELS

Traditional models for electroencephalography (EEG) decoding were mainly task-specific, leading to several core limitations that motivated the shift towards foundation models. These challenges include (Kuruppu et al., 2025; Xiong et al., 2025; Wang et al., 2024c): **1) Poor Generalizability:** Models trained for a specific task, dataset, or individual struggled to transfer knowledge to new contexts. **2) Data Heterogeneity:** EEG data varies significantly across studies in terms of electrode configurations, signal lengths, and sampling rates, creating barriers to cross-dataset learning. **3) Expensive Annotation:** Labeling EEG data requires significant domain expertise and is time-consuming, making large-scale supervised datasets scarce. To overcome these issues, researchers have turned to developing Brain Foundation Models (BFMs), which learn universal and transferable neural representations through self-supervised pre-training on large, diverse datasets (Li et al., 2025; Babu et al., 2025; Aristimunha et al., 2025).

### A.2    INNOVATIONS IN BRAIN FOUNDATION MODELS

Existing BFMs have advanced the field through innovations in architecture and pre-training strategies, primarily centered around self-supervised learning.

#### A.2.1    RECONSTRUCTION-BASED PRE-TRAINING

This paradigm, inspired by masked autoencoders, is the most common approach. Models are trained to reconstruct masked portions of the EEG signal. **LaBraM** (Jiang et al., 2024b) pioneered the use of a vector-quantized neural tokenizer to convert continuous EEG signals into discrete neural codes. By pre-training the model to predict these masked codes, it effectively mitigates signal noise. It has been applied to tasks such as abnormal detection and emotion recognition. **EEGPT** (Wang et al., 2024a) introduced a dual self-supervised learning strategy that combines masked reconstruction with spatio-temporal representation alignment. Instead of reconstructing the raw signal, it aligns the features of masked segments with those of the complete signal, thereby improving representation quality. **BrainWave** (Yuan et al., 2024) is the first foundation model jointly pre-trained on both invasive (iEEG) and non-invasive (EEG) neural signals. It has demonstrated strong zero-shot and few-shot classification abilities in the diagnosis and identification of various neurological disorders. **CBraMod** (Wang et al., 2024b) employs a crisscross transformer backbone to model the EEG signals by processing spatial and temporal dependencies in parallel and uses an asymmetric conditional positional encoding to adapt to diverse EEG formats. It has been evaluated on over 10 BCI tasks, including emotion recognition and seizure detection.

#### A.2.2    ARCHITECTURES TAILORED TO EEG STRUCTURE

**CSBrain** (Zhou et al., 2025b) addresses the intrinsic crossscale nature of EEG signals by introducing Cross-scale Spatiotemporal Tokenization and Structured Sparse Attention. This design explicitly models neural patterns at multiple resolutions to suit diverse task requirements. **CodeBrain** (Ma et al., 2025)introduces a TFDual-Tokenizer to independently encode temporal and frequency components. Its backbone is an efficient State Space Model designed to capture the sparse, long-range dependencies characteristic of brain.

#### A.2.3    DOMAIN-SPECIFIC AND LLM-INTEGRATED MODELS

**NeuroLM** (Jiang et al., 2024a) and **UniMind** (Lu et al., 2025) are pioneering models that integrate EEG encoders with Large Language Models (LLMs) to create unified, multi-task decoders that operate via instruction tuning. To bridge the significant modality gap, **NeuroLM** uses adversarial training to align EEG and text embedding spaces, while **UniMind** designs a Neuro-Language Connector and a Task-aware Query Selection module to distill neural patterns into LLM-interpretable representations. These models are applied to a wide range of tasks including sleep staging and clinical event classification .

## B   THE USE OF LARGE LANGUAGE MODELS(LLMs)

During the preparation of this manuscript, we used Google's Gemini, as a writing-assistance tool. The use of the LLM was strictly limited to improving the language and readability of our text. Key applications included:

- Proofreading for grammatical errors, spelling mistakes, and incorrect punctuation.
- Rephrasing sentences to enhance clarity and conciseness.
- Ensuring a consistent and formal academic tone throughout the document.

Crucially, the LLM was not used for any core scientific aspects of this work. All conceptual contributions, including the formulation of the research problem, the development of the methodology, and the execution of experiments, are exclusively the work of the human authors. The authors have reviewed and edited all text, and take full responsibility for the scientific integrity and final content of this paper.

## C   DESCRIPTIONS OF EVALUATION METRICS

In a classification context, a confusion matrix is used to visualize the performance of an algorithm. For a binary classification problem, the matrix consists of four fundamental quantities:

- **True Positives (TP):** The number of positive instances that were correctly classified as positive.
- **True Negatives (TN):** The number of negative instances that were correctly classified as negative.
- **False Positives (FP):** The number of negative instances that were incorrectly classified as positive. This is also known as a Type I error.
- **False Negatives (FN):** The number of positive instances that were incorrectly classified as negative. This is also known as a Type II error.

Based on these four values, we can define several key evaluation metrics.

### DETAILED EVALUATION METRICS

#### C.1   BALANCED ACCURACY

Balanced Accuracy is the arithmetic mean of the recall for each class. It is particularly useful for datasets with imbalanced class distributions as it avoids inflated performance estimates.

For a problem with $C$ classes, the recall for class $i$ is first calculated as:

$$\text{Recall}_i = \frac{\text{TP}_i}{\text{TP}_i + \text{FN}_i} \tag{17}$$

where $\text{TP}_i$ and $\text{FN}_i$ are the true positives and false negatives for class $i$, respectively.

The Balanced Accuracy is then computed by averaging these recall values:

$$\text{Balanced Accuracy} = \frac{1}{C}\sum_{i=1}^{C}\text{Recall}_i = \frac{1}{C}\sum_{i=1}^{C}\frac{\text{TP}_i}{\text{TP}_i + \text{FN}_i} \tag{18}$$

#### C.2   AUROC (AREA UNDER THE RECEIVER OPERATING CHARACTERISTIC CURVE)

The AUROC metric evaluates a model's ability to distinguish between classes across all possible classification thresholds. The ROC curve is a plot of the True Positive Rate (TPR) against the False

Positive Rate (FPR).

$$\text{TPR (Sensitivity)} = \frac{\text{TP}}{\text{TP} + \text{FN}} \tag{19}$$

$$\text{FPR (1 - Specificity)} = \frac{\text{FP}}{\text{FP} + \text{TN}} \tag{20}$$

Mathematically, the area under this curve is calculated by integrating the TPR function with respect to the FPR:

$$\text{AUROC} = \int_0^1 \text{TPR}(\text{FPR}) \, d(\text{FPR}) \tag{21}$$

AUROC is the area under this curve, with a value ranging from 0 to 1. A value of 1 indicates a perfect classifier, while 0.5 suggests performance no better than random chance.

### C.3 AUC-PR (AREA UNDER THE PRECISION-RECALL CURVE)

The AUC-PR metric is the area under the curve that plots Precision against Recall at various thresholds.

$$\text{Precision} = \frac{\text{TP}}{\text{TP} + \text{FP}} \tag{22}$$

$$\text{Recall} = \frac{\text{TP}}{\text{TP} + \text{FN}} \tag{23}$$

The area is computed by integrating the precision function $P(r)$ with respect to recall $r$:

$$\text{AUC-PR} = \int_0^1 P(r) \, dr \tag{24}$$

This metric is especially informative for imbalanced datasets, as its calculation does not depend on the number of True Negatives. A higher AUC-PR value indicates better model performance.

### C.4 COHEN'S KAPPA ($\kappa$)

Cohen's Kappa coefficient ($\kappa$) measures the agreement between a classifier's predictions and the ground truth, correcting for the probability of agreement occurring by chance. It is a more robust metric than simple accuracy on imbalanced datasets. The formula is:

$$\kappa = \frac{p_o - p_e}{1 - p_e} \tag{25}$$

where:

- $p_o$ is the observed agreement (i.e., overall accuracy):

$$p_o = \frac{\text{TP} + \text{TN}}{\text{TP} + \text{TN} + \text{FP} + \text{FN}}$$

- $p_e$ is the expected probability of chance agreement. For a binary case, it is calculated as:

$$p_e = \frac{(\text{TP} + \text{FP})(\text{TP} + \text{FN}) + (\text{FN} + \text{TN})(\text{FP} + \text{TN})}{(\text{Total Samples})^2}$$

### C.5 F1-SCORE

The F1-Score is the harmonic mean of Precision and Recall. It provides a single score that balances both concerns, making it a useful metric when both false positives and false negatives are important.

$$F_1 = 2 \cdot \frac{\text{Precision} \cdot \text{Recall}}{\text{Precision} + \text{Recall}} = \frac{2\text{TP}}{2\text{TP} + \text{FP} + \text{FN}} \tag{26}$$

# D  COMPREHENSIVE COMPARISON OF COMPLEXITY AND PERFORMANCE

Table 5: Comprehensive Comparison of Complexity and Performance.

| Model | Model Complexity (Parameters) | | Binary Classification | | | Multi-class Classification | | |
|---|---|---|---|---|---|---|---|---|
| | Total | Infer. | Bal. Acc. | AUC-PR | AUROC | Bal. Acc. | Kappa | W-F1 |
| *Traditional Task-specific Methods* | | | | | | | | |
| SPARCNet (Jing et al., 2023) | 0.79 M | 0.79 M | 71.16 | 76.79 | 78.83 | 55.17 | 59.65 | 40.41 |
| EEGNet (Lawhern et al., 2018) | 2.8 K | 2.8 K | 71.04 | 74.40 | 78.40 | 56.79 | 48.13 | 59.79 |
| CNN-Transformer (Peh et al., 2022) | 3.2 M | 3.2 M | 71.83 | 75.01 | 76.04 | 56.18 | 48.01 | 60.86 |
| EEGConformer (Song et al., 2022) | 1 M | 1 M | 74.19 | 79.11 | 77.90 | 59.82 | 50.42 | 62.63 |
| FFCL (Li et al., 2022) | 2.4 M | 2.4 M | 74.74 | 79.80 | 80.52 | 55.99 | 45.41 | 59.68 |
| ST-Transformer (Song et al., 2021) | 3.5 M | 3.5 M | 72.36 | 75.19 | 77.60 | 54.17 | 42.75 | 57.92 |
| ContraWR (Yang et al., 2023b) | 1.6 M | 1.6 M | 75.08 | 81.58 | 80.33 | 57.13 | 47.49 | 60.81 |
| *Foundation Models* | | | | | | | | |
| NeuroLM-XL (Jiang et al., 2024a) | 1.7 B | 1.7 B | 73.14 | 68.63 | 71.89 | 58.77 | 48.91 | 63.86 |
| EEGPT (Wang et al., 2024a) | 25 M | 25 M | 74.72 | 75.41 | 80.50 | 65.81 | 58.28 | 68.94 |
| BIOT (Yang et al., 2023a) | 3.6 M | 3.6 M | 75.60 | 81.58 | 81.29 | 62.52 | 49.51 | 60.73 |
| LaBraM-Base (Jiang et al., 2024b) | 5.8 M | 5.8 M | 76.25 | 81.87 | 82.07 | 69.55 | 62.25 | 73.67 |
| CBraMod (Wang et al., 2024b) | 4.0 M | 4.0 M | 75.70 | 80.71 | 81.19 | 68.96 | 60.75 | 71.68 |
| *Proposed Models (Uni-NTFM)* | | | | | | | | |
| **Uni-NTFM**$_{tiny}$ | 57 M | **19 M** | 74.78 | 79.57 | 80.66 | 68.88 | 62.19 | 72.48 |
| **Uni-NTFM**$_{small}$ | 427 M | 74 M | 76.35 | 81.07 | 82.31 | 70.10 | 63.35 | 73.34 |
| **Uni-NTFM**$_{middle}$ | 912 M | 148 M | 76.78 | 81.83 | 83.05 | 70.56 | 63.61 | 73.65 |
| **Uni-NTFM**$_{large}$ | 1.9 B | 307 M | **77.37** | **82.69** | **84.01** | **71.25** | **64.36** | **74.26** |

*Note: Infer.: Inference Parameters; Bal. Acc.: Balanced Accuracy; W-F1: Weighted F1.*

Table 6 presents a comparative analysis of model complexity, explicitly distinguishing between Total Parameters and Inference Parameters, which are indicative of the model's learning capacity and the actual computational cost during deployment, respectively.

A fundamental constraint of biological brains is that the cortex operates on a "sparse coding" principle, where only some of the neurons activate for any given stimulus to minimize metabolic cost. Standard dense foundation models violate this principle by forcing full network activation for every input. Uni-NTFM addresses this by implementing a sparse activation mechanism via the Mixture-of-Experts architecture. As shown in Table 6, while Uni-NTFM$_{large}$ scales to 1.9 billion parameters to capture the immense heterogeneity of human neurophysiology, its dynamic routing ensures that only 307 million parameters are active per signal during inference. This effectively decouples the model's knowledge capacity from its execution cost, achieving a considerable efficiency improvement compared to dense baselines.

We argue that model scale corresponds to the diversity of functional experts required to decode the complex signal of the brain. Traditional lightweight models (e.g., EEGNet, 2.8K params) function like specialized reflex circuits, which is efficient but limited in scope. In contrast, Uni-NTFM functions like the neocortex: it has various specialized experts to handle rare epilepsies, subtle emotional shifts, or artifacts. Our Uni-NTFM$_{tiny}$ exemplifies this balance: it leverages 57M total capacity to store generalized representations while maintaining an inference of just 19M parameters (lower than the dense EEGPT). Thus, our architecture effectively balances the conflicting biological demands of high-capacity storage for generalization and rapid neural activation for real-time execution.

# E  PSEUDOCODE

## E.1  DETAILED EXPLAINATION OF HETEROGENEOUS FEATURE PROJECTION MODEL

---
**Algorithm 1** Heterogeneous Feature Projection Model

---
**Input:** preprocessed EEG data $X \in \mathbb{R}^{B \times R \times E \times T}$
**Output:** three feature sequence matrices $H_T, H_F, H_R \in \mathbb{R}^{B \times L \times T}$
 1:  $X_{\text{reshaped}} \leftarrow \text{reshape}(X, (B \cdot L, T))$                           $\triangleright L = R \times E$
 2:  Initialize $H_T, H_F, H_R$ as empty lists
 3:  **for all** $i \in \{1, \ldots, B \cdot L\}$ **do**
 4:      $x_i \leftarrow X_{\text{reshaped}}[i, :]$
 5:      $h_{i,T} \leftarrow \Phi_T(x_i)$                          $\triangleright$ Dynamics Waveform Encoder (Time Path)
 6:      $P_b(x_i) \leftarrow \text{CalculateBandPower}(x_i)$
 7:      $h_{i,F} \leftarrow \Phi_F(P_b(x_i))$          $\triangleright$ Frequency Decomposition Encoder (Frequency Path)
 8:      $h_{i,R} \leftarrow \Phi_R(x_i)$                          $\triangleright$ Standard Projection Encoder (Raw Path)
 9:      Append $h_{i,T}, h_{i,F}, h_{i,R}$ to $H_T, H_F, H_R$ respectively
10:  **end for**
11:  $H_T, H_F, H_R \leftarrow$ stack and reshape each list to $\mathbb{R}^{B \times L \times D}$
12:  **return** $H_T, H_F, H_R$

---

**Input**  $X$: with dimensions $X \in \mathbb{R}^{B \times R \times E \times T}$, where $B$ is the batch size, $R$ is the number of regions, $E$ is the number of electrodes per region, and $T$ is the number of time steps.

**Line 1: Reshape Data**  This step reshapes the original 4D tensor $X$ into a 2D matrix. It merges the region ($R$) and electrode ($E$) dimensions into a new dimension $L$ and flattens the batch ($B$) dimension into it. This is done to treat the time series of each electrode as an independent sample, facilitating subsequent individual processing.

$$X_{\text{reshaped}} \in \mathbb{R}^{(B \cdot L) \times T}, \quad \text{where } L = R \times E \tag{27}$$

After reshaping, we obtain $B \times L$ independent time-series sequences, each of length $T$.

**Lines 3-10: Iterative Feature Extraction**  The algorithm iterates through all $B \times L$ time-series sequences, performing three independent feature encoding steps for each sequence $x_i \in \mathbb{R}^T$:

1) TEMPORAL FEATURE EXTRACTION (LINE 5)  The time series $x_i$ is imported into a Dynamics Waveform Encoder $\Phi_T$. This encoder is typically a neural network designed to capture the dynamic characteristics of the signal in the time domain.

$$h_{i,T} = \Phi_T(x_i), \quad \text{where } \Phi_T : \mathbb{R}^T \to \mathbb{R}^D \tag{28}$$

It maps the sequence of length $T$ to a feature vector $h_{i,T}$ of dimension $D$.

2) FREQUENCY FEATURE EXTRACTION (LINES 6-7)  This process is divided into two steps. First, the "CalculateBandPower" function computes the power of the time series $x_i$ across $N_b$ predefined frequency bands (e.g., $\delta$, $\theta$, $\alpha$). After obtaining the band power vector $p_i$, it is fed into the Frequency Decomposition Encoder $\Phi_F$.

$$p_i = \text{CalculateBandPower}(x_i), \quad p_i \in \mathbb{R}^{N_b} \tag{29}$$

$$h_{i,F} = \Phi_F(p_i), \quad \text{where } \Phi_F : \mathbb{R}^{N_b} \to \mathbb{R}^D \tag{30}$$

This process results in a feature vector $h_{i,F}$ of dimension $D$.

3) STANDARD FEATURE EXTRACTION (LINE 8)  The original time series $x_i$ is also passed to a third, independent Standard Projection Encoder $\Phi_R$. This encoder can use a different network architecture from $\Phi_T$ to provide a complementary feature perspective.

$$h_{i,R} = \Phi_R(x_i), \quad \text{where } \Phi_R : \mathbb{R}^T \to \mathbb{R}^D \tag{31}$$

The output is also a feature vector $h_{i,R}$ of dimension $D$.

**Line 11: Stack and Reshape**   After the loop completes, the three lists of representations are stacked into matrices of shape $(B \cdot L) \times D$ and then reshaped into the final tensor shape of $\mathbb{R}^{B \times L \times D}$, restoring the batch dimension.

### E.2   DETAILED EXPLAINATION OF TOPOLOGICAL EMBEDDING

---

**Algorithm 2** Topological Embedding

---

**Require:** Temporal, frequency, and standard features $H_T, H_F, H_R \in \mathbb{R}^{B \times L \times D}$
**Ensure:** Temporal-frequency-topological features $H_{\text{in}} \in \mathbb{R}^{B \times L \times D}$
 1: $I_{\text{region}} \leftarrow$ Generate Region Indices$(B, L, R, E)$
 2: $I_{\text{intra}} \leftarrow$ Generate Intra Region Indices$(B, L, R, E)$
 3: $I_{\text{abs}} \leftarrow$ torch.arange$(L)$.expand$(B, -1)$
 4: $E_{\text{region\_emb}} \leftarrow \mathbf{E}_{\text{region}}[I_{\text{region}}]$
 5: $E_{\text{intra\_emb}} \leftarrow \mathbf{E}_{\text{intra}}[I_{\text{intra}}]$
 6: $E_{\text{abs\_emb}} \leftarrow \mathbf{E}_{\text{abs}}[I_{\text{abs}}]$
 7: $H_{\text{in}} \leftarrow H_{\text{fused}} + H_R + E_{\text{region\_emb}} + E_{\text{intra\_emb}} + E_{\text{abs\_emb}}$
 8: **return** $H_{\text{in}}$

---

**Lines 1-3: Generate Positional Indices**   The algorithm first generates three types of integer indices, all resulting in tensors of shape $(B, L)$, to encode the spatial hierarchy of the electrodes:

$I_{\text{region}}$: **Region Index**, which identifies the brain region $(0, \ldots, R-1)$ that each of the $L$ electrodes belongs to.

$I_{\text{intra}}$: **Intra-Region Index**, which indicates the relative position $(0, \ldots, E-1)$ of an electrode within its specific region.

$I_{\text{abs}}$: **Absolute Position Index**, which gives the absolute position $(0, \ldots, L-1)$ of each electrode in the flattened sequence.

For any given electrode at absolute position $j \in \{0, \ldots, L-1\}$, its indices are formally generated as:

$$I_{\text{region}}^{(j)} = \lfloor j/E \rfloor \tag{32}$$

$$I_{\text{intra}}^{(j)} = j \pmod{E} \tag{33}$$

where $\lfloor \cdot \rfloor$ is the floor operation and $\pmod{E}$ gives the remainder of a division by $E$.

**Lines 4-6: Lookup Embeddings**   Using the generated indices, the algorithm retrieves corresponding feature vectors from three distinct, learnable embedding matrices. This process maps the discrete integer indices to dense, continuous vector representations. The learnable matrices are:

- $\mathbf{E}_{\text{region}} \in \mathbb{R}^{R \times D}$: An embedding matrix for the $R$ brain regions.
- $\mathbf{E}_{\text{intra}} \in \mathbb{R}^{E \times D}$: An embedding matrix for the $E$ intra-region positions.
- $\mathbf{E}_{\text{abs}} \in \mathbb{R}^{L \times D}$: An embedding matrix for the $L$ absolute positions.

The lookup operation for an electrode at absolute position $j$ can be expressed as:

$$E_{\text{region\_emb}}^{(j)} = \mathbf{E}_{\text{region}}[I_{\text{region}}^{(j)}] \tag{34}$$

This operation is performed for all indices and all three embedding matrices, resulting in three embedding tensors $(E_{\text{region\_emb}}, E_{\text{intra\_emb}}, E_{\text{abs\_emb}})$, each of shape $\mathbb{R}^{B \times L \times D}$.

**Line 7: Final Feature Fusion**   Finally, the algorithm performs an element-wise addition to combine the time-frequency fused features $(H_{\text{fused}})$, the standard projection features $(H_R)$, and the three structural prior embeddings.

$$H_{\text{in}} = H_{\text{fused}} + H_R + E_{\text{region\_emb}} + E_{\text{intra\_emb}} + E_{\text{abs\_emb}} \tag{35}$$

The result, $H_{\text{in}}$, is a unified representation that incorporates temporal, frequency, raw signal, and spatial information, ready for a downstream model.

# F DATASETS

Table 6 provides a detailed record of the nine public EEG datasets used to construct the large-scale pre-training corpus, a critical prerequisite for training a universal foundation model. The table not only lists basic parameters such as the name, sampling rate, and channel count for each dataset but also highlights the diversity of their origins in the "Description" column. This includes recordings from resting-state conditions (e.g., REEG-BACA), emotion induction tasks (e.g., Emobrain, SEED-series), and large-scale clinical data (e.g., TUEG, CAUEEG). This significant heterogeneity in recording equipment, experimental paradigms, and subject populations provides a robust data foundation for the model to learn truly generalizable and resilient neural representations.

Table 6: Information of datasets used for pre-training.

| Dataset | Rate (Hz) | Channels | Time (H) | Subjects | Description |
|---|---|---|---|---|---|
| Emobrain (Savran[1] et al., 2006) | 1024 | 64 | 4.94 | 16 | The multimodal emotion dataset contains recordings from 16 participants. Emotional states were induced by presenting the subjects with a curated selection of stimuli from the International Affective Picture System (IAPS) dataset. |
| REEG-BACA (Getzmann et al., 2024) | 1000 | 64 | 121.6 | 608 | The dataset is composed of 64-channel resting-state EEG recordings from an initial cohort of 608 participants, of whom 61.8% were female, with an age range of 20 to 70 years. Furthermore, a longitudinal component of the study involved follow-up measurements for 208 of these participants. |
| SEED-series (Zheng et al., 2018; Liu et al., 2021; 2022) | 1000 | 62 | 170.54 | 51 | This series includes SEED-IV, SEED-V, SEED-GER, and SEED-FRA, with subject counts of 15, 20, 8, and 8, respectively. |
| CAUEEG (Kim et al., 2023) | 200 | 19 | 306 | 1388 | The CAUEEG dataset is recorded at Chung-Ang University Hospital from August 24, 2012, to March 12, 2020. All recordings adhered to the International 10-20 system, utilizing a linked earlobe referencing method. |
| TUEG (Obeid & Picone, 2016) | 250-1024 | 17-23 | 27100 | 14987 | Temple University Hospital EEG corpus has over 40 distinct channel setups and inconsistent recording lengths, and most data use sampling frequency of 256 Hz. |
| Raw EEG Data (Trujillo, 2020) | 256 | 64 | 34.35 | — | Datasets are in BioSemi Data Format (BDF), which were recorded during the reported Information-Integration categorization task and reported multidimensional Rule-Based categorization task. |
| BCI Competition IV-1 (Blankertz et al., 2007) | 1000 | 59 | 8.21 | 7 | The EEG data were acquired using multi-channel amplifiers, sampled at 1000 Hz, and band-pass filtered from 0.05 to 200 Hz. |
| Resting State EEG Data (Trujillo et al., 2017) | 256 | 64 | 3.04 | 22 | A total of 22 undergraduate students from Texas State University (11 female, 11 male; mean age: 21.1 ± 0.52 years; age range: 18–26) took part in this research. |
| Siena Scalp EEG Database (Detti et al., 2020) | 512 | 31 | 30.47 | 14 | Data for this study were sourced from 14 epileptic subjects, whose cerebral activity was recorded via video scalp EEG. The signals were sampled at 512 Hz, and the electrodes were arranged according to the international 10–20 system. |

Table 7 serves as the core reference for validating the generalization abilities of the Uni-NTFM model, systematically organizing the nine benchmark datasets used for downstream task evaluation. To comprehensively assess the model's performance, the selected tasks cover a range of important domains from clinical diagnostics to Brain-Computer Interfaces, such as TUAB for abnormal EEG detection, TDBrain for psychiatric disorder classification, and BCIC-IV-2a for motor imagery recognition. The table clearly specifies the dataset for each task, the number of subjects, and the clas-

Table 7: Information of datasets used for downstream evaluation.

| Task | Dataset | Rate (Hz) | Channels | Subjects | Label |
|---|---|---|---|---|---|
| Abnormal Detection | TUAB (Harati et al., 2015) | 256 | 23 | 2,383 | 2-class |
| Event Type Classification | TUEV (Harati et al., 2015) | 256 | 23 | 370 | 6-class |
| Emotion Recognition | SEED (Zheng & Lu, 2015) | 1000 | 62 | 16 | 5-class |
| Psychiatric Dysfunction Classification | TDBrain (Van Dijk et al., 2022) | 500 | 26 | 1274 | 2-class |
| Neurodegenerative Disorder Classification | ADFTD (Miltiadous et al., 2023) | 500 | 19 | 65 | 3-class |
| Motor Imagery Classification | BCIC-IV-2a (Brunner et al., 2008) | 250 | 22 | 9 | 4-class |
| Cognitive Workload Classification | Workload (Zyma et al., 2019) | 500 | 19 | 36 | 2-class |
| Sleep Staging | HMC (Alvarez-Estevez & Rijsman, 2021) | 256 | 4 | 151 | 5-class |
| Slowing Event Classification | TUSL (von Weltin et al., 2017) | 256 | 23 | 28 | 3-class |

sification target (number of labels), providing a clear context for the rigorous evaluation protocols, which include both Linear Probing and full Fine-tuning.

## G    MODEL SETTINGS OF DIFFERENT SCALES

Table 8 offers a detailed configuration settings for the four core Uni-NTFM model variants designed in this research, demonstrating the progressive scaling from the 57M-parameter Uni-NTFM$_{tiny}$ to the 1.9B-parameter Uni-NTFM$_{large}$. It clearly outlines the differences in key architectural parameters among the variants, including embedding dimension, Transformer network depth, and the number of experts in the MoE module. Furthermore, the table specifies the base hyperparameters, such as learning rate, optimizer weight decay, and gradient clipping threshold, to ensure a fair comparison between models of different scales and to support the reproducibility of the experiments.

Table 8: Configurations and hyperparameters for different variants of Uni-NTFM.

| Settings | Uni-NTFM$_{tiny}$ | Uni-NTFM$_{small}$ | Uni-NTFM$_{middle}$ | Uni-NTFM$_{large}$ |
|---|---|---|---|---|
| Model size | 57M | 427M | 912M | 1.9B |
| Emb_dim * | 256 | 512 | 512 | 768 |
| Transformer depth | 12 | 12 | 26 | 24 |
| Number of experts | 8 | 16 | 16 | 16 |
| Batch size | 128 | 128 | 64 | 32 |
| Numbers of GPU | 16 | 16 | 16 | 32 |
| GPU | NVIDIA A100-SXM4-80G | | | |
| Number of regions | 5 | | | |
| Sequence of length | 1600 | | | |
| Dropout ratio | 0.1 | | | |
| Mask ratio | 0.25 | | | |
| Number of frequency bands | 5 | | | |
| Freqency loss weight | 0.2 | | | |
| Time loss weight | 0.8 | | | |
| Learning rate | 3e-5 | | | |
| Weight decay | 1e-4 | | | |
| Total epochs | 50 | | | |
| Warmup epochs | 5 | | | |
| Gradient clipping | 1.0 | | | |
| Use AMP | True | | | |

* Emb_dim refers to Transformer Embedding dimension.

Table 9, 10, 11 collectively form the detailed technical appendix for the model scaling law experiments, ensuring the transparency and reproducibility of this part of the research. They systematically record the precise architectural settings for a series of twelve models ranging in size from 10M to 1B parameters. Readers can clearly observe how key parameters like embedding dimension, Transformer depth, and the number of experts were carefully adjusted to achieve specific model sizes. This series of detailed configurations ensures that the investigation into the relationship between model performance and parameter count was conducted under controlled and systematic conditions.

Table 9: Configurations and hyperparameters for $10M \sim 200M$ of Uni-NTFM.

| Settings | Uni-NTFM$_{size1}$ | Uni-NTFM$_{size2}$ | Uni-NTFM$_{size3}$ | Uni-NTFM$_{size4}$ |
|---|---|---|---|---|
| Model size | 10M | 50M | 100M | 200M |
| Emb_dim * | 128 | 256 | 256 | 256 |
| Transformer depth | 8 | 10 | 16 | 32 |
| Number of experts | 8 | 8 | 12 | 16 |
| Batch size | 128 | 128 | 128 | 128 |
| Numbers of GPU | 4 | 4 | 4 | 4 |
| GPU | NVIDIA A100-SXM4-80G | | | |
| Number of regions | 5 | | | |
| Sequence of length | 1600 | | | |
| Dropout ratio | 0.1 | | | |
| Mask ratio | 0.25 | | | |
| Number of frequency bands | 5 | | | |
| Freqency loss weight | 0.2 | | | |
| Time loss weight | 0.8 | | | |
| Learning rate | 3e-5 | | | |
| Weight decay | 1e-4 | | | |
| Total epochs | 50 | | | |
| Warmup epochs | 5 | | | |
| Gradient clipping | 1.0 | | | |
| Use AMP | True | | | |

\* Emb_dim refers to Transformer Embedding dimension.

Table 10: Configurations and hyperparameters for $300M \sim 600M$ of Uni-NTFM.

| Settings | Uni-NTFM$_{size5}$ | Uni-NTFM$_{size6}$ | Uni-NTFM$_{size7}$ | Uni-NTFM$_{size8}$ |
|---|---|---|---|---|
| Model size | 300M | 400M | 500M | 600M |
| Emb_dim * | 512 | 512 | 512 | 512 |
| Transformer depth | 11 | 11 | 14 | 17 |
| Number of experts | 12 | 16 | 16 | 16 |
| Batch size | 128 | 128 | 128 | 128 |
| Numbers of GPU | 8 | 8 | 8 | 8 |
| GPU | NVIDIA A100-SXM4-80G | | | |
| Number of regions | 5 | | | |
| Sequence of length | 1600 | | | |
| Dropout ratio | 0.1 | | | |
| Mask ratio | 0.25 | | | |
| Number of frequency bands | 5 | | | |
| Freqency loss weight | 0.2 | | | |
| Time loss weight | 0.8 | | | |
| Learning rate | 3e-5 | | | |
| Weight decay | 1e-4 | | | |
| Total epochs | 50 | | | |
| Warmup epochs | 5 | | | |
| Gradient clipping | 1.0 | | | |
| Use AMP | True | | | |

\* Emb_dim refers to Transformer Embedding dimension.

Table 11: Configurations and hyperparameters for $700M \sim 1B$ of Uni-NTFM.

| Settings | Uni-NTFM$_{size9}$ | Uni-NTFM$_{size10}$ | Uni-NTFM$_{size11}$ | Uni-NTFM$_{size12}$ |
|---|---|---|---|---|
| Model size | 700M | 800M | 900M | 1B |
| Emb_dim * | 512 | 512 | 512 | 768 |
| Transformer depth | 20 | 23 | 26 | 13 |
| Number of experts | 16 | 16 | 16 | 16 |
| Batch size | 64 | 64 | 64 | 64 |
| Numbers of GPU | 8 | 8 | 8 | 8 |
| GPU | NVIDIA A100-SXM4-80G | | | |
| Number of regions | 5 | | | |
| Sequence of length | 1600 | | | |
| Dropout ratio | 0.1 | | | |
| Mask ratio | 0.25 | | | |
| Number of frequency bands | 5 | | | |
| Freqency loss weight | 0.2 | | | |
| Time loss weight | 0.8 | | | |
| Learning rate | 3e-5 | | | |
| Weight decay | 1e-4 | | | |
| Total epochs | 50 | | | |
| Warmup epochs | 5 | | | |
| Gradient clipping | 1.0 | | | |
| Use AMP | True | | | |

\* Emb_dim refers to Transformer Embedding dimension.

## H    DETAILED RESULTS OF SCALING LAW

To systematically discuss the impact of model and data scale on the performance of Uni-NTFM, we conducted two controlled scaling law experiments. The specific configurations for each model and other detailed hyperparameter settings are provided in Table 9, 10, 11. The results visualized in Figure 4 intuitively demonstrate that the representational quality of Uni-NTFM scales positively with both the number of model parameters and the volume of pre-training data. The detailed data results are shown in Table 12 and 13.

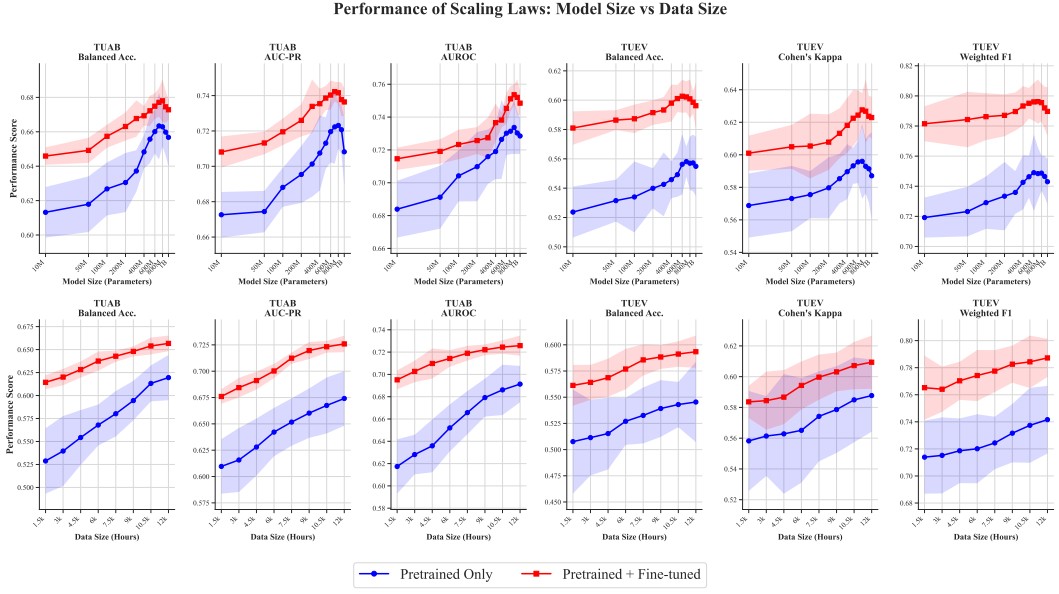

Figure 4: This figure shows the scaling laws of the Uni-NTFM model, revealing a positive correlation between performance and both model size and data volume. The top row of plots indicates that with a fixed pre-training data size, model performance steadily improves as the number of parameters increases. The bottom row shows that for a fixed model size, performance also scales positively with more of pre-training data. All plots contrast the performance under "Pretrained Only" (blue curve) and "Pretrained - Fine-tuned" (red curve) evaluation settings.

**1) Impact of Model Size:** We first evaluated the effect of model scale by training models ranging from approximately 10M to 1B parameters on a fixed pre-training corpus of 10,000 hours. As shown in Table 12, downstream performance on both TUAB and TUEV tasks exhibits a clear and positive relationship with model size. Specifically, in the fine-tuned setting on the TUAB task, the AUROC score consistently improves from 0.7146 for the 10M model to a peak of 0.7536 for the 800M model. This trend holds across all metrics for both linear probing and fine-tuning detection, confirming that larger models can learn more powerful and universal representations. However, performance appears to saturate and slightly reduce for models larger than 800M, suggesting that the corpus of 10,000 hours may be insufficient to fully release the potential of models with billion-parameter.

**2) Impact of Data Size:** We fixed the model size to our 57M parameter variant (Uni-NTFM$_{tiny}$) and varied the pre-training corpus from 1,500 to 12,000 hours. The results in Figure 4 and Table 13 indicate a strong dependence of performance on data size. On the TUEV task, for example, the fine-tuned Balanced Accuracy steadily increases from 0.5613 with 1,500 hours of data to 0.5934 with 12,000 hours. The performance curves do not show signs of saturation, indicating that even the smaller 57M model could benefit from further pre-training on a larger corpus.

Table 12 presents the core experimental data from the scaling law analysis, focusing on the impact of model size. It quantitatively demonstrates the specific performance of Uni-NTFM on the TUAB and TUEV downstream tasks as its parameter count increases from approximately 10M to 1B. The data clearly show that, in both linear probing and fine-tuning settings, larger models generally learn more powerful universal representations, as reflected by the steady improvement across most evaluation

metrics. The data in this table also reveal an interesting phenomenon: performance slightly saturates or declines when model size exceeds 800M, and we infer that the training corpus may be insufficient to fully unlock the potential of the largest models.

Table 12: Quantitative analysis of the impact of model size on TUAB and TUEV.

| Model Size | TUAB (2-class) | | | TUEV (6-class) | | |
|---|---|---|---|---|---|---|
| | Balanced Acc. | AUC-PR | AUROC | Balanced Acc. | Cohen's Kappa | Weighted F1 |
| *Only Pretrained Foundation Models (Multi-tasks)* | | | | | | |
| 10,301,200 ($\sim$10M) | $61.32_{\pm1.47}$ | $67.26_{\pm1.29}$ | $68.39_{\pm1.73}$ | $52.37_{\pm1.74}$ | $56.88_{\pm1.97}$ | $71.92_{\pm1.33}$ |
| 48,335,760 ($\sim$50M) | $61.79_{\pm1.62}$ | $67.44_{\pm1.17}$ | $69.12_{\pm1.92}$ | $53.15_{\pm1.43}$ | $57.31_{\pm2.01}$ | $72.32_{\pm1.65}$ |
| 108,808,592 ($\sim$100M) | $62.68_{\pm1.54}$ | $68.81_{\pm1.10}$ | $70.42_{\pm1.56}$ | $53.40_{\pm2.42}$ | $57.55_{\pm1.45}$ | $72.91_{\pm1.74}$ |
| 203,197,584 ($\sim$200M) | $63.05_{\pm1.72}$ | $69.54_{\pm1.61}$ | $70.98_{\pm2.11}$ | $53.99_{\pm1.37}$ | $57.97_{\pm1.88}$ | $73.34_{\pm2.27}$ |
| 299,943,056 ($\sim$300M) | $63.72_{\pm1.21}$ | $70.13_{\pm1.85}$ | $71.59_{\pm1.66}$ | $54.26_{\pm2.18}$ | $58.53_{\pm1.52}$ | $73.59_{\pm1.42}$ |
| 392,352,912 ($\sim$400M) | $64.83_{\pm0.97}$ | $70.75_{\pm2.12}$ | $71.90_{\pm1.68}$ | $54.58_{\pm1.26}$ | $58.96_{\pm1.69}$ | $74.26_{\pm1.81}$ |
| 496,317,072 ($\sim$500M) | $65.57_{\pm1.39}$ | $71.31_{\pm1.46}$ | $72.64_{\pm2.41}$ | $54.92_{\pm1.51}$ | $59.32_{\pm1.11}$ | $74.63_{\pm1.40}$ |
| 600,281,232 ($\sim$600M) | $66.01_{\pm1.27}$ | $71.96_{\pm1.74}$ | $73.02_{\pm1.33}$ | $55.63_{\pm2.06}$ | $59.56_{\pm1.30}$ | $\mathbf{74.90}_{\pm2.53}$ |
| 704,245,392 ($\sim$700M) | $\mathbf{66.34}_{\pm1.95}$ | $\mathbf{72.23}_{\pm2.09}$ | $73.14_{\pm1.40}$ | $\mathbf{55.81}_{\pm1.04}$ | $\mathbf{59.60}_{\pm2.31}$ | $74.84_{\pm1.75}$ |
| 808,209,552 ($\sim$800M) | $66.29_{\pm1.15}$ | $72.31_{\pm1.31}$ | $\mathbf{73.37}_{\pm1.61}$ | $55.70_{\pm1.82}$ | $59.29_{\pm1.66}$ | $74.87_{\pm1.18}$ |
| 912,173,712 ($\sim$900M) | $65.98_{\pm1.36}$ | $72.07_{\pm1.40}$ | $73.03_{\pm1.29}$ | $55.72_{\pm1.94}$ | $59.14_{\pm2.10}$ | $74.65_{\pm1.37}$ |
| 1,035,695,760 ($\sim$1B) | $65.68_{\pm1.70}$ | $70.82_{\pm1.92}$ | $72.85_{\pm1.08}$ | $55.49_{\pm2.01}$ | $58.71_{\pm2.67}$ | $74.30_{\pm1.49}$ |
| *Pretrained and Fine-tuned Foundation Models (Single-task)* | | | | | | |
| 10,301,200 ($\sim$10M) | $64.59_{\pm0.51}$ | $70.81_{\pm0.88}$ | $71.46_{\pm0.68}$ | $58.10_{\pm1.14}$ | $60.10_{\pm1.08}$ | $78.15_{\pm1.16}$ |
| 48,335,760 ($\sim$50M) | $64.92_{\pm0.73}$ | $71.32_{\pm0.64}$ | $71.91_{\pm0.74}$ | $58.64_{\pm0.68}$ | $60.49_{\pm1.35}$ | $78.43_{\pm1.84}$ |
| 108,808,592 ($\sim$100M) | $65.74_{\pm0.67}$ | $71.95_{\pm0.75}$ | $72.33_{\pm0.90}$ | $58.73_{\pm0.97}$ | $60.54_{\pm1.99}$ | $78.62_{\pm1.56}$ |
| 203,197,584 ($\sim$200M) | $66.31_{\pm0.82}$ | $72.60_{\pm0.91}$ | $72.57_{\pm0.82}$ | $59.15_{\pm0.84}$ | $60.78_{\pm1.71}$ | $78.71_{\pm1.39}$ |
| 299,943,056 ($\sim$300M) | $66.77_{\pm1.06}$ | $73.39_{\pm1.52}$ | $72.74_{\pm1.24}$ | $59.33_{\pm0.89}$ | $61.32_{\pm1.54}$ | $78.96_{\pm1.27}$ |
| 392,352,912 ($\sim$400M) | $66.93_{\pm0.58}$ | $73.54_{\pm0.85}$ | $73.66_{\pm1.15}$ | $59.80_{\pm1.26}$ | $61.81_{\pm1.68}$ | $79.33_{\pm1.31}$ |
| 496,317,072 ($\sim$500M) | $67.22_{\pm0.88}$ | $73.86_{\pm0.69}$ | $73.82_{\pm0.47}$ | $60.11_{\pm0.93}$ | $62.25_{\pm1.73}$ | $79.51_{\pm1.02}$ |
| 600,281,232 ($\sim$600M) | $67.49_{\pm0.75}$ | $74.03_{\pm0.82}$ | $74.53_{\pm0.95}$ | $\mathbf{60.27}_{\pm1.12}$ | $62.44_{\pm1.52}$ | $79.60_{\pm1.26}$ |
| 704,245,392 ($\sim$700M) | $67.71_{\pm0.83}$ | $\mathbf{74.22}_{\pm0.56}$ | $75.11_{\pm0.79}$ | $60.22_{\pm1.07}$ | $\mathbf{62.76}_{\pm1.04}$ | $\mathbf{79.62}_{\pm1.47}$ |
| 808,209,552 ($\sim$800M) | $\mathbf{67.80}_{\pm1.24}$ | $74.16_{\pm0.61}$ | $\mathbf{75.36}_{\pm0.62}$ | $60.09_{\pm1.32}$ | $62.68_{\pm1.95}$ | $79.55_{\pm1.25}$ |
| 912,173,712 ($\sim$900M) | $67.45_{\pm0.92}$ | $73.77_{\pm0.97}$ | $75.20_{\pm1.09}$ | $59.86_{\pm0.79}$ | $62.37_{\pm1.26}$ | $79.20_{\pm1.41}$ |
| 1,035,695,760 ($\sim$1B) | $67.28_{\pm0.61}$ | $73.64_{\pm0.76}$ | $74.85_{\pm0.71}$ | $59.62_{\pm1.55}$ | $62.29_{\pm1.28}$ | $78.97_{\pm1.61}$ |

Table 13: Quantitative analysis of the impact of data size on n TUAB and TUEV.

| Data Size | TUAB (2-class) | | | TUEV (6-class) | | |
|---|---|---|---|---|---|---|
| | Balanced Acc. | AUC-PR | AUROC | Balanced Acc. | Cohen's Kappa | Weighted F1 |
| *Only Pretrained Foundation Models (Multi-tasks)* | | | | | | |
| $\sim$1500H | $52.88_{\pm3.56}$ | $60.96_{\pm2.59}$ | $61.74_{\pm2.44}$ | $50.76_{\pm4.97}$ | $55.82_{\pm3.25}$ | $71.39_{\pm2.70}$ |
| $\sim$3000H | $53.96_{\pm3.81}$ | $61.57_{\pm3.04}$ | $62.81_{\pm1.78}$ | $51.14_{\pm3.61}$ | $56.14_{\pm2.60}$ | $71.52_{\pm2.81}$ |
| $\sim$4500H | $55.43_{\pm2.99}$ | $62.80_{\pm2.71}$ | $63.59_{\pm2.37}$ | $51.53_{\pm3.43}$ | $56.28_{\pm3.89}$ | $71.86_{\pm2.39}$ |
| $\sim$6000H | $56.79_{\pm2.23}$ | $64.22_{\pm2.30}$ | $65.20_{\pm2.11}$ | $52.70_{\pm2.25}$ | $56.51_{\pm3.40}$ | $72.01_{\pm2.55}$ |
| $\sim$7500H | $58.02_{\pm2.47}$ | $65.17_{\pm2.25}$ | $66.57_{\pm1.94}$ | $53.26_{\pm2.67}$ | $57.42_{\pm2.93}$ | $72.44_{\pm1.94}$ |
| $\sim$9000H | $59.45_{\pm2.15}$ | $66.02_{\pm2.35}$ | $67.92_{\pm1.70}$ | $53.92_{\pm2.71}$ | $57.86_{\pm2.82}$ | $73.16_{\pm2.16}$ |
| $\sim$10500H | $61.32_{\pm1.96}$ | $66.75_{\pm2.66}$ | $68.61_{\pm2.26}$ | $54.30_{\pm3.83}$ | $58.49_{\pm2.76}$ | $73.74_{\pm2.77}$ |
| $\sim$12000H | $\mathbf{61.95}_{\pm2.46}$ | $\mathbf{67.41}_{\pm2.59}$ | $\mathbf{69.13}_{\pm1.63}$ | $\mathbf{54.54}_{\pm3.83}$ | $58.77_{\pm2.35}$ | $\mathbf{74.16}_{\pm2.50}$ |
| *Pretrained and Fine-tuned Foundation Models (Single-task)* | | | | | | |
| $\sim$1500H | $61.44_{\pm0.76}$ | $67.60_{\pm0.69}$ | $69.51_{\pm0.85}$ | $56.13_{\pm1.95}$ | $58.36_{\pm1.05}$ | $76.52_{\pm2.37}$ |
| $\sim$3000H | $62.01_{\pm0.90}$ | $68.44_{\pm0.93}$ | $70.27_{\pm0.97}$ | $56.42_{\pm1.67}$ | $58.44_{\pm1.89}$ | $76.41_{\pm1.67}$ |
| $\sim$4500H | $62.83_{\pm0.88}$ | $69.11_{\pm0.85}$ | $70.98_{\pm1.35}$ | $56.86_{\pm1.82}$ | $58.67_{\pm1.76}$ | $77.03_{\pm1.41}$ |
| $\sim$6000H | $63.75_{\pm1.04}$ | $70.02_{\pm0.68}$ | $71.43_{\pm0.75}$ | $57.69_{\pm1.93}$ | $59.43_{\pm1.56}$ | $77.42_{\pm1.89}$ |
| $\sim$7500H | $64.28_{\pm0.61}$ | $71.24_{\pm0.54}$ | $71.90_{\pm0.66}$ | $58.56_{\pm1.49}$ | $59.97_{\pm1.47}$ | $77.75_{\pm1.55}$ |
| $\sim$9000H | $64.80_{\pm0.55}$ | $71.96_{\pm0.81}$ | $72.21_{\pm0.48}$ | $58.84_{\pm1.14}$ | $60.31_{\pm1.26}$ | $78.26_{\pm1.37}$ |
| $\sim$10500H | $65.39_{\pm0.92}$ | $72.32_{\pm0.56}$ | $72.45_{\pm0.61}$ | $59.12_{\pm1.22}$ | $60.72_{\pm1.54}$ | $78.43_{\pm1.94}$ |
| $\sim$12000H | $\mathbf{65.67}_{\pm0.85}$ | $\mathbf{72.59}_{\pm0.79}$ | $72.58_{\pm0.90}$ | $\mathbf{59.34}_{\pm1.54}$ | $\mathbf{60.94}_{\pm1.73}$ | $\mathbf{78.72}_{\pm1.42}$ |

Table 13 complements the scaling law analysis by validating its other critical dimension: the importance of data volume. With the model size fixed to the 57M-parameter Uni-NTFM$_{tiny}$ variant, the table documents how model performance on the TUAB and TUEV tasks evolves as the amount of pre-training data increases from approximately 1,500 to 12,000 hours. The results show a clear and consistent upward trend in performance with more data, for both linear probing and fine-tuning evaluations.

# I  T-SNE VISUALIZATION OF LEARNED FEATURE REPRESENTATIONS

To provide a qualitative and intuitive assessment of our model's representation learning abilities, we employed the t-SNE dimensionality reduction technique to visualize the learned feature spaces. Figure 5 shows this analysis on two distinct downstream tasks: the 4-class motor imagery task (BCIC-IV-2a) and the more complex 6-class clinical event detection task (TUEV).

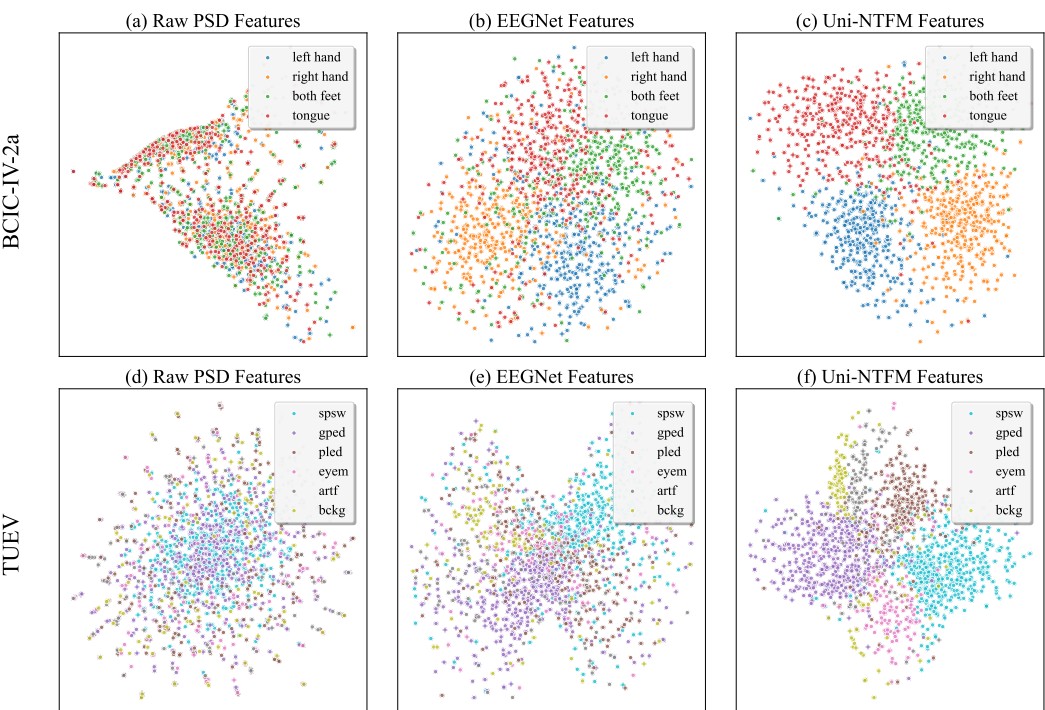

Figure 5: t-SNE Visualization of Learned Feature Representations. This figure provides a qualitative comparison of feature spaces learned on the BCIC-IV-2a (top row) and TUEV (bottom row) datasets. The columns represent features from different sources: (a, d) Raw Power Spectral Density (PSD) features, which serve as a baseline; (b, e) features extracted from a trained EEGNet model, representing a standard deep learning approach; and (c, f) features from our only pre-trained Uni-NTFM model. Each color corresponds to a distinct class within the respective dataset. The clear formation of well-separated and compact clusters in the rightmost column (c, f) visually demonstrates Uni-NTFM's superior ability to learn discriminative and generalizable neural representations.

As expected, (a, d) exhibit no discernible class structure, with points from all classes aggregated in a single and messy cloud. This establishes the inherent difficulty of separating these classes directly from traditional features. The middle column (b, e) shows the feature space after processing by a trained, task-specific EEGNet model. They remain diffuse and suffer from significant overlap at their boundaries. This indicates that while standard deep learning architectures can learn some useful patterns, they struggle to create truly separable representations, especially for the more challenging TUEV dataset.

In contrast, the column (c, f) shows the feature space learned by our pre-trained Uni-NTFM. Even for the challenging 6-class TUEV task, Uni-NTFM effectively disentangles the different clinical event types into distinct regions of the feature space. This visual evidence corroborates our quantitative results, confirming that the paradigm of Uni-NTFM enables it to learn far more powerful and generalizable representations than standard methods.

# J    EFFICIENCY ANALYSIS OF UNI-NTFM AND DENSE BASELINES

Table 14: Efficiency Comparison between Uni-NTFM$_{small}$ (MoE) and Dense Baselines. **Same-FLOPs** denotes matched inference compute, while **Same-Params** denotes matched total capacity.

| Model Variant | Architecture | Total Params (Capacity) | Active Params (Inference Cost) | Total GFLOPs | Inference GFLOPs |
|---|---|---|---|---|---|
| Dense-Small (Same-FLOPs) | Dense Transformer | 74 M | 74 M | 16.08 | 16.08 |
| Dense-Large (Same-Params) | Dense Transformer | 421 M | 421 M | 93.15 | 93.15 |
| **Uni-NTFM$_{small}$ (Ours)** | **Sparse MoE** | **427 M** | **74 M** | **100.40** | **15.87** |

*Note: Active Params for MoE assumes Top-2 gating with 16 experts.*

To evaluate the effectiveness of the proposed MoE architecture, we conducted a controlled comparison between Uni-NTFM$_{small}$ and two dense baselines: Dense-Small (matched for inference compute, denoted as Same-FLOPs) and Dense-Large (matched for total parameter capacity, denoted as Same-Params). As illustrated in Table 14, Uni-NTFM$_{small}$ successfully decouples model capacity from computational cost. The model possesses a total parameter count of 427 M, comparable to the Dense-Large baseline (421 M). This ensures the model retains a high capacity for knowledge encoding. Besides, through sparse activation (Top-2 gating), the active parameters during inference are limited to 74 M, resulting in an inference cost of 15.87 GFLOPs. Notably, this is even lower than the Dense-Small baseline (16.08 GFLOPs).This structural advantage confirms that Uni-NTFM allows for the scaling of total parameters without the increase in deployment latency.

Table 15: Performance Comparison on Downstream Tasks.

| Method | Workload (2-class) | | | ADFTD (3-class) | | | BCIC-IV-2a (4-class) | | |
|---|---|---|---|---|---|---|---|---|---|
| | Balanced Acc. | AUC-PR | AUROC | Balanced Acc. | Cohen's Kappa | Weighted F1 | Balanced Acc. | Cohen's Kappa | Weighted F1 |
| *Only Pretrained Foundation Models (Multi-tasks)* | | | | | | | | | |
| Dense-Small (Same-FLOPs) | 62.43 | 64.62 | 68.90 | 65.61 | 66.25 | 69.82 | 49.79 | 39.21 | 49.58 |
| Dense-Large (Same-Params) | 65.29 | 69.14 | 71.32 | 70.04 | 71.91 | 71.83 | 52.45 | 40.47 | 51.82 |
| Uni-NTFM$_{small}$ | 64.11 | 67.71 | 70.26 | 67.35 | 67.44 | 71.31 | 52.09 | 40.44 | 51.23 |
| *Pretrained and Fine-tuned Foundation Models (Single-task)* | | | | | | | | | |
| Dense-Small (Same-FLOPs) | 63.97 | 68.11 | 70.30 | 74.35 | 73.97 | 76.57 | 51.83 | 40.18 | 52.06 |
| Dense-Large (Same-Params) | 67.35 | 73.22 | 75.06 | 76.63 | 75.84 | 77.02 | 55.75 | 42.13 | 53.61 |
| Uni-NTFM$_{small}$ | 66.72 | 71.73 | 73.88 | 75.68 | 76.32 | 76.80 | 55.59 | 42.01 | 54.65 |

Table 15 details the performance on downstream tasks. Under the same computational budget, Uni-NTFM$_{small}$ consistently outperforms Dense-Small across all metrics. For instance, in the fine-tuning setting for the ADFTD task, Uni-NTFM achieves a Balanced Accuracy of 75.68%, surpassing Dense-Small (74.35%) clearly. Similarly, in the BCIC-IV-2a task, Uni-NTFM (54.65%) significantly exceeds Dense-Small (52.06%) in Weighted F1. This demonstrates that the additional inactive parameters in the MoE architecture contribute significantly to representation quality without increasing inference costs. Besides, despite activating only 17% of its parameters, Uni-NTFM$_{small}$ achieves performance highly comparable to the fully activated Dense-Large model. In the challenging BCIC-IV-2a fine-tuning task, Uni-NTFM$_{small}$ even surpasses Dense-Large in terms of Weighted F1 (54.65% vs. 53.61%). While there is a performance gap in some tasks, it is acceptable compared to the reduction in inference GFLOPs (15.87 vs. 93.15). These results shows that Uni-NTFM effectively uses the learning capacity of large-scale models while maintaining the inference agility of small-scale models, validating the necessity of the MoE design for EEG foundation model.

## K  IMPACT OF DATA AUGMENTATION

Table 16: Performance Comparison on Workload, ADFTD, and BCIC-IV-2a.

| Method | Workload (2-class) | | | ADFTD (3-class) | | | BCIC-IV-2a (4-class) | | |
|---|---|---|---|---|---|---|---|---|---|
| | Balanced Acc. | AUC-PR | AUROC | Balanced Acc. | Cohen's Kappa | Weighted F1 | Balanced Acc. | Cohen's Kappa | Weighted F1 |
| *Only Pretrained Foundation Models (Multi-tasks)* | | | | | | | | | |
| Uni-NTFM$_{small}$ (*w/o Aug.*) | 62.83 | 66.95 | 69.79 | 66.52 | 67.06 | 70.10 | 51.44 | 40.15 | 50.78 |
| Uni-NTFM$_{small}$ (*w/ Aug.*) | 64.11 | 67.71 | 70.26 | 67.35 | 67.44 | 71.31 | 52.09 | 40.44 | 51.23 |
| *Pretrained and Fine-tuned Foundation Models (Single-task)* | | | | | | | | | |
| Uni-NTFM$_{small}$ (*w/o Aug.*) | 65.39 | 69.47 | 71.91 | 75.20 | 74.34 | 76.13 | 54.63 | 41.35 | 54.06 |
| Uni-NTFM$_{small}$ (*w/ Aug.*) | 66.72 | 71.73 | 73.88 | 75.68 | 76.32 | 76.80 | 55.59 | 42.01 | 54.65 |

To quantify the contribution of the data augmentation strategy ($f_{aug}$) introduced in Section 2.1, we conducted an ablation study across three distinct downstream tasks: Workload, ADFTD, and BCIC-IV-2a. The results are summarized in Table 16.

The exclusion of augmentation results in a noticeable reduction in feature quality in the Only Pre-trained setting. For instance, on the Workload dataset, the Balanced Accuracy decreases from 64.11% to 62.83%, and on the BCIC-IV-2a dataset, it drops from 52.09% to 51.44%. This indicates that simulating noise, channel loss, and temporal drift during pre-training is essential for learning representations that are invariant to common EEG signals. In the Fine-tuned setting, the Weighted F1 score on BCIC-IV-2a drops from 54.65% to 54.06%, and the Balanced Accuracy on the Workload task declines by 1.33%. These results domenstrate augmentation enhances generalization and robustness, and removing data augmentation leads to a performance degradation across all datasets and evaluation settings. Notably, on the ADFTD dataset, the model without augmentation still achieves a Balanced Accuracy of 75.20% of Balanced Accuracy, which is similer to the 75.68% achieved with augmentation. This suggests that while our data augmentation module ($f_{aug}$) provides a valuable performance boost by enhancing robustness against signal variations, the core abilities of the model are primarily from the proposed MoE architecture.

The results confirm that the proposed data augmentation strategy effectively improves the model's generalization ability. Besides, the high baseline performance of the unaugmented model serves as strong evidence for the effectiveness of the Uni-NTFM framework.

## L  ABLATION STUDY OF TOPOLOGICAL EMBEDDING COMPONENTS

Table 17: Detailed Ablation Study of Topological Embedding Components on TUEV dataset.

| Variant | $E_{abs}$ | $E_{region}$ | $E_{intra}$ | Balanced Acc. (%) | Cohen's Kappa (%) | Weighted F1 (%) |
|---|---|---|---|---|---|---|
| Variant A | ✗ | ✗ | ✗ | 63.16 | 64.80 | 80.19 |
| Variant B | ✓ | ✗ | ✗ | 63.57 | 65.12 | 80.70 |
| Variant C | ✗ | ✓ | ✗ | 62.72 | 64.98 | 80.35 |
| Variant D | ✗ | ✓ | ✓ | 63.81 | 65.63 | 81.24 |
| **Uni-NTFM**$_{small}$ | ✓ | ✓ | ✓ | **64.05** | **65.82** | **81.74** |

We performed a fine-grained ablation study to decouple the contributions of the three components within our Topological Embedding (TE) module: Region Embedding ($E_{region}$), Intra-region Embedding ($E_{intra}$), and Global Absolute Embedding ($E_{abs}$). The results on the TUEV dataset are presented in Table 17.

Variant B with only the global absolute index obtains a Balanced Accuracy of 63.57%, but the improvement is limited than the baseline Variant A. This suggests that treating EEG electrodes merely as a flat sequence fails to capture the intricate spatial relationships required for effective decoding. Notably, employing region-level embeddings alone results in a performance reduce to 62.72%, which is lower than the baseline. We think that providing only regional information introduces ambiguity, as the model cannot distinguish between different electrodes within the same brain region, leading to a loss of channel-specific resolution. However, when intra-region embeddings are introduced to resolve the ambiguity of Variant C, performance significantly improves to 63.81%. This highlights the necessity of the hierarchical structure: $E_{region}$ provides the macroscopic functional

context, while $E_{intra}$ restores the microscopic spatial resolution. This combination validates that structured spatial priors are more effective than arbitrary sequence indices.

Uni-NTFM$_{small}$ achieves the highest performance across all metrics by integrating all three components. This demonstrates that $E_{abs}$ provides a unique global identifier to ensure absolute channel independence, while the $E_{region}$ and $E_{intra}$ injects neuroanatomical topology. The complete TE module establishes a robust neural coordinate system that is essential for representation ability.

Table 18: Stress Test: Robustness to Missing Channels.

| Dataset | Metric | Method | Original $(k=0)$ | Randomly Dropped Channels ($k$) | | | |
|---|---|---|---|---|---|---|---|
| | | | | $k=1$ | $k=3$ | $k=5$ | $k=7$ |
| BCIC-IV-2a | Balanced Accuracy | Uni-NTFM$_{small}$ (w/o TE) | 52.39 | 52.46 (+0.07%) | 50.14 (-2.25%) | 45.93 (-6.46%) | 40.40 (-11.99%) |
| | | Uni-NTFM$_{small}$ (w/ TE) | 55.59 | 55.31 (-0.28%) | 54.52 (-1.07%) | 51.86 (-3.73%) | 47.25 (-8.34%) |
| | Cohen's Kappa | Uni-NTFM$_{small}$ (w/o TE) | 40.87 | 40.50 (-0.37%) | 37.68 (-3.19%) | 33.93 (-6.94%) | 32.12 (-8.75%) |
| | | Uni-NTFM$_{small}$ (w/ TE) | 42.01 | 41.83 (-0.18%) | 40.46 (-1.55%) | 38.61 (-3.40%) | 36.10 (-5.91%) |
| | Weighted F1 | Uni-NTFM$_{small}$ (w/o TE) | 53.23 | 52.81 (-0.42%) | 51.36 (-1.87%) | 48.12 (-5.11%) | 44.97 (-8.26%) |
| | | Uni-NTFM$_{small}$ (w/ TE) | 54.65 | 54.71 (+0.06%) | 53.93 (-0.72%) | 51.70 (-2.95%) | 49.24 (-5.41%) |
| SEED | Balanced Accuracy | Uni-NTFM$_{small}$ (w/o TE) | 66.85 | 66.97 (+0.12%) | 66.42 (-0.43%) | 65.10 (-1.75%) | 63.77 (-3.08%) |
| | | Uni-NTFM$_{small}$ (w/ TE) | 72.02 | 71.98 (-0.04%) | 71.67 (-0.35%) | 71.10 (-0.92%) | 70.18 (-1.84%) |
| | Cohen's Kappa | Uni-NTFM$_{small}$ (w/o TE) | 55.42 | 55.35 (-0.07%) | 55.12 (-0.30%) | 54.01 (-1.41%) | 53.30 (-2.12%) |
| | | Uni-NTFM$_{small}$ (w/ TE) | 58.59 | 58.68 (+0.09%) | 58.55 (-0.04%) | 58.51 (-0.38%) | 57.93 (-0.96%) |
| | Weighted F1 | Uni-NTFM$_{small}$ (w/o TE) | 68.98 | 68.87 (-0.11%) | 68.84 (-0.14%) | 68.09 (-0.89%) | 67.03 (-1.95%) |
| | | Uni-NTFM$_{small}$ (w/ TE) | 72.74 | 72.79 (+0.05%) | 72.66 (-0.07%) | 72.50 (-0.24%) | 71.93 (-0.81%) |

*Note: (w/ TE) denotes the model equipped with TE module, (w/o TE) means without the TE module.*

Table 19: Cross-Montage Generalization on SEED.

| Method | Training Montage | Inference Montage | Performance Metrics (%) | | |
|---|---|---|---|---|---|
| | | | Balanced Accuracy | Cohen's Kappa | Weighted F1 |
| Uni-NTFM$_{small}$ (w/ TE) | 62 Channels | 62 Channels | 72.02 | 58.59 | 72.74 |
| Uni-NTFM$_{small}$ (w/o TE) | 62 Channels | 19 Channels | 61.45 | 45.52 | 62.41 |
| Uni-NTFM$_{small}$ (w/ TE) | 62 Channels | 19 Channels | 65.88 | 51.16 | 66.73 |

To simulate real-world sensor malfunctions or signal interruptions, we randomly masked $k \in \{1, 3, 5, 7\}$ channels during the inference phase on both the BCIC-IV-2a (22 channels) and SEED (62 channels) datasets. We compare the performance reduction rate of Uni-NTFM$_{small}$(w/ TE) against Uni-NTFM$_{small}$(w/o TE) to quantify robustness. Bssides, to evaluate the model's ability to generalize across different layouts without retraining, we fine-tuned the model on the high-density SEED dataset (62 channels) but evaluated exclusively on the sparse sub-montage corresponding to the international 10-20 system (19 channels).

In Table 18, the Uni-NTFM$_{small}$(w/ TE) model consistently exhibits a slower rate of performance decay compared to the Uni-NTFM$_{small}$(w/o TE). For instance, on the BCIC-IV-2a dataset, when 7 channels are dropped (representing 30% information loss for a 22-channel setup), the Balanced Accuracy of the baseline plummets by 11.99%. In contrast, the model with TE mitigates this loss to 8.34%, effectively alleviating performance reduction. On the 62-channel SEED dataset, the effect of TE is obvious. Even with 7 channels missing, the Weighted F1 score of Uni-NTFM$_{small}$(w/ TE) drops by only 0.81%, while the Uni-NTFM$_{small}$(w/o TE) drops by nearly 2%. These results confirms that TE successfully encodes a latent functional topography. Instead of treating channel loss as the sequence loss, the model uses spatial priors ($E_{intra}$ and $E_{region}$) to infer missing information from neighboring electrodes.

Table 19 evaluates a realistic transfer scenario, where a model trained on a high-density research montage is deployed on a sparse standard clinical montage without re-training. The Uni-NTFM$_{small}$(w/o TE) suffers a dramatic performance reduction, with Cohen's Kappa dropping to 45.52%. This indicates that standard positional encodings fail to adapt to the spatial change caused by the different montage. By explicitly injecting topological identity, Uni-NTFM$_{small}$(w/ TE) maintains a robust Cohen's Kappa of 51.16% and a Balanced Accuracy of 65.88%. This result strongly supports our claim that TE effectively unifies diverse montages by anchoring them to a common neural coordinate system.

## M COMPARISON WITH STFT BASELINE ON WORKLOAD AND TUEV

Table 20: Comparison with STFT Baseline.

| Method | Input Design | Infer. GFLOPs | Workload (2-class) | | | TUEV (6-class) | | |
|---|---|---|---|---|---|---|---|---|
| | | | Bal. Acc. | AUC-PR | AUROC | Bal. Acc. | Kappa | W-F1 |
| STFT-MoE (Baseline) | STFT (Mag. + Phase) | 16.14 | 62.32 | 68.39 | 68.81 | 52.37 | 54.26 | 76.92 |
| **Uni-NTFM$_{small}$ (Ours)** | **Decoupled (Time + Freq)** | **15.87** | 66.72 | 71.73 | 73.88 | 66.94 | 67.96 | 83.25 |

*Note: The STFT-MoE baseline feeds concatenated STFT magnitude and phase spectrograms into the MoE-Transformer. Results correspond to the fine-tuned setting. Bal. Acc.: Balanced Accuracy; Kappa: Cohen's Kappa; W-F1: Weighted F1.*

To rigorously validate the necessity of our proposed multi-stream architecture (HFPM) and dual-domain fusion (DCM), we implemented STFT-MoE as the baseline. This baseline feeds stacked STFT magnitude and phase spectrograms into the same MoE backbone used by Uni-NTFM, with patch size and embedding dimensions adjusted to match inference GFLOPs (≈16 GFLOPs).

As shown in Table 20, while maintaining a lower computational cost, Uni-NTFM$_{small}$ significantly outperforms the STFT-MoE baseline. On the TUEV dataset, Uni-NTFM achieves a Balanced Accuracy of 66.94%, surpassing the STFT-MoE baseline by a remarkable 14.57%. Similarly, Cohen's Kappa improves by 13.7%. On the Workload dataset, our model maintains a superior performance, improving Balanced Accuracy by 4.4% and AUROC by 5.07% compared to the STFT-MoE. The results means that our HFPM can preserve the high temporal resolution of waveform morphology of time-domain, and capture pectral rhythms of frequency-domain. Furthermore, the STFT-MoE baseline relies on irregular fusion, while our DCM enables context-aware fusion. This allows the model to demonstrate the superior robustness across diverse tasks.

## N LABEL EFFICIENCY ANALYSIS

Table 21: Label Efficiency Analysis.

| Dataset | Metric | Labeled Data | | | | |
|---|---|---|---|---|---|---|
| | | 1% | 10% | 30% | 50% | 70% |
| **SEED (3-class)** | Balanced Acc. | 67.04 | 68.85 | 70.97 | 71.60 | 72.02 |
| | Cohen's Kappa | 52.38 | 56.23 | 57.63 | 58.28 | 58.59 |
| | Weighted F1 | 67.40 | 69.90 | 71.77 | 72.53 | 72.74 |
| **Workload (2-class)** | Balanced Acc. | 61.34 | 64.21 | 65.69 | 66.31 | 66.72 |
| | AUC-PR | 64.52 | 67.66 | 70.65 | 71.26 | 71.73 |
| | AUROC | 66.19 | 70.35 | 72.47 | 73.30 | 73.88 |

To evaluate the label efficiency of Uni-NTFM, we conducted fine-tuning experiments on the SEED and Workload datasets using varying fractions of labeled data, ranging from 1% to 70%. The results presented in Table 21 demonstrate that our model significantly reduces the dependency on large-scale annotations. The model exhibits remarkable robustness in extreme low-resource settings. On the Workload dataset, with only 10% of the labeled data, Uni-NTFM achieves a Balanced Accuracy of 64.21%. When compared to the performance at 70% data (66.72%), the model retains approximately 92% of its peak performance using merely a portion of the training samples. This suggests that the pre-trained backbone has already learned highly discriminative representations, requiring very few examples to align with the downstream task. These results validate that the pre-training of Uni-NTFM successfully instills universal neural priors, enabling the model to generalize effectively even when labeled data is limited. This characteristic is particularly valuable for BCI applications where data labeling is expensive and time-consuming.

## O    VISUAL ANALYSIS OF MOE

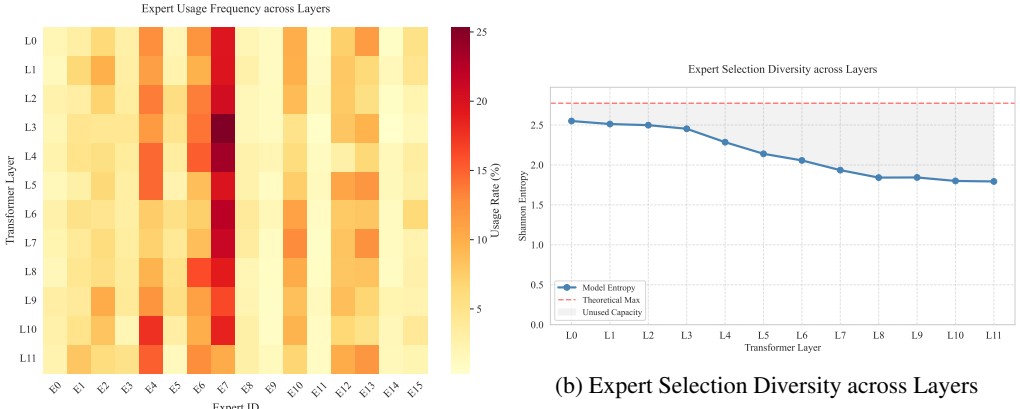

(a) Expert Usage Frequency across Layers

(b) Expert Selection Diversity across Layers

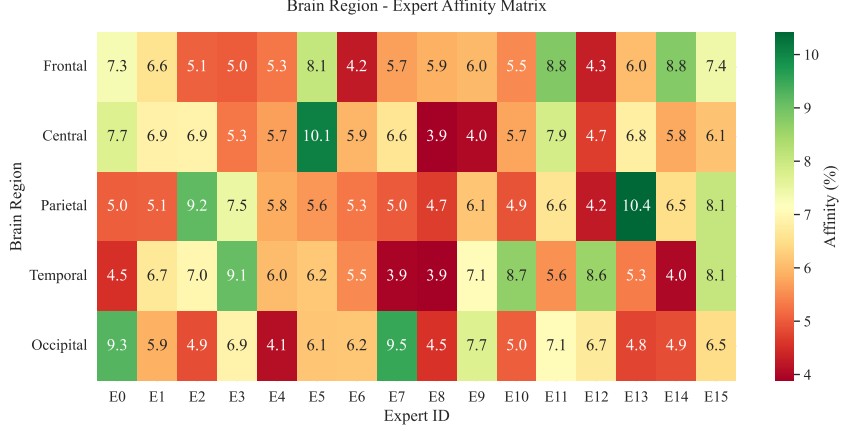

(c) Brain Region - Expert Affinity Matrix

Figure 6: Visual analysis of MoE on the Workload dataset.

To validate the functional specialization of our Mixture-of-Experts (MoE) module and address concerns regarding expert redundancy, we conducted diagnostic analyses on three diverse downstream datasets: Workload, TUEV, and BCIC-IV-2a, and the results are shown in Fig 6, Fig 7, and 8. The visualizations reveal highly consistent and interpretable patterns of expert behavior across all tasks. In detail, **Expert Usage Frequency heatmap** (subfigures a) displays the activation frequency of each expert across different Transformer layers. A darker color indicates higher usage, revealing whether specific experts dominate processing at certain depths. Furthermore, **Expert Selection Diversity curve** (subfigures b) tracks the Shannon Entropy of the expert gating distribution layer-by-layer. High entropy signifies broad collaboration, while low entropy indicates concentrated and specialized processing. Moreover, **Brain Region - Expert Affinity Matrix** (subfigures c) quantifies the routing probability of signals from specific brain regions to specific experts. It serves as direct evidence of whether experts specialize in processing spatially distinct neural patterns.

Across all three datasets, deep layers of the Expert Usage Frequency heatmaps (subfigures a) consistently show concentrated activation of specific experts. Some experts are heavily activated (dark red), while others remain inactive, which indicates that the router has learned to selectively assigned tokens to specific experts based on their features, rather than randomly distributing the load.

In Expert Selection Diversity curve (subfigures b), high entropy in shallow layers demonstrates that experts collaborate to extract features, and the reduction of entropy in middle layers signifies a transition towards specialized processing, where specific experts are responsible for distinct signal

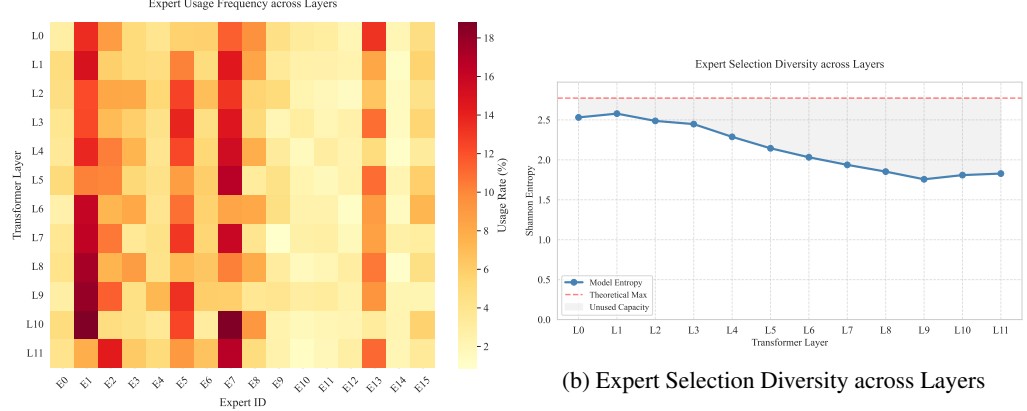

(a) Expert Usage Frequency across Layers

(b) Expert Selection Diversity across Layers

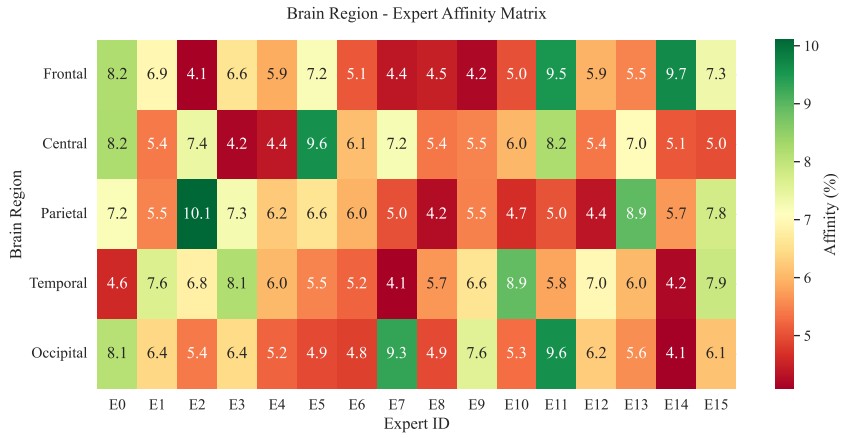

(c) Brain Region - Expert Affinity Matrix

Figure 7: Visual analysis of MoE on the TUEV dataset.

components. Besides, the Brain Region-Expert Affinity matrix (subfigures c) can confirm that the MoE effectively uses the spatial priors injected by our Topological Embedding module, and routes signals from different functional brain areas to experts forspecialized processing. These results confirm that Uni-NTFM's experts are functionally specialized modules that dynamically adapt to both the hierarchical depth of the network and the task-relevant spatial topology of the brain.

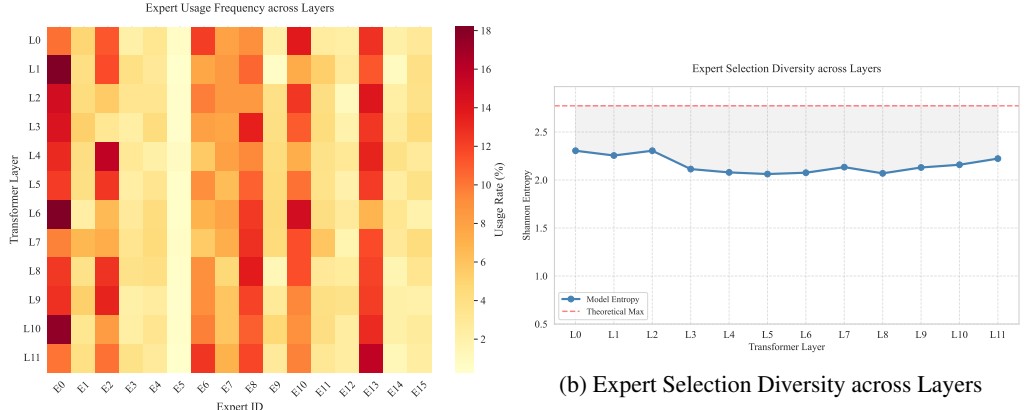

(a) Expert Usage Frequency across Layers

(b) Expert Selection Diversity across Layers

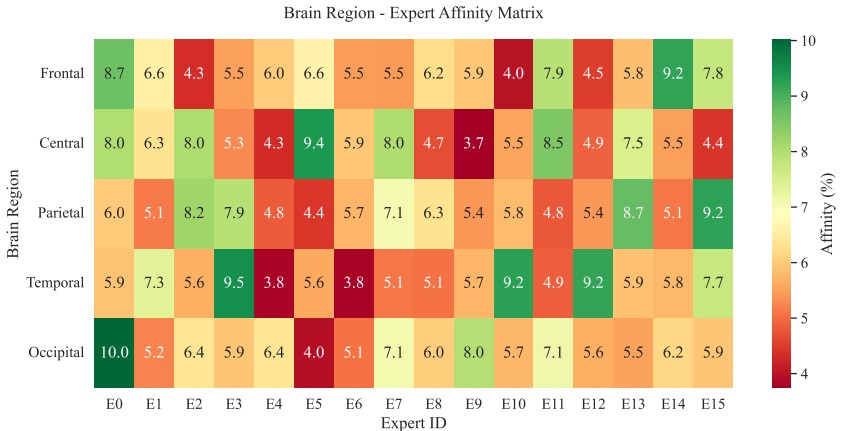

(c) Brain Region - Expert Affinity Matrix

Figure 8: Visual analysis of MoE on the BCIC-IV-2a dataset.

