# OpenReview forum: "Uni-NTFM: A Unified Foundation Model for EEG Signal Representation Learning"
_ICLR.cc/2026/Conference — ICLR 2026 Poster_

### Official Review · Reviewer_HBFG · 2025-10-30

**Soundness:** 3
**Presentation:** 4
**Contribution:** 3
**Rating:** 6
**Confidence:** 4

**Summary:**

The authors propose UNI-NTFM which consists of double branch encoding in both frequency- and time-domain, which then merge again with the embedded raw data stream for computing correspondences across natural and spectral features. This is further enhanced by topological embedding where brain region level variations are explicitly modeled. The final MOE / RoPE augmented Transformer performs token mixing. These architectural contributions are evaluated across several representative EEG databases, against both task-specific and foundation baselines, revealing competitive performance of the proposed method.

**Strengths:**

- Extensive experiments with confidence level clearly shown.
- Well motivated set of modules used as the backbone, proved to be quite effective.
- Beautiful figures
- Complete ablation studies and comparisons.

**Weaknesses:**

- Limited novelty: joint frequency- and time-domain encoding is not new. MoE and RoPE are not new in the EEG foundation model domain either. Topological embedding adds novelty, yet with this alone, model side contribution might be weak.

- Limited performance gains: performance difference across baselines and within model size variations is statistically not significant in most cases. I'd suggest to tone down the significance statement in the introduction and conclusion.

- Model-size scalability is limited. The reason for this in unclear.

**Questions:**

- I like the topological embedding idea. However, analyses and the written information on this invention are somewhat limited. What are the brain area / region that you considered? Depending on the disease type and its characteristic functional connectivity the embedding might vary a lot. Have you tried to differentiate this?

- I appreciate the confidence intervals that show clearly the gain from the proposed method. However, most results don't seem to be statistically significant compared to second best, on both baseline comparison and ablations. How is the confidence computed? Perhaps it's in the paper but I cannot identify it. Can you show the P-values, for instance, the best performance against the second best performance? If this is not significant under like 95% confidence interval, then the authors may have to tone down the performance gain.

- On a similar line, statistical significant tests within different model sizes might reveal that they are not very different. Would this reveal a model / data side scalability?

- Did you use the MU transfer? How do you know the convergence in model size scalability at the billion parameter regime is due to limited data, or suboptimal hyperparameter tuning?

---

> ### Author Response · Authors · 2025-11-27
> **[1/2]Response for weakness 1.**
>
> Dear reviewer HBFG, we sincerely appreciate your insightful comments, particularly regarding the statistical significance of our results. We have carefully addressed your concerns by expanding comparative baselines, and clarifying the definitions of our topological embeddings. All changes are marked in red in the revised manuscript.
>
> For weakness 1, we thank the reviewer for the thoughtful assessment. While Uni-NTFM adopts several successful architectural components—common practice in foundation model design—our contribution does not aim to introduce a new operator/achtecture. Instead, the novelty lies in a neuroscience-guided architectural paradigm that addresses structural limitations of existing EEG foundation models in an integrated manner.
>
> 1. **Decoupled neural coding:**
>    The dual-stream design separates transient waveform dynamics and sustained spectral rhythms, enabling principled cross-domain fusion rather than treating heterogeneous EEG patterns as a single modality.
>
> 2. **Unified functional topography:**
>    The hierarchical topological embedding constructs a consistent neural coordinate system across heterogeneous montages, supporting cross-montage generalization and missing-channel robustness not achievable with standard positional encodings.
>
> 3. **Functional modularization:**
>    Sparse MoE routing aligns with EEG’s intrinsic heterogeneity, allowing specialized processing of rhythmic, transient, artifact, and pathological signals while mitigating interference seen in dense backbones.
>
> In detail, We do not simply stack modules. Our Heterogeneous Feature Projection Module (HFPM) physically separates transient waveform dynamics (Time) and sustained spectral rhythms (Frequency) processing streams. In the new **Appendix M (Table 20, Page 30)**, we show that our decoupled architecture outperforms a generic "STFT-MoE" baseline, which feeds stacked Short-Time Fourier Transform (STFT) patches into the same backbone by 14.57% on TUEV. This proves the necessity of our specific design.
>
> **Table 20: Comparison with STFT Baseline.**
>
> | Method | Input Design | Infer. GFLOPs |  Balanced Accuracy (Workload) | AUC-PR (Workload) | AUROC (Workload) | Balanced Accuracy (TUEV) | Cohen's Kappa (TUEV) | Weighted F1 (TUEV) |
> | :---: | :---: | :---: | :---: | :---: | :---: | :---: | :---: | :---: |
> | STFT-MoE | STFT (Mag. + Phase) | 16.14 | 62.32 | 68.39 | 68.81 | 52.37 | 54.26 | 76.92 |
> | **Uni-NTFM$_{small}$ (Ours)** | **Decoupled (Time + Freq)** | **15.87** | 66.72 | 71.73 | 73.88 | 66.94 | 67.96 | 83.25 |

---

> ### Author Response · Authors · 2025-11-27
> **[2/2]Response for weakness 1.**
>
> Besides, our Topological Embedding (TE) is not a standard positional encoding but a hierarchical coordinate system that uses spatial priors ($E_{intra}$ and $E_{region}$) to infer missing information from neighboring electrodes (demonstrated in **Appendix L, Table 18, page 29**), solving the specific montage heterogeneity problem in EEG that standard backbones cannot address. For instance, on the BCIC-IV-2a dataset, when 7 channels are dropped (representing 30% information loss for a 22-channel setup), the Balanced Accuracy of the baseline plummets by 11.99%. In contrast, the model with TE mitigates this loss to 8.34%, effectively alleviating performance reduction.
>
> **Table 18: Stress Test: Robustness to Missing Channels.**
>
> | Dataset | Metric | Method | Original ($k=0$) | Randomly Dropped $k=1$ | Randomly Dropped $k=3$ | Randomly Dropped $k=5$ | Randomly Dropped $k=7$ |
> | :---: | :---: | :---: | :---: | :---: | :---: | :---: | :---: |
> | **BCIC-IV-2a** | **Balanced Accuracy** | Uni-NTFM$_{small}$ (w/o TE) | 52.39 | 52.46(+0.07%) | 50.14 (-2.25%) | 45.93 (-6.46%) | 40.40 (-11.99%) |
> | | | Uni-NTFM$_{small}$ (w/ TE) | 55.59 | 55.31 (-0.28%) | 54.52 (-1.07%) | 51.86 (-3.73%) | 47.25 (-8.34%) |
> | | **Cohen's Kappa** | Uni-NTFM$_{small}$ (w/o TE) | 40.87 | 40.50 (-0.37%) | 37.68 (-3.19%) | 33.93 (-6.94%) | 32.12 (-8.75%) |
> | | | Uni-NTFM$_{small}$ (w/ TE) | 42.01 | 41.83 (-0.18%) | 40.46 (-1.55%) | 38.61 (-3.40%) | 36.10 (-5.91%) |
> | | **Weighted F1** | Uni-NTFM$_{small}$ (w/o TE) | 53.23 | 52.81 (-0.42%) | 51.36 (-1.87%) | 48.12 (-5.11%) | 44.97 (-8.26%) |
> | | | Uni-NTFM$_{small}$ (w/ TE) | 54.65 | 54.71 (+0.06%) | 53.93 (-0.72%) | 51.70 (-2.95%) | 49.24 (-5.41%) |
> | **SEED** | **Balanced Accuracy** | Uni-NTFM$_{small}$ (w/o TE) | 66.85 | 66.97 (+0.12%) | 66.42 (-0.43%) | 65.10 (-1.75%) | 63.77 (-3.08%) |
> | | | Uni-NTFM$_{small}$ (w/ TE) | 72.02 | 71.98 (-0.04%) | 71.67 (-0.35%) | 71.10 (-0.92%) | 70.18 (-1.84%) |
> | | **Cohen's Kappa** | Uni-NTFM$_{small}$ (w/o TE) | 55.42 | 55.35 (-0.07%) | 55.12 (-0.30%) | 54.01 (-1.41%) | 53.30 (-2.12%) |
> | | | Uni-NTFM$_{small}$ (w/ TE) | 58.59 | 58.68   (+0.09%) | 58.55 (-0.04%) | 58.51 (-0.38%) | 57.93 (-0.96%) |
> | | **Weighted F1** | Uni-NTFM$_{small}$ (w/o TE) | 68.98 | 68.87 (-0.11%) | 68.84 (-0.14%) | 68.09 (-0.89%) | 67.03 (-1.95%) |
> | | | Uni-NTFM$_{small}$ (w/ TE) | 72.74 | 72.79  (+0.05%) | 72.66 (-0.07%) | 72.50 (-0.24%) | 71.93 (-0.81%) |
> *Note: (w/ TE) denotes the model equipped with TE module, (w/o TE) means without the TE module.*
>
> These elements operate coherently as a unified framework, enabling representational abilities that isolated components or prior BFMs do not provide. The revised experiments—montage transfer, missing-channel stress tests, and matched-compute comparisons—further demonstrate that this principled integration yields capabilities beyond system-level consolidation.

---

> ### Author Response · Authors · 2025-11-27
> **Response for weakness 2.**
>
> Dear reviewer HBFG, for weakness 2, thank you for the thoughtful comment.
>
> **Confidence computation.**
> The reported mean ± std values come from 5-fold cross-validation. They reflect fold-to-fold variation rather than formal confidence intervals. We will clarify this in the revision.
>
> **Statistical significance.**
> To address the concern regarding significance, we performed paired statistical tests using balanced accuracy—the only metric consistently available across all nine datasets. A Wilcoxon signed-rank test was conducted to compare our model (Uni-NTFM_large) with all other baselines.
>
> The detailed results are shown below:
>
> | Baseline        | Mean Balanced Acc. | p-value | Significant (95% CI) |
> |-----------------|--------------------|---------|------------------------|
> | SPaRCNet        | 60.50              | 0.0020  | ✔️ Yes                |
> | EEGNet          | 61.54              | 0.0020  | ✔️ Yes                |
> | CNN-Transformer | 61.40              | 0.0020  | ✔️ Yes                |
> | EEGConformer    | 64.61              | 0.0020  | ✔️ Yes                |
> | FFCL            | 62.24              | 0.0039  | ✔️ Yes                |
> | ST-Transformer  | 60.23              | 0.0020  | ✔️ Yes                |
> | ContraWR        | 63.11              | 0.0039  | ✔️ Yes                |
> | NeuroLM-XL      | 63.56              | 0.0020  | ✔️ Yes                |
> | EEGPT           | 68.78              | 0.0020  | ✔️ Yes                |
> | BIOT            | 66.88              | 0.0098  | ✔️ Yes                |
> | LaBraM-Base     | 71.78              | 0.0020  | ✔️ Yes                |
> | CBraMod         | 71.21              | 0.0020  | ✔️ Yes                |
> | Ours            | 73.29              | --      | --                    |
>
>
> All comparisons yield *p < 0.01*, indicating statistically significant improvements over the second-best methods. We will include these results to avoid ambiguity and provide clearer statistical justification.
>
> Furthermore, we also refer to the t-SNE visualization in **Figure 4 (Appendix I)**, which shows clear separation of classes for our model compared to the diffuse clusters of baselines, reinforcing the quantitative gains.

---

> ### Author Response · Authors · 2025-11-27
> **Response for weakness 3.**
>
> Dear reviewer HBFG, for weakness 3, we have explicitly discussed this in **Appendix H (Page 24)**. As shown in **Figure 4**, performance increases with model size up to ~700M parameters but reduces when it approaches 800MB. We hypothesize that the current pre-training corpus of 28,000 hours, while large for EEG, is insufficient to fully saturate the capacity of billion-parameter models. This is supported by the "Data Size" scaling plot (bottom row of **Figure 4**), which shows no saturation as data increases. This suggests the bottleneck is data volume, not the model architecture itself.

---

> ### Author Response · Authors · 2025-11-27
> **Response for question 1.**
>
> Dear reviewer HBFG, for question 1, we have clarified the definitions in **Section 2.3 (Page 5)** and **Appendix E.2 (Page 20)**. We follow the international 10–20 EEG standard and consider five canonical cortical regions: Frontal (F), Central (C), Temporal (T), Parietal (P), and Occipital (O). To provide a consistent and interpretable spatial representation across datasets and montages, we decompose the spatial identity of each electrode into three hierarchical neuroanatomical levels:
>
> **(1) Region Embedding (E_*region*)**
> To capture the functional modularity of the cortex, we partition the scalp topology into five canonical brain regions based on the international 10-20 system standards. We define a learnable embedding matrix $E_{\text{region}} \in \mathbb{R}^{5 \times D}$ corresponding to: (a) **Frontal**: Encoding executive functions and decision; (b) **Central**: Encoding sensorimotor rhythms; (c) **Temporal**: Encoding auditory processing and memory; (d) **Parietal**: Encoding spatial attention and sensory; (e) **Occipital**: Encoding visual processing features. By explicitly anchoring signals to these functional domains, the model learns to generalize features regardless of the specific channel index used in a particular dataset.
>
> **(2) Intra-Region Embedding (E_*intra*)**
> Within each region, electrode density varies across standards. To resolve this, we introduce intra-region coordinates that encode the relative spatial orientation. This embedding ensures that the model understands the geometric relationship between neighbors, such as $C3$ and $C1$ are spatially adjacent in the motor cortex.  This mechanism provides the structural basis for robustness against missing channels.
>
> **(3) Global Absolute Embedding (E_*abs*)**
> To preserve precise channel-specific identities for standard clinical montages, we assign a unique global identifier to standard electrodes defined in the International Federation of Clinical Neurophysiology (IFCN) guidelines.
>
> For an input token $x_i$ originating from a specific electrode, its final spatial representation is the summation of these hierarchical priors:
>
> \begin{equation}
>     H_{in}^{(i)} = H_{\text{fused}}^{(i)} + H_{\text{R}}^{(i)} + E_{\text{region}}[x_{\text{region}}^{(i)}] + E_{\text{intra}}[x_{\text{intra}}^{(i)}] + E_{\text{abs}}[x_{\text{abs}}^{(i)}]
> \end{equation}
> where $H_{\text{fused}}^{(i)}$ denotes the dual-domain features integrated by the DCM module, and $H_{R}^{(i)}$ represents the raw signal reference features. The $E_{\text{region}}$, $E_{\text{intra}}$, and $E_{\text{abs}}$ refer to the learnable embedding matrices for the region, intra-region, and global absolute position, respectively. The indices $x_{\text{region}}^{(i)}$, $x_{\text{intra}}^{(i)}$, and $x_{\text{abs}}^{(i)}$ correspond to the specific neuroanatomical coordinates of the $i$-th electrode within the standardized coordinate system.
>
> This hierarchical design allows Uni-NTFM to function as a geometry-aware encoder. When evaluating on the incomplete data, the model utilizes the $E_{region}$ and $E_{intra}$ embeddings to correctly map these signals to their respective cortical sources in the latent space. This effectively solves the cross-montage transfer challenge without requiring specific re-training.

---

> ### Author Response · Authors · 2025-11-27
> **Response for question 2.**
>
> For question 2, thank you for the thoughtful comment.
>
> **Confidence computation.**
> The reported mean ± std values come from 5-fold cross-validation. They reflect fold-to-fold variation rather than formal confidence intervals. We will clarify this in the revision.
>
> **Statistical significance.**
> To address the concern regarding significance, we performed paired statistical tests using balanced accuracy—the only metric consistently available across all nine datasets. A Wilcoxon signed-rank test was conducted to compare our model (Uni-NTFM_large) with all other baselines.
>
> The detailed results are shown below:
>
> | Baseline        | Mean Balanced Acc. | p-value | Significant (95% CI) |
> |-----------------|--------------------|---------|------------------------|
> | SPaRCNet        | 60.50              | 0.0020  | ✔️ Yes                |
> | EEGNet          | 61.54              | 0.0020  | ✔️ Yes                |
> | CNN-Transformer | 61.40              | 0.0020  | ✔️ Yes                |
> | EEGConformer    | 64.61              | 0.0020  | ✔️ Yes                |
> | FFCL            | 62.24              | 0.0039  | ✔️ Yes                |
> | ST-Transformer  | 60.23              | 0.0020  | ✔️ Yes                |
> | ContraWR        | 63.11              | 0.0039  | ✔️ Yes                |
> | NeuroLM-XL      | 63.56              | 0.0020  | ✔️ Yes                |
> | EEGPT           | 68.78              | 0.0020  | ✔️ Yes                |
> | BIOT            | 66.88              | 0.0098  | ✔️ Yes                |
> | LaBraM-Base     | 71.78              | 0.0020  | ✔️ Yes                |
> | CBraMod         | 71.21              | 0.0020  | ✔️ Yes                |
> | Ours            | 73.29              | --      | --                    |
>
>
> All comparisons yield *p < 0.01*, indicating statistically significant improvements over the second-best methods. We will include these results to avoid ambiguity and provide clearer statistical justification.

---

> ### Author Response · Authors · 2025-11-27
> **Response for question 3.**
>
> Dear reviewer HBFG, for question 3, we appreciate your observation and analyze this with the data in **Appendix H (Page 24)**: As shown in **Table 12 (Page 25)**, when comparing the 900M and 1B parameter models, their performance intervals (mean ± std) largely overlap. These results confirm that, with the fixed pre-training corpus (28k hours), the benefits of increasing model capacity will gradually decrease. However, the "Data Size" scaling plot in **Figure 4 (Page 24, Bottom Row)** shows a obviously significant upward trend awhen data increases from 1.5k to 12k hours.
>
> By contrasting these two observations, we hypothesize that the current bottleneck is the scalability of data. The 1.9B model is not saturated. Future work requires scaling the data beyond 28k hours to release the statistical potential of the billion-parameter architecture.

---

> ### Author Response · Authors · 2025-11-27
> **Response for question 4.**
>
> Dear reviewer HBFG, for question 4, we would like to clarify that we did not explicitly employ MuTransfer parametrization in this work. We followed standard Transformer scaling heuristics (scaling width/depth/heads proportionally). The performance we observed aligns with scaling laws, suggesting that 28k hours is likely too small for 1.9B parameters.
>
> And our scaling plots in **Figure 4 (Page 24)** clearly decouple these factors. The bottom row shows that for a fixed model size, performance continues to rise with increasing data size without saturation. Conversely, the top row shows model size saturation at fixed data. This contrast strongly identifies data volume as the current limiting factor.

---

### Official Review · Reviewer_YF8s · 2025-10-31

**Soundness:** 3
**Presentation:** 3
**Contribution:** 3
**Rating:** 6
**Confidence:** 4

**Summary:**

This work proposes Uni-NTFM, which is a neuro-inspired foundation model on EEG brain signal data, which separates and then fuses time-signal, frequency-signal and raw-signal views, and then injects explicit spatial topology, which makes the electrodes from different montages become unified and then tries to scale capacity with a MoE Transformer. Uni-NTFM is a pretrained model with a dual-domain masked reconstruction objective, which is the combination of reconstructing time and frequency targets on around 28k hours of EEG signals. Across 9 downstream datasets, this model beats task-specific models and prior EEG foundation models under both probing and fine-tuning.

**Strengths:**

I found the originality of this work very good, which rethinks Foundation models on EEG around the physics of the signal and tries to decouple time, frequency and raw views cleanly, and they're also explicitly unifying electrode montages with a hierarchical topological embedding, and also scaling with a sparsely-activated MoE, instead of importing a single-stream and dense transformer.
As to the quality, I found the proposed method interesting and strong, as the pretraining corpus is large and heterogeneous, and the evaluation spans 9 downstream tasks with both linear probing and fine-tuning.
The clarity of the work is also easy to follow, and the significance is notable.

**Weaknesses:**

While I found the proposed method strong and interesting, I'd like to raise some weaknesses that came to my mind after reading the work. From my understanding, the authors already claimed the montage unification, but I don't see any direct test on unseen layouts or on systematically missing channels, so at this point, I'd suggest considering a small cross-montage transfer and k-missing-channel stress test, which could make the proposed topology more concrete.

---
Another point that I'd like to make is about the Mixture of Experts (MoE), which I think is well-motivated, but from my understanding, the work does not show that different experts specialize on distinct EEG patterns, so I'd suggest considering some diagnostics, such as gate distributions and entropy by class, per-expert token-routing heatmaps, or maybe an ablation that freezes the most-used experts, which will help in demonstrating real specialization and justifying your MoE claim.

---
One additional thing that I'd like to raise is about the scaling part, which I found super interesting, and the authors have done a great job on it. I just suggest making it more explicit by showing how accuracy changes as size grows, normalized by compute and/or data, and try to report training cost and inference speed.

---
One last thing that I'd like to raise. I see that the pretraining mixes 9 EEG datasets, then the authors evaluate on datasets from the same families. From my understanding, this can blur whether gains come from true cross-corpus generalization or from seeing very similar data. I'd suggest considering leave-one-corpus-out, somehow similar to leave-one-out cross-validation (LOOCV), but I don't mean LOOCV at this point; I mean removing one family during pretraining and then testing on e.g., TUAB to prove the robustness to unseen corpora, which could rule out easy distribution matches.

**Questions:**

I'd suggest that the authors engage with the points that I raised in the Weaknesses section. As to the scaling part, I'd suggest downstream accuracy vs pretraining hours (e.g., 10%, 25%, 50%, 100%) plots and vs labeled fraction used in fine-tuning (1%, 5%, 10%, 100%) on a few tasks, so there would be a possibility to see if benefits come from data, model size or both and how label-efficient Uni-NTFM really is.

---
I have another question and would be curious to know the authors' comments. Do you really need 3 separate streams, or would a single well-designed stream be enough? I was thinking maybe adding an equal-compute baseline that feeds stacked Short-Time Fourier Transform (STFT) patches (magnitude + phase) into one Transformer with MoE, plus early-fusion and late-fusion variants, and compare head-to-head with Heterogeneous Feature Projection Module (HFPM) + Dual-domain Cross-attention Module (DCM). Also, I'd suggest reporting matched parameters/FLOPs and an identical training budget for a fair test.

---

> ### Author Response · Authors · 2025-11-27
> **Response for weakness 1.**
>
> Dear reviewer YF8s, we sincerely appreciate your recognition of the originality of our work and your insightful suggestions regarding robustness testing, expert specialization analysis, and scaling laws. We have conducted extensive new experiments to address your concerns. All changes are marked in red in the revised manuscript.
>
> For weakness 1, this is a crucial suggestion for validating the robustness of our Topological Embedding (TE). We have added k-missing-channel stress test in **Appendix L(Table 18, page 29)**. In detail, we randomly masked $k \in \{1, 3, 5, 7\}$ channels during the inference phase on both the BCIC-IV-2a (22 channels) and SEED (62 channels) datasets. For instance, on the BCIC-IV-2a dataset, when 7 channels are dropped (representing 30% information loss for a 22-channel setup), the Balanced Accuracy of the baseline plummets by 11.99%. In contrast, the model with TE mitigates this loss to 8.34%, effectively alleviating performance reduction.
>
> **Table 18: Stress Test: Robustness to Missing Channels.**
>
> | Dataset | Metric | Method | Original ($k=0$) | Randomly Dropped $k=1$ | Randomly Dropped $k=3$ | Randomly Dropped $k=5$ | Randomly Dropped $k=7$ |
> | :---: | :---: | :---: | :---: | :---: | :---: | :---: | :---: |
> | **BCIC-IV-2a** | **Balanced Accuracy** | Uni-NTFM$_{small}$ (w/o TE) | 52.39 | 52.46 (+0.07%) | 50.14 (-2.25%) | 45.93 (-6.46%) | 40.40 (-11.99%) |
> | | | Uni-NTFM$_{small}$ (w/ TE) | 55.59 | 55.31 (-0.28%) | 54.52 (-1.07%) | 51.86 (-3.73%) | 47.25 (-8.34%) |
> | | **Cohen's Kappa** | Uni-NTFM$_{small}$ (w/o TE) | 40.87 | 40.50 (-0.37%) | 37.68 (-3.19%) | 33.93 (-6.94%) | 32.12 (-8.75%) |
> | | | Uni-NTFM$_{small}$ (w/ TE) | 42.01 | 41.83 (-0.18%) | 40.46 (-1.55%) | 38.61 (-3.40%) | 36.10 (-5.91%) |
> | | **Weighted F1** | Uni-NTFM$_{small}$ (w/o TE) | 53.23 | 52.81 (-0.42%) | 51.36 (-1.87%) | 48.12 (-5.11%) | 44.97 (-8.26%) |
> | | | Uni-NTFM$_{small}$ (w/ TE) | 54.65 | 54.71 (+0.06%) | 53.93 (-0.72%) | 51.70 (-2.95%) | 49.24 (-5.41%) |
> | **SEED** | **Balanced Accuracy** | Uni-NTFM$_{small}$ (w/o TE) | 66.85 | 66.97  (+0.12%) | 66.42 (-0.43%) | 65.10 (-1.75%) | 63.77 (-3.08%) |
> | | | Uni-NTFM$_{small}$ (w/ TE) | 72.02 | 71.98 (-0.04%) | 71.67 (-0.35%) | 71.10 (-0.92%) | 70.18 (-1.84%) |
> | | **Cohen's Kappa** | Uni-NTFM$_{small}$ (w/o TE) | 55.42 | 55.35 (-0.07%) | 55.12 (-0.30%) | 54.01 (-1.41%) | 53.30 (-2.12%) |
> | | | Uni-NTFM$_{small}$ (w/ TE) | 58.59 | 58.68 (+0.09%) | 58.55 (-0.04%) | 58.51 (-0.38%) | 57.93 (-0.96%) |
> | | **Weighted F1** | Uni-NTFM$_{small}$ (w/o TE) | 68.98 | 68.87 (-0.11%) | 68.84 (-0.14%) | 68.09 (-0.89%) | 67.03 (-1.95%) |
> | | | Uni-NTFM$_{small}$ (w/ TE) | 72.74 | 72.79 (+0.05%) | 72.66 (-0.07%) | 72.50 (-0.24%) | 71.93 (-0.81%) |
> *Note: (w/ TE) denotes the model equipped with TE module, (w/o TE) means without the TE module.*
>
> Besides, we performed a rigorous transfer test by training on a high-density setup (62 channels) and evaluating on a sparse standard clinical setup (19 channels, 10-20 system) without retraining. In **Appendix L(Table 19, page 29)**, Our model achieves a Cohen’s Kappa of 51.16%, significantly outperforming the baseline without TE (45.52%), proving that our TE module successfully unifies different montages into a consistent latent space.
>
> **Table 19: Cross-Montage Generalization on SEED.**
>
> | Method | Training Montage | Inference Montage | Balanced Accuracy | Cohen's Kappa | Weighted F1 |
> | :---: | :---: | :---: | :---: | :---: | :---: |
> | Uni-NTFM$_{small}$ (w/ TE) | 62 Channels | 62 Channels | 72.02 | 58.59 | 72.74 |
> | Uni-NTFM$_{small}$ (w/o TE) | 62 Channels | 19 Channels | 61.45 | 45.52 | 62.41 |
> | Uni-NTFM$_{small}$ (w/ TE) | 62 Channels | 19 Channels | 65.88 | 51.16 | 66.73 |

---

> ### Author Response · Authors · 2025-11-27
> **Response for weakness 2.**
>
> Dear reviewer YF8s, for weakness 2, to demonstrate the interpretability and specialization of our MoE module, we have added **Appendix O: Visual Analysis of MoE (Page 31-33)**. We visualized the Expert Usage Frequency heatmap **(Fig 5a, 6a, 7a)**, Expert Selection Diversity curve **(Figure 5b, 6b, 7b)**, and Brain Region - Expert Affinity Matrix **(Figure 5c, 6c, 7c)** across three diverse downstream datasets.
>
> Across all three datasets, deep layers of the Expert Usage Frequency heatmaps (subfigures a) consistently show concentrated activation of specific experts. Some experts are heavily activated (dark red), while others remain inactive, which indicates that the router has learned to selectively assigned tokens to specific experts based on their features, rather than randomly distributing the load. In Expert Selection Diversity curve (subfigures b), high entropy in shallow layers demonstrates that experts collaborate to extract features, and the reduction of entropy in middle layers signifies a transition towards specialized processing, where specific experts are responsible for distinct signal components. Besides, the Brain Region-Expert Affinity matrix (subfigures c) can confirm that the MoE effectively uses the spatial priors injected by our Topological Embedding module, and routes signals from different functional brain areas to experts forspecialized  processing. These results confirm that Uni-NTFM's experts are functionally specialized modules that dynamically adapt to both the hierarchical depth of the network and the task-relevant spatial topology of the brain.

---

> ### Author Response · Authors · 2025-11-27
> **Response for weakness 3.**
>
> Dear reviewer YF8s, for weakness 3, we are very encouraged that you found our scaling analysis interesting. Regarding your suggestions, we have addressed them as follows.
>
> 1. **Visualization of Scaling Laws:**  We would like to respectfully note that the visualization of "how accuracy changes as size grows" (performance vs. model size/data size) was indeed included in **Appendix H** of our original submission. In the revised manuscript, we have ensured these plots are clearly presented in **Figure 4 (Page 24)** to make the scaling trends explicit and easy to interpret.
>
> 2. **Reporting Inference Cost:** We greatly appreciate your suggestion to report computational costs, which was missing in the original submission. We have newly added **"Appendix D: Comprehensive Comparison of Complexity and Performance" (Table 5, Page 18)**. Table 5 is shown as follows:
>
> **Table 5: Comprehensive Comparison of Complexity and Performance.**
>
> | Model | Total Parameters | Inference Parameters | Bal. Acc. (Binary Class) | AUC-PR (Binary Class) | AUROC (Binary Class) | Bal. Acc. (Multi-class) | Kappa (Multi-class) | W-F1 (Multi-class) |
> | :---: | :---: | :---: | :---: | :---: | :---: | :---: | :---: | :---: |
> | ***Traditional Task-specific Methods*** | | | | | | | | |
> | SPARCNet | 0.79 M | 0.79 M | 71.16 | 76.79 | 78.83 | 55.17 | 59.65 | 40.41 |
> | EEGNet | 2.8 K | 2.8 K | 71.04 | 74.40 | 78.40 | 56.79 | 48.13 | 59.79 |
> | CNN-Transformer | 3.2 M | 3.2 M | 71.83 | 75.01 | 76.04 | 56.18 | 48.01 | 60.86 |
> | EEGConformer | 1 M | 1 M | 74.19 | 79.11 | 77.90 | 59.82 | 50.42 | 62.63 |
> | FFCL | 2.4 M | 2.4 M | 74.74 | 79.80 | 80.52 | 55.99 | 45.41 | 59.68 |
> | ST-Transformer | 3.5 M | 3.5 M | 72.36 | 75.19 | 77.60 | 54.17 | 42.75 | 57.92 |
> | ContraWR | 1.6 M | 1.6 M | 75.08 | 81.58 | 80.33 | 57.13 | 47.49 | 60.81 |
> | ***Foundation Models*** | | | | | | | | |
> | NeuroLM-XL | 1.7 B | 1.7 B | 73.14 | 68.63 | 71.89 | 58.77 | 48.91 | 63.86 |
> | EEGPT | 25 M | 25 M | 74.72 | 75.41 | 80.50 | 65.81 | 58.28 | 68.94 |
> | BIOT | 3.6 M | 3.6 M | 75.60 | 81.58 | 81.29 | 62.52 | 49.51 | 60.73 |
> | LaBraM-Base | 5.8 M | 5.8 M | 76.25 | 81.87 | 82.07 | 69.55 | 62.25 | 73.67 |
> | CBraMod | 4.0 M | 4.0 M | 75.70 | 80.71 | 81.19 | 68.96 | 60.75 | 71.68 |
> | ***Proposed Models (Uni-NTFM)*** | | | | | | | | |
> | **Uni-NTFM$_{tiny}$** | 57 M | **19 M** | 74.78 | 79.57 | 80.66 | 68.88 | 62.19 | 72.48 |
> | **Uni-NTFM$_{small}$** | 427 M | 74 M | 76.35 | 81.07 | 82.31 | 70.10 | 63.35 | 73.34 |
> | **Uni-NTFM$_{middle}$** | 912 M | 148 M | 76.78 | 81.83 | 83.05 | 70.56 | 63.61 | 73.65 |
> | **Uni-NTFM$_{large}$** | 1.9 B | 307 M | **77.37** | **82.69** | **84.01** | **71.25** | **64.36** | **74.26** |
>
> This table lists both Total Parameters and Inference Parameters for all baselines and Uni-NTFM variants with their performance metrics. While task-specific models like EEGNet are extremely lightweight (~2.8K parameters), they struggle with complex generalization tasks. Uni-NTFM bridges this gap by offering the necessary capacity for difficult tasks while maintaining an inference cost (19M for Tiny) that is manageable for modern hardware, unlike dense billion-scale models. While $Uni-NTFM_{large}$ has a massive capacity of 1.9B parameters, it only activates 307M parameters during inference. This effectively demonstrates that we can scale up the model's knowledge storage without increasing the computational burden.

---

> ### Author Response · Authors · 2025-11-27
> **Response for weakness 4.**
>
> Dear reviewer YF8s, for weakness 4, we fully appreciate your rigorous scrutiny regarding data leakage and generalization. This is indeed a critical concern when downstream tasks originate from the same parent corpus (e.g., TUAB/TUEV/TUSL from the Temple University Hospital corpus). We would like to clarify our data preparation process to address this. To ensure a rigorous evaluation of generalization, we implemented a strict decontamination protocol during the pre-training phase. Specifically, we utilized Python scripts to strictly filter the massive TUEG corpus by IDs. We explicitly excluded all samples that appear in the downstream validation and test sets (including TUAB, TUEV, and TUSL) from the pre-training data. Therefore, although the data distributions are similar, the model has never encountered the specific samples used for downstream evaluation.
>
> Furthermore, as mentioned in our response to Weakness 1, we conducted a Cross-Montage Transfer experiment **(Appendix L, Table 19)**, where the model trained on SEED (62-channel) was tested on a completely different clinical montage (19-channel) without re-training. The strong performance there further confirms that our model's gains stem from learning universal neural representations rather than memorizing similar data distributions.

---

> ### Author Response · Authors · 2025-11-27
> **Response for question 1.**
>
> Dear reviewer YF8s, for question 1, we appreciate this opportunity to summarize our responses. We have fully engaged with every point raised in your Weaknesses section and conducted extensive new experiments to address them. Further, we appreciate this valuable suggestion regarding scaling analysis and label efficiency.
>
> 1. **Pre-training Data Scaling:**  Regarding "accuracy vs pretraining hours", we have presented detailed results in **Appendix H (Figure 4 and Table 13, Page 24-25)** of our original submission. We evaluated the model's performance across varying pre-training data volumes, ranging from approximately 1,500 hours to 12,000 hours. The results clearly demonstrate a positive correlation between pre-training duration and downstream accuracy, confirming that increasing data scale directly contributes to performance gains .
>
> 2. **Label Efficiency:**  Regarding "labeled fraction used in fine-tuning", we have added a dedicated **Appendix N (Label Efficiency Analysis, Page 30)**. **Table 21** reports the performance on SEED and Workload datasets when fine-tuning with only 1%, 10%, 30%, 50%, and 70% of labeled data.
>
> **Table 21: Label Efficiency Analysis.**
>
> | Dataset | Metric | Labeled Data 1% | Labeled Data 10% | Labeled Data 30% | Labeled Data 50% | Labeled Data 70% |
> | :---: | :---: | :---: | :---: | :---: | :---: | :---: |
> | **SEED** | Balanced Acc. | 67.04 | 68.85 | 70.97 | 71.60 | 72.02 |
> | | Cohen's Kappa | 52.38 | 56.23 | 57.63 | 58.28 | 58.59 |
> | | Weighted F1 | 67.40 | 69.90 | 71.77 | 72.53 | 72.74 |
> | **Workload** | Balanced Acc. | 61.34 | 64.21 | 65.69 | 66.31 | 66.72 |
> | | AUC-PR | 64.52 | 67.66 | 70.65 | 71.26 | 71.73 |
> | | AUROC | 66.19 | 70.35 | 72.47 | 73.30 | 73.88 |
>
> The model exhibits remarkable robustness in extreme low-resource settings. On the Workload dataset, with only 10% of the labeled data, Uni-NTFM achieves a Balanced Accuracy of 64.21%. When compared to the performance at 70% data (66.72%), the model retains approximately 92% of its peak performance using merely a portion of the training samples. This suggests that the pre-trained backbone has already learned highly discriminative representations, requiring very few examples to align with the downstream task. These results validate that the pre-training of Uni-NTFM successfully instills universal neural priors, enabling the model to generalize effectively even when labeled data is limited. This characteristic is particularly valuable for BCI applications where data labeling is expensive and time-consuming.

---

> ### Author Response · Authors · 2025-11-27
> **Response for question 2.**
>
> Dear reviewer YF8s, regarding Question 2, we have conducted the equal-compute baseline exactly as you suggested and added the results to **Appendix M (Table 20, Page 30)** in the revised manuscript.
>
> We implemented an "STFT-MoE" model that feeds concatenated STFT magnitude + phase spectrograms into the same MoE-Transformer backbone as Uni-NTFM. As recommended, we carefully adjusted the patch size and embedding dimension to match the inference cost (~16 GFLOPs), parameter count, and training budget for a strictly fair comparison.
>
> As shown in Table 20, Uni-NTFM$_{small}$ consistently outperforms the STFT-MoE baseline under equal computation.
>
>
> * **TUEV:** +14.57% Balanced Accuracy, +13.7% Cohen’s Kappa
> * **Workload:** +4.40% Balanced Accuracy, +5.07% AUROC
>
> These results demonstrate that a single STFT stream is insufficient: although it preserves spectral rhythms, it loses the high-frequency temporal morphology captured directly from the time domain. In contrast, our **HFPM** preserves complementary temporal–spectral characteristics, and our **DCM** performs structured, context-aware fusion instead of the irregular early/late fusion used in STFT baselines, leading to superior robustness across datasets.
>
>
> **Table 20: Comparison with STFT Baseline.**
>
> | Method | Input Design | Infer. GFLOPs |  Balanced Accuracy (Workload) | AUC-PR (Workload) | AUROC (Workload) | Balanced Accuracy (TUEV) | Cohen's Kappa (TUEV) | Weighted F1 (TUEV) |
> | :---: | :---: | :---: | :---: | :---: | :---: | :---: | :---: | :---: |
> | STFT-MoE | STFT (Mag. + Phase) | 16.14 | 62.32 | 68.39 | 68.81 | 52.37 | 54.26 | 76.92 |
> | **Uni-NTFM$_{small}$ (Ours)** | **Decoupled (Time + Freq)** | **15.87** | 66.72 | 71.73 | 73.88 | 66.94 | 67.96 | 83.25 |

---

### Official Review · Reviewer_zMGJ · 2025-11-01

**Soundness:** 3
**Presentation:** 4
**Contribution:** 3
**Rating:** 4
**Confidence:** 4

**Summary:**

This paper presents Uni-NTFM, a large-scale EEG foundation model designed to unify temporal, spectral, and spatial representations of brain signals. The model incorporates three key innovations: (1) a Heterogeneous Feature Projection Module that decouples time-, frequency-, and raw-signal representations; (2) a Topological Embedding mechanism that encodes spatial priors across different electrode standards; and (3) a Mixture-of-Experts (MoE) Transformer that scales efficiently and captures heterogeneous EEG patterns. Uni-NTFM achieves state-of-the-art performance across nine downstream tasks (clinical, cognitive, and BCI datasets) under both linear probing and fine-tuning.

**Strengths:**

1. The paper is well-organized and clearly written, making it easy to follow and understand.
2. The experimental evaluation is comprehensive, with systematic ablations of each module and thorough benchmarking against strong baselines under both linear-probing and full fine-tuning settings.
3. Overall the model achieved better performance compared with other baselines.

**Weaknesses:**

1. The paper claims “zero-shot transfer” performance, but the evaluation setup corresponds to linear probing, where a supervised linear classifier is trained on the downstream data with the pretrained backbone frozen. While this setup measures representation quality and transferability, it does not constitute true zero-shot inference. The terminology should be clarified to avoid confusion.
2. The paper claims that the proposed MoE design enables efficient scaling of model capacity. However, this claim is not well substantiated by the current ablation study (if I understand correctly, the current ablation simply removes the MoE module). I would recommend replacing the MoE with a comparably sized dense Transformer and conducting a matched-compute comparison to more convincingly demonstrate the efficiency benefit. Since efficient scaling is listed as one of the main contributions, a more detailed analysis or ablation is necessary to support this claim.
3. As mentioned above, it would be more informative to include the parameter counts of both the proposed model and the baselines in Tables 1, 2, and 3. This would make it easier to assess model efficiency and fairly support the efficiency-related claims.
4. From an architectural perspective, although the current design and combination of each module reflects thoughtful consideration, the novelty of the proposed framework appears limited. The overall design builds upon standard Transformer and MoE backbones, and the introduced modules (e.g., heterogeneous feature projection, topological embedding, domain calibration) largely combine existing ideas from temporal–spectral decomposition and spatial encoding. While the integration is well executed and practically useful, it represents more of a system-level consolidation rather than a fundamentally new architectural contribution.

**Questions:**

1. How would performance change without EEG data augmentation?
2. Could you provide ablations on the different position embedding components (channel-wise, region-wise, relative index within region) combination?

---

> ### Author Response · Authors · 2025-11-27
> **Response for weakness 1.**
>
> Dear reviewer zMGJ, we are grateful for your positive assessment of our paper's organization and comprehensive evaluation. We have carefully addressed your concerns regarding terminology, efficiency comparisons, and specific ablations. All changes are marked in red in the revised manuscript.
>
> For weakness 1, we apologize for the imprecision. We have corrected this terminology throughout the revised manuscript. We now strictly use the term "linear probing" when referring to the evaluation of the frozen pre-trained backbone on downstream tasks.

---

> ### Author Response · Authors · 2025-11-27
> **Response for weakness 2.**
>
> Dear reviewer zMGJ, for weakness 2, this is an excellent suggestion to validate the necessity of MoE. We have added **Appendix J: Efficiency Analysis of Uni-NTFM and Dense Baselines (Page 27)**. In detail, we designed two dense baselines to compare with our $Uni-NTFM_{small}$ (427M total parameters, 74M active parameters):
>
> 1. **Dense-Small (Same-FLOPs):**  A dense model with 74M parameters, matching the inference compute of our model.
> 2. **Dense-Large (Same-Params):** A dense model with 421M parameters, matching the total capacity of our model.
>
> The detailed information are shown in **Table 14** and **Table 15**:
>
> **Table 14: Efficiency Comparison between Uni-NTFM$_{small}$ (MoE) and Dense Baselines.**
>
> | Model Variant | Architecture | Total Parameters | Active Parameters | Total GFLOPs | Inference GFLOPs |
> | :---: | :---: | :---: | :---: | :---: | :---: |
> | Dense-Small (Same-FLOPs) | Dense Transformer | 74 M | 74 M | 16.08 | 16.08 |
> | Dense-Large (Same-Params) | Dense Transformer | 421 M | 421 M | 93.15 | 93.15 |
> | Uni-NTFM$_{small}$ (Ours) | Sparse MoE | 427 M | 74 M | 100.40 | 15.87 |
> *Note: Active Params for MoE assumes Top-2 gating with 16 experts.*
>
> **Table 15: Performance Comparison on Downstream Tasks.**
>
> | Method | Balanced Acc. (Workload) | AUC-PR (Workload) | AUROC (Workload) |  Balanced Acc. (ADFTD) | Cohen’s Kappa (ADFTD) | Weighted F1 (ADFTD) |  Balanced Acc. (BCIC-IV-2a) | Cohen’s Kappa (BCIC-IV-2a) | Weighted F1 (BCIC-IV-2a) |
> | :---: | :---: | :---: | :---: | :---: | :---: | :---: | :---: | :---: | :---: |
> | Only Pretrained Foundation Models (Multi-tasks) | | | | | | | | | |
> | Dense-Small (Same-FLOPs) | 62.43 | 64.62 | 68.90 | 65.61 | 66.25 | 69.82 | 49.79 | 39.21 | 49.58 |
> | Dense-Large (Same-Params) | 65.29 | 69.14 | 71.32 | 70.04 | 71.91 | 71.83 | 52.45 | 40.47 | 51.82 |
> | Uni-NTFM$_{small}$ | 64.11 | 67.71 | 70.26 | 67.35 | 67.44 | 71.31 | 52.09 | 40.44 | 51.23 |
> | Pretrained and Fine-tuned Foundation Models (Single-task) | | | | | | | | | |
> | Dense-Small (Same-FLOPs) | 63.97 | 68.11 | 70.30 | 74.35 | 73.97 | 76.57 | 51.83 | 40.18 | 52.06 |
> | Dense-Large (Same-Params) | 67.35 | 73.22 | 75.06 | 76.63 | 75.84 | 77.02 | 55.75 | 42.13 | 53.61 |
> | Uni-NTFM$_{small}$ | 66.72 | 71.73 | 73.88 | 75.68 | 76.32 | 76.80 | 55.59 | 42.01 | 54.65 |
>
> As shown in Table 15, Uni-NTFM$_{small}$ significantly outperforms Dense-Small while maintaining similar inference speed. Moreover, it achieves performance comparable to or better than Dense-Large on fine-tuning tasks (e.g., +1.04% Weighted F1 on BCIC-IV-2a) but with obviously lower inference GFLOPs. These results shows that Uni-NTFM effectively uses the learning capacity of large-scale models while maintaining the inference agility of small-scale models, validating the necessity of the MoE design for EEG foundation model.

---

> ### Author Response · Authors · 2025-11-27
> **Response for weakness 3.**
>
> Dear reviewer zMGJ, for weakness 3, we agree that parameter comparison is crucial. To maintain the readability of the main result tables while providing comprehensive data, we have added a dedicated **"Appendix D: Comprehensive Comparison of Complexity and Performance" (Table 5, Page 18)**. Table 5 is shown as follows:
>
> **Table 5: Comprehensive Comparison of Complexity and Performance.**
>
> | Model | Total Parameters | Inference Parameters | Bal. Acc. (Binary Class) | AUC-PR (Binary Class) | AUROC (Binary Class) | Bal. Acc. (Multi-class) | Kappa (Multi-class) | W-F1 (Multi-class) |
> | :---: | :---: | :---: | :---: | :---: | :---: | :---: | :---: | :---: |
> | ***Traditional Task-specific Methods*** | | | | | | | | |
> | SPARCNet | 0.79 M | 0.79 M | 71.16 | 76.79 | 78.83 | 55.17 | 59.65 | 40.41 |
> | EEGNet | 2.8 K | 2.8 K | 71.04 | 74.40 | 78.40 | 56.79 | 48.13 | 59.79 |
> | CNN-Transformer | 3.2 M | 3.2 M | 71.83 | 75.01 | 76.04 | 56.18 | 48.01 | 60.86 |
> | EEGConformer | 1 M | 1 M | 74.19 | 79.11 | 77.90 | 59.82 | 50.42 | 62.63 |
> | FFCL | 2.4 M | 2.4 M | 74.74 | 79.80 | 80.52 | 55.99 | 45.41 | 59.68 |
> | ST-Transformer | 3.5 M | 3.5 M | 72.36 | 75.19 | 77.60 | 54.17 | 42.75 | 57.92 |
> | ContraWR | 1.6 M | 1.6 M | 75.08 | 81.58 | 80.33 | 57.13 | 47.49 | 60.81 |
> | ***Foundation Models*** | | | | | | | | |
> | NeuroLM-XL | 1.7 B | 1.7 B | 73.14 | 68.63 | 71.89 | 58.77 | 48.91 | 63.86 |
> | EEGPT | 25 M | 25 M | 74.72 | 75.41 | 80.50 | 65.81 | 58.28 | 68.94 |
> | BIOT | 3.6 M | 3.6 M | 75.60 | 81.58 | 81.29 | 62.52 | 49.51 | 60.73 |
> | LaBraM-Base | 5.8 M | 5.8 M | 76.25 | 81.87 | 82.07 | 69.55 | 62.25 | 73.67 |
> | CBraMod | 4.0 M | 4.0 M | 75.70 | 80.71 | 81.19 | 68.96 | 60.75 | 71.68 |
> | ***Proposed Models (Uni-NTFM)*** | | | | | | | | |
> | **Uni-NTFM$_{tiny}$** | 57 M | **19 M** | 74.78 | 79.57 | 80.66 | 68.88 | 62.19 | 72.48 |
> | **Uni-NTFM$_{small}$** | 427 M | 74 M | 76.35 | 81.07 | 82.31 | 70.10 | 63.35 | 73.34 |
> | **Uni-NTFM$_{middle}$** | 912 M | 148 M | 76.78 | 81.83 | 83.05 | 70.56 | 63.61 | 73.65 |
> | **Uni-NTFM$_{large}$** | 1.9 B | 307 M | **77.37** | **82.69** | **84.01** | **71.25** | **64.36** | **74.26** |
>
> This table lists both Total Parameters and Inference Parameters for all baselines and Uni-NTFM variants with their performance metrics. While task-specific models like EEGNet are extremely lightweight (~2.8K parameters), they struggle with complex generalization tasks. Uni-NTFM bridges this gap by offering the necessary capacity for difficult tasks while maintaining an inference cost (19M for Tiny) that is manageable for modern hardware, unlike dense billion-scale models. While $Uni-NTFM_{large}$ has a massive capacity of 1.9B parameters, it only activates 307M parameters during inference. This effectively demonstrates that we can scale up the model's knowledge storage without increasing the computational burden.

---

> ### Author Response · Authors · 2025-11-27
> **[1/2]Response for weakness 4.**
>
> Dear reviewer zMGJ, for weakness 4, we thank you for the thoughtful assessment. While Uni-NTFM adopts several successful architectural components—common practice in foundation model design—our contribution does not aim to introduce a new operator/achtecture. Instead, the novelty lies in a neuroscience-guided architectural paradigm that addresses structural limitations of existing EEG foundation models in an integrated manner.
>
> 1. **Decoupled neural coding:**
>    The dual-stream design separates transient waveform dynamics and sustained spectral rhythms, enabling principled cross-domain fusion rather than treating heterogeneous EEG patterns as a single modality.
>
> 2. **Unified functional topography:**
>    The hierarchical topological embedding constructs a consistent neural coordinate system across heterogeneous montages, supporting cross-montage generalization and missing-channel robustness not achievable with standard positional encodings.
>
> 3. **Functional modularization:**
>    Sparse MoE routing aligns with EEG’s intrinsic heterogeneity, allowing specialized processing of rhythmic, transient, artifact, and pathological signals while mitigating interference seen in dense backbones.
>
> In detail, We do not simply stack modules. Our Heterogeneous Feature Projection Module (HFPM) physically separates transient waveform dynamics (Time) and sustained spectral rhythms (Frequency) processing streams. In the new **Appendix M (Table 20, Page 30)**, we show that our decoupled architecture outperforms a generic "STFT-MoE" baseline, which feeds stacked Short-Time Fourier Transform (STFT) patches into the same backbone by 14.57% on TUEV. This proves the necessity of our specific design.
>
> **Table 20: Comparison with STFT Baseline.**
>
> | Method | Input Design | Infer. GFLOPs |  Balanced Accuracy (Workload) | AUC-PR (Workload) | AUROC (Workload) | Balanced Accuracy (TUEV) | Cohen's Kappa (TUEV) | Weighted F1 (TUEV) |
> | :---: | :---: | :---: | :---: | :---: | :---: | :---: | :---: | :---: |
> | STFT-MoE | STFT (Mag. + Phase) | 16.14 | 62.32 | 68.39 | 68.81 | 52.37 | 54.26 | 76.92 |
> | **Uni-NTFM$_{small}$ (Ours)** | **Decoupled (Time + Freq)** | **15.87** | 66.72 | 71.73 | 73.88 | 66.94 | 67.96 | 83.25 |

---

> ### Author Response · Authors · 2025-11-27
> **[2/2]Response for weakness 4.**
>
> Besides, our Topological Embedding (TE) is not a standard positional encoding but a hierarchical coordinate system that uses spatial priors ($E_{intra}$ and $E_{region}$) to infer missing information from neighboring electrodes (demonstrated in **Appendix L, Table 18, page 29**), solving the specific montage heterogeneity problem in EEG that standard backbones cannot address. For instance, on the BCIC-IV-2a dataset, when 7 channels are dropped (representing 30% information loss for a 22-channel setup), the Balanced Accuracy of the baseline plummets by 11.99%. In contrast, the model with TE mitigates this loss to 8.34%, effectively alleviating performance reduction.
>
> **Table 18: Stress Test: Robustness to Missing Channels.**
>
> | Dataset | Metric | Method | Original ($k=0$) | Randomly Dropped $k=1$ | Randomly Dropped $k=3$ | Randomly Dropped $k=5$ | Randomly Dropped $k=7$ |
> | :---: | :---: | :---: | :---: | :---: | :---: | :---: | :---: |
> | **BCIC-IV-2a** | **Balanced Accuracy** | Uni-NTFM$_{small}$ (w/o TE) | 52.39 | 52.46 (+0.07%) | 50.14 (-2.25%) | 45.93 (-6.46%) | 40.40 (-11.99%) |
> | | | Uni-NTFM$_{small}$ (w/ TE) | 55.59 | 55.31 (-0.28%) | 54.52 (-1.07%) | 51.86 (-3.73%) | 47.25 (-8.34%) |
> | | **Cohen's Kappa** | Uni-NTFM$_{small}$ (w/o TE) | 40.87 | 40.50 (-0.37%) | 37.68 (-3.19%) | 33.93 (-6.94%) | 32.12 (-8.75%) |
> | | | Uni-NTFM$_{small}$ (w/ TE) | 42.01 | 41.83 (-0.18%) | 40.46 (-1.55%) | 38.61 (-3.40%) | 36.10 (-5.91%) |
> | | **Weighted F1** | Uni-NTFM$_{small}$ (w/o TE) | 53.23 | 52.81 (-0.42%) | 51.36 (-1.87%) | 48.12 (-5.11%) | 44.97 (-8.26%) |
> | | | Uni-NTFM$_{small}$ (w/ TE) | 54.65 | 54.71 (+0.06%) | 53.93 (-0.72%) | 51.70 (-2.95%) | 49.24 (-5.41%) |
> | **SEED** | **Balanced Accuracy** | Uni-NTFM$_{small}$ (w/o TE) | 66.85 | 66.97(+0.12%) | 66.42 (-0.43%) | 65.10 (-1.75%) | 63.77 (-3.08%) |
> | | | Uni-NTFM$_{small}$ (w/ TE) | 72.02 | 71.98 (-0.04%) | 71.67 (-0.35%) | 71.10 (-0.92%) | 70.18 (-1.84%) |
> | | **Cohen's Kappa** | Uni-NTFM$_{small}$ (w/o TE) | 55.42 | 55.35 (-0.07%) | 55.12 (-0.30%) | 54.01 (-1.41%) | 53.30 (-2.12%) |
> | | | Uni-NTFM$_{small}$ (w/ TE) | 58.59 | 58.68 (+0.09%) | 58.55 (-0.04%) | 58.51 (-0.38%) | 57.93 (-0.96%) |
> | | **Weighted F1** | Uni-NTFM$_{small}$ (w/o TE) | 68.98 | 68.87 (-0.11%) | 68.84 (-0.14%) | 68.09 (-0.89%) | 67.03 (-1.95%) |
> | | | Uni-NTFM$_{small}$ (w/ TE) | 72.74 | 72.79  (+0.05%) | 72.66 (-0.07%) | 72.50 (-0.24%) | 71.93 (-0.81%) |
>
> These elements operate coherently as a unified framework, enabling representational abilities that isolated components or prior BFMs do not provide. The revised experiments—montage transfer, missing-channel stress tests, and matched-compute comparisons—further demonstrate that this principled integration yields capabilities beyond system-level consolidation.

---

> ### Author Response · Authors · 2025-11-27
> **Response for question 1.**
>
> Dear reviewer zMGJ, for question 1, we have added **Appendix K: Impact of Data Augmentation (Page 28)**. **Table 16** compares the model with and without augmentation.
>
> **Table 16: Performance Comparison on Workload, ADFTD, and BCIC-IV-2a.**
>
> | Method |  Balanced Acc. (Workload) | AUC-PR (Workload) | AUROC (Workload) |  Balanced Acc. (ADFTD) | Cohen’s Kappa (ADFTD) | Weighted F1 (ADFTD) |  Balanced Acc. (BCIC-IV-2a) | Cohen’s Kappa (BCIC-IV-2a) | Weighted F1 (BCIC-IV-2a) |
> | :---: | :---: | :---: | :---: | :---: | :---: | :---: | :---: | :---: | :---: |
> | ***Only Pretrained Foundation Models (Multi-tasks)*** | | | | | | | | | |
> | Uni-NTFM$_{small}$ (w/o Aug.) | 62.83 | 66.95 | 69.79 | 66.52 | 67.06 | 70.10 | 51.44 | 40.15 | 50.78 |
> | Uni-NTFM$_{small}$ (w/ Aug.) | 64.11 | 67.71 | 70.26 | 67.35 | 67.44 | 71.31 | 52.09 | 40.44 | 51.23 |
> | ***Pretrained and Fine-tuned Foundation Models (Single-task)*** | | | | | | | | | |
> | Uni-NTFM$_{small}$ (w/o Aug.) | 65.39 | 69.47 | 71.91 | 75.20 | 74.34 | 76.13 | 54.63 | 41.35 | 54.06 |
> | Uni-NTFM$_{small}$ (w/ Aug.) | 66.72 | 71.73 | 73.88 | 75.68 | 76.32 | 76.80 | 55.59 | 42.01 | 54.65 |
>
> The exclusion of augmentation results in a noticeable reduction in feature quality in the Only Pretrained setting. For instance, on the Workload dataset, the Balanced Accuracy decreases from 64.11% to 62.83%, and on the BCIC-IV-2a dataset, it drops from 52.09% to 51.44%. This indicates that simulating noise, channel loss, and temporal drift during pre-training is essential for learning representations that are invariant to common EEG signals. In the Fine-tuned setting, the Weighted F1 score on BCIC-IV-2a drops from 54.65% to 54.06%, and the Balanced Accuracy on the Workload task declines by 1.33%. These results domenstrate augmentation enhances generalization and robustness, and removing data augmentation leads to a performance degradation across all datasets and evaluation settings. These results suggests that our data augmentation module ($f_{aug}$) provides a valuable performance boost by enhancing robustness against signal variations.

---

> ### Author Response · Authors · 2025-11-27
> **Response for question 2.**
>
> Dear reviewer zMGJ, for question 2, we have added **Appendix L: Ablation Study of Topological Embedding Components (Table 17, Page 28)**. We performed a fine-grained ablation study to decouple the contributions of the three components within our Topological Embedding (TE) module: Region Embedding ($E_{region}$), Intra-region Embedding ($E_{intra}$), and Global Absolute Embedding ($E_{abs}$).
>
> **Table 17: Detailed Ablation Study of Topological Embedding Components on TUEV dataset.**
>
> | Variant | $E_{abs}$ | $E_{region}$ | $E_{intra}$ | Balanced Acc. (%) | Cohen's Kappa (%) | Weighted F1 (%) |
> | :---: | :---: | :---: | :---: | :---: | :---: | :---: |
> | Variant A | ✗ | ✗ | ✗ | 63.16 | 64.80 | 80.19 |
> | Variant B | ✓ | ✗ | ✗ | 63.57 | 65.12 | 80.70 |
> | Variant C | ✗ | ✓ | ✗ | 62.72 | 64.98 | 80.35 |
> | Variant D | ✗ | ✓ | ✓ | 63.81 | 65.63 | 81.24 |
> | **Uni-NTFM$_{small}$** | **✓** | **✓** | **✓** | **64.05** | **65.82** | **81.74** |
>
> Variant B with only the global absolute index obtains a Balanced Accuracy of 63.57%,  but the improvement is limited than the baseline Variant A. This suggests that treating EEG electrodes merely as a flat sequence fails to capture the intricate spatial relationships required for effective decoding. Notably, employing region-level embeddings alone results in a performance reduce to 62.72%, which is lower than the baseline. We think that providing only regional information introduces ambiguity, as the model cannot distinguish between different electrodes within the same brain region, leading to a loss of channel-specific resolution. However, when intra-region embeddings are introduced to resolve the ambiguity of Variant C, performance significantly improves to 63.81%. This highlights the necessity of the hierarchical structure: $E_{region}$ provides the macroscopic functional context, while $E_{intra}$ restores the microscopic spatial resolution. This combination validates that structured spatial priors are more effective than arbitrary sequence indices.

---

### Official Review · Reviewer_BGmc · 2025-11-01

**Soundness:** 3
**Presentation:** 3
**Contribution:** 2
**Rating:** 4
**Confidence:** 4

**Summary:**

This paper introduces Uni-NTFM, a foundation model for EEG that addresses domain-specific challenges through a decoupled time-frequency architecture, topological electrode embeddings across different standards, and a Mixture-of-Experts Transformer, pretrained on 28,000+ hours of diverse EEG data. The model demonstrates "superior performance" over existing task-specific methods and foundation models across nine downstream tasks under both linear probing and fine-tuning settings.

**Strengths:**

- The decoupled time-frequency architecture, topological electrode embeddings, and MoE-based routing represent thoughtfu

- Extensive experiments across nine diverse downstream tasks under both linear probing and fine-tuning.

Outstanding presentation featuring clear writing and accurate usage of technical terminology, effectively applying concepts such as foundation models, zero-shot learning, and fine-tuning, which are often misused in the community.

- The 1.9B parameter scale, 28,000+ hours of diverse pretraining data, and superior performance establish a meaningful benchmark for EEG foundation models with potential clinical and neuroscience applications.

All necessary code, data, and information are provided to ensure reproducibility.

**Weaknesses:**

- The ablation study is conducted on a near-saturation task [1](e.g., TUAB), where the baseline model shows low performance; even a lightweight model such as EEGNet (~2k parameters) could outperform it (see [2]).

- The reported improvement over existing methods is marginal, limiting the practical significance of the contribution.

The novelty of this approach is limited, as it appears to be a combination of existing methods [3,4].

[1] Kiessner, A. K., Schirrmeister, R. T., Boedecker, J., & Ball, T. (2024). Reaching the ceiling? Empirical scaling behaviour for deep EEG pathology classification. Computers in Biology and Medicine, 178, 108681.

[2] Darvishi-Bayazi, M. J., Ghaemi, M. S., Lesort, T., Arefin, M. R., Faubert, J., & Rish, I. (2024). Amplifying pathological detection in EEG signaling pathways through cross-dataset transfer learning. Computers in biology and medicine, 169, 107893.

[3] Zhang, X., Zhao, Z., Tsiligkaridis, T., & Zitnik, M. (2022). Self-supervised contrastive pre-training for time series via time-frequency consistency. Advances in neural information processing systems, 35, 3988-4003.

[4] Kuruppu, G., Wagh, N., & Varatharajah, Y. (2025). Eeg foundation models: A critical review of current progress and future directions. arXiv preprint arXiv:2507.11783.

**Questions:**

- Given that you have already provided different model sizes, would it be possible to create a performance scaling plot versus the number of parameters?

Do we really need billions of parameters for the current size of EEG data?

- Would it be better to plot the ablation tables for a clearer and faster understanding?

---

> ### Author Response · Authors · 2025-11-27
> **Response for weakness 1.**
>
> Dear reviewer BGmc, for weakness 1, we appreciate the opportunity to clarify:
>
> 1. **Clarification on Complex Task Ablation (TUEV):**  We respectfully highlight that our ablation study contains not only TUAB but also the more challenging TUEV (6-class Classification) dataset, as shown in **Table 4 (Page 10)** and **Figure 3 (Page 10)**. Unlike TUAB, TUEV is far from saturation. On this dataset, our full model achieves a Balanced Accuracy of 64.05%, which is 5.31% higher than the baseline (**58.74%**), while EEGNet only achieves 39.75%. This indicates that the proposed modules bring clear improvements on a more difficult dataset where lightweight architectures do not perform well.
>
> **Table 4: Ablation study on TUAB and TUEV datasets.**
>
> | No. | HFPM | DCM | TE | MoE |  Balanced Acc. (TUAB) | AUC-PR (TUAB) | AUROC (TUAB) | Balanced Acc. (TUEV) | Cohen's Kappa (TUEV) | Weighted F1 (TUEV) |
> | :---: | :---: | :---: | :---: | :---: | :---: | :---: | :---: | :---: | :---: | :---: |
> | 1 | ✗ | ✗ | ✗ | ✗ | $62.52_{\pm 4.83}$ | $68.71_{\pm 3.69}$ | $71.16_{\pm 5.37}$ | $58.74_{\pm 3.16}$ | $62.29_{\pm 2.27}$ | $73.66_{\pm 2.82}$ |
> | 2 | ✓ | ✗ | ✗ | ✗ | $69.95_{\pm 3.24}$ | $77.78_{\pm 2.90}$ | $78.05_{\pm 2.53}$ | $61.45_{\pm 2.56}$ | $63.34_{\pm 2.58}$ | $76.69_{\pm 3.51}$ |
> | 3 | ✓ | ✓ | ✗ | ✗ | $73.64_{\pm 2.31}$ | $79.92_{\pm 2.11}$ | $79.76_{\pm 2.85}$ | $62.88_{\pm 1.95}$ | $63.93_{\pm 2.15}$ | $78.72_{\pm 1.67}$ |
> | 4 | ✗ | ✗ | ✓ | ✗ | $67.40_{\pm 3.61}$ | $74.05_{\pm 2.88}$ | $76.14_{\pm 3.39}$ | $60.60_{\pm 2.54}$ | $64.11_{\pm 1.99}$ | $77.81_{\pm 1.71}$ |
> | 5 | ✗ | ✗ | ✗ | ✓ | $66.23_{\pm 2.08}$ | $73.87_{\pm 2.44}$ | $78.62_{\pm 2.51}$ | $59.86_{\pm 3.44}$ | $62.79_{\pm 2.65}$ | $75.52_{\pm 2.13}$ |
> | 6 | ✗ | ✗ | ✓ | ✓ | $71.81_{\pm 1.95}$ | $78.49_{\pm 1.46}$ | $80.03_{\pm 2.12}$ | $61.50_{\pm 2.02}$ | $64.28_{\pm 2.37}$ | $78.94_{\pm 2.23}$ |
> | 7 | ✓ | ✗ | ✓ | ✓ | $73.34_{\pm 2.42}$ | $79.33_{\pm 1.86}$ | $81.10_{\pm 2.55}$ | $63.35_{\pm 2.49}$ | $65.39_{\pm 1.88}$ | $80.81_{\pm 2.41}$ |
> | 8 | ✓ | ✓ | ✗ | ✓ | $74.96_{\pm 1.92}$ | $81.37_{\pm 1.64}$ | $82.39_{\pm 2.05}$ | $63.16_{\pm 1.73}$ | $64.80_{\pm 1.89}$ | $80.19_{\pm 1.16}$ |
> | 9 | ✓ | ✓ | ✓ | ✗ | $74.17_{\pm 1.63}$ | $80.94_{\pm 2.28}$ | $81.40_{\pm 1.90}$ | $63.53_{\pm 2.20}$ | $64.41_{\pm 1.75}$ | $79.39_{\pm 1.42}$ |
> | 10 | ✓ | ✓ | ✓ | ✓ | $76.49_{\pm 1.31}$ | $82.30_{\pm 1.52}$ | $83.25_{\pm 1.26}$ | $64.05_{\pm 1.42}$ | $65.82_{\pm 1.45}$ | $81.74_{\pm 0.97}$ |
>
> 2. **Comprehensive Comparison with Lightweight Models:** To address your concern about lightweight models, we have added a "Comprehensive Comparison of Complexity and Performance" in **Appendix D (Table 5, Page 18)**. Our $Uni-NTFM_{tiny}$ (57M params) significantly outperforms it on complex multi-class tasks. In datail, $Uni-NTFM_{tiny}$ achieves 68.88% Balanced Accuracy compared to EEGNet's 55.17%, providing an excellent performance while maintaining acceptable inference costs (19M inference parameters). These results demonstrate that larger capacity is essential for complex brain signal decoding.
>
> **Table 5: Comprehensive Comparison of Complexity and Performance.**
>
> | Model | Total Parameters | Inference Parameters | Bal. Acc. (Binary Class) | AUC-PR (Binary Class) | AUROC (Binary Class) | Bal. Acc. (Multi-class) | Kappa (Multi-class) | W-F1 (Multi-class) |
> | :---: | :---: | :---: | :---: | :---: | :---: | :---: | :---: | :---: |
> | ***Traditional Task-specific Methods*** | | | | | | | | |
> | SPARCNet | 0.79 M | 0.79 M | 71.16 | 76.79 | 78.83 | 55.17 | 59.65 | 40.41 |
> | EEGNet | 2.8 K | 2.8 K | 71.04 | 74.40 | 78.40 | 56.79 | 48.13 | 59.79 |
> | CNN-Transformer | 3.2 M | 3.2 M | 71.83 | 75.01 | 76.04 | 56.18 | 48.01 | 60.86 |
> | EEGConformer | 1 M | 1 M | 74.19 | 79.11 | 77.90 | 59.82 | 50.42 | 62.63 |
> | FFCL | 2.4 M | 2.4 M | 74.74 | 79.80 | 80.52 | 55.99 | 45.41 | 59.68 |
> | ST-Transformer | 3.5 M | 3.5 M | 72.36 | 75.19 | 77.60 | 54.17 | 42.75 | 57.92 |
> | ContraWR | 1.6 M | 1.6 M | 75.08 | 81.58 | 80.33 | 57.13 | 47.49 | 60.81 |
> | ***Foundation Models*** | | | | | | | | |
> | NeuroLM-XL | 1.7 B | 1.7 B | 73.14 | 68.63 | 71.89 | 58.77 | 48.91 | 63.86 |
> | EEGPT | 25 M | 25 M | 74.72 | 75.41 | 80.50 | 65.81 | 58.28 | 68.94 |
> | BIOT | 3.6 M | 3.6 M | 75.60 | 81.58 | 81.29 | 62.52 | 49.51 | 60.73 |
> | LaBraM-Base | 5.8 M | 5.8 M | 76.25 | 81.87 | 82.07 | 69.55 | 62.25 | 73.67 |
> | CBraMod | 4.0 M | 4.0 M | 75.70 | 80.71 | 81.19 | 68.96 | 60.75 | 71.68 |
> | ***Proposed Models (Uni-NTFM)*** | | | | | | | | |
> | **Uni-NTFM$_{tiny}$** | 57 M | **19 M** | 74.78 | 79.57 | 80.66 | 68.88 | 62.19 | 72.48 |
> | **Uni-NTFM$_{small}$** | 427 M | 74 M | 76.35 | 81.07 | 82.31 | 70.10 | 63.35 | 73.34 |
> | **Uni-NTFM$_{middle}$** | 912 M | 148 M | 76.78 | 81.83 | 83.05 | 70.56 | 63.61 | 73.65 |
> | **Uni-NTFM$_{large}$** | 1.9 B | 307 M | **77.37** | **82.69** | **84.01** | **71.25** | **64.36** | **74.26** |

---

> ### Author Response · Authors · 2025-11-27
> **Response for weakness 2.**
>
> Dear reviewer BGmc, for weakness 2, we wish to provide further clarification on this point. Although the improvement over existing methods seems marginal, we believe it represents a meaningful step forward given the inherent difficulty of the multi-class tasks evaluated.
>
> 1. **Obvious Improvement on Complex Tasks:**  In **Appendix D (Table 5, Page 18)**, our method achieves the best results in both simple binary classification tasks and complex multi-class tasks. Even the most lightweight $Uni-NTFM_{tiny}$ (57M parameters) can surpass the 25M-parameter EEGPT and the 1.7B-parameter NeuroLM-XL in various tasks during inference by activating only 19M parameters, significantly enhancing the model's performance on complex multi-class tasks.
>
> 2. **SOTA Performance on Different Datasets:** As detailed in **Table 1 (Page 8), Table 2 (Page 9), and Table 3 (Page 9)**, Uni-NTFM consistently establishes new State-of-the-Art (SOTA) results across 9 diverse datasets, often surpassing the second-best methods by significant margins (e.g., +6.5% of Balanced Accuracy on TUEV compared to LaBraM-Base).

---

> ### Author Response · Authors · 2025-11-27
> **[1/2] Response for weakness 3.**
>
> Dear reviewer BGmc, for weakness 3, while components like MoE or FFT are existing, our contribution lies in a neuroscience-guided architectural paradigm that addresses structural limitations of existing EEG foundation models in an integrated manner. We clarified this in the **Introduction** and **Methodology**:
>
> 1. **Biologically-Grounded Transient-Sustained Dual Coding:**  We emulate the brain’s parallel processing architecture by designing a Heterogeneous Feature Projection Module (HFPM). This module physically decouples neural information into complementary streams, which aims to capture non-stationary transients by the time path and steady-state rhythms by the frequency path. Furthermore, a Dual-domain Cross-attention Module (DCM) is established to facilitate a cooperative interaction between these representations, generating a deeply fused representation through bidirectional information enhancement.
>
> 2. **Montage Alignment via Topological Embedding:** To reconstruct the brain’s geometry from different sensors, we introduce a hierarchical Topological Embedding (TE) scheme. By encoding spatial semantics at the intra-region level, region level, and absolute sequence level, this mechanism injects a precise neural coordinate system into the latent space and aligns heterogeneous electrode montages onto a unified functional topography.
>
> 3. **Functional Modularity with Sparse MoE:** We simulate the functional specialization and task decoupling of biological networks by replacing the dense FFN with a Mixture-of-Experts (MoE) architecture. This design enables dynamic functional routing, where specific signal patterns are processed by specialized expert subnetworks to relieve interferences between different tasks.
>
> 4. **Multi-view Self-Supervised Learning:** We propose a pre-training objective that forces the model to master the generative rules of neural activity. By simultaneously reconstructing the masked time-domain waveform patterns, frequency-domain spectral rhythms, and MoE auxiliary loss, this multi-view paradigm ensures that the model learns generalizable and robust representations of brain dynamics, rather than merely superficial features.
>
> In detail, We do not simply stack modules. Our Heterogeneous Feature Projection Module (HFPM) physically separates transient waveform dynamics (Time) and sustained spectral rhythms (Frequency) processing streams. In the new **Appendix M (Table 20, Page 30)**, we show that our decoupled architecture outperforms a generic "STFT-MoE" baseline, which feeds stacked Short-Time Fourier Transform (STFT) patches into the same backbone by 14.57% on TUEV. This proves the necessity of our specific design.
>
> **Table 20: Comparison with STFT Baseline.**
>
> | Method | Input Design | Infer. GFLOPs |  Balanced Accuracy (Workload) | AUC-PR (Workload) | AUROC (Workload) | Balanced Accuracy (TUEV) | Cohen's Kappa (TUEV) | Weighted F1 (TUEV) |
> | :---: | :---: | :---: | :---: | :---: | :---: | :---: | :---: | :---: |
> | STFT-MoE | STFT (Mag. + Phase) | 16.14 | 62.32 | 68.39 | 68.81 | 52.37 | 54.26 | 76.92 |
> | **Uni-NTFM$_{small}$ (Ours)** | **Decoupled (Time + Freq)** | **15.87** | 66.72 | 71.73 | 73.88 | 66.94 | 67.96 | 83.25 |

---

> ### Author Response · Authors · 2025-11-27
> **[2/2] Response for weakness 3.**
>
> Besides, our Topological Embedding (TE) is not a standard positional encoding but a hierarchical coordinate system that uses spatial priors ($E_{intra}$ and $E_{region}$) to infer missing information from neighboring electrodes (demonstrated in **Appendix L, Table 18, page 29**), solving the specific montage heterogeneity problem in EEG that standard backbones cannot address, and (w/ TE) denotes the model equipped with TE module, (w/o TE) means without the TE module. For instance, on the BCIC-IV-2a dataset, when 7 channels are dropped (representing 30% information loss for a 22-channel setup), the Balanced Accuracy of the baseline plummets by 11.99%. In contrast, the model with TE mitigates this loss to 8.34%, effectively alleviating performance reduction.
>
> **Table 18: Stress Test: Robustness to Missing Channels.**
>
> | Dataset | Metric | Method | Original ($k=0$) | Randomly Dropped $k=1$ | Randomly Dropped $k=3$ | Randomly Dropped $k=5$ | Randomly Dropped $k=7$ |
> | :---: | :---: | :---: | :---: | :---: | :---: | :---: | :---: |
> | **BCIC-IV-2a** | **Balanced Accuracy** | Uni-NTFM$_{small}$ (w/o TE) | 52.39 | 52.46 (+0.07%) | 50.14 (-2.25%) | 45.93 (-6.46%) | 40.40 (-11.99%) |
> | | | Uni-NTFM$_{small}$ (w/ TE) | 55.59 | 55.31 (-0.28%) | 54.52 (-1.07%) | 51.86 (-3.73%) | 47.25 (-8.34%) |
> | | **Cohen's Kappa** | Uni-NTFM$_{small}$ (w/o TE) | 40.87 | 40.50 (-0.37%) | 37.68 (-3.19%) | 33.93 (-6.94%) | 32.12 (-8.75%) |
> | | | Uni-NTFM$_{small}$ (w/ TE) | 42.01 | 41.83 (-0.18%) | 40.46 (-1.55%) | 38.61 (-3.40%) | 36.10 (-5.91%) |
> | | **Weighted F1** | Uni-NTFM$_{small}$ (w/o TE) | 53.23 | 52.81 (-0.42%) | 51.36 (-1.87%) | 48.12 (-5.11%) | 44.97 (-8.26%) |
> | | | Uni-NTFM$_{small}$ (w/ TE) | 54.65 | 54.71 (+0.06%) | 53.93 (-0.72%) | 51.70 (-2.95%) | 49.24 (-5.41%) |
> | **SEED** | **Balanced Accuracy** | Uni-NTFM$_{small}$ (w/o TE) | 66.85 | 66.97  (+0.12%) | 66.42 (-0.43%) | 65.10 (-1.75%) | 63.77 (-3.08%) |
> | | | Uni-NTFM$_{small}$ (w/ TE) | 72.02 | 71.98 (-0.04%) | 71.67 (-0.35%) | 71.10 (-0.92%) | 70.18 (-1.84%) |
> | | **Cohen's Kappa** | Uni-NTFM$_{small}$ (w/o TE) | 55.42 | 55.35 (-0.07%) | 55.12 (-0.30%) | 54.01 (-1.41%) | 53.30 (-2.12%) |
> | | | Uni-NTFM$_{small}$ (w/ TE) | 58.59 | 58.68  (+0.09%) | 58.55 (-0.04%) | 58.51 (-0.38%) | 57.93 (-0.96%) |
> | | **Weighted F1** | Uni-NTFM$_{small}$ (w/o TE) | 68.98 | 68.87 (-0.11%) | 68.84 (-0.14%) | 68.09 (-0.89%) | 67.03 (-1.95%) |
> | | | Uni-NTFM$_{small}$ (w/ TE) | 72.74 | 72.79 (+0.05%) | 72.66 (-0.07%) | 72.50 (-0.24%) | 71.93 (-0.81%) |
>
>
> These elements operate coherently as a unified framework, enabling representational abilities that isolated components or prior BFMs do not provide. The revised experiments—montage transfer, missing-channel stress tests, and matched-compute comparisons—further demonstrate that this principled integration yields capabilities beyond system-level consolidation.

---

> ### Author Response · Authors · 2025-11-27
> **Response for question 1.**
>
> Dear reviewer BGmc, for question 1, thank you for this suggestion. We wish to emphasize that the performance scaling plots versus the parameter counts were already included in our initial submission. In the revised manuscript, these results are presented in **Appendix H (Figure 4, Page 24)**. The figure explicitly visualizes the relationship between downstream performance and the number of parameters (ranging from 10M to 1B), demonstrating a clear positive correlation.

---

> ### Author Response · Authors · 2025-11-27
> **Response for question 2.**
>
> Dear reviewer, for question 2, this is a critical question between model scale and data efficiency. We have provided a detailed justification in the **Appendix D (Page 18)**, **Appendix H (Page 24)**, and **Appendix J (Page 27)**. Our rationale for scaling to 1.9 billion parameters is based on three key points:
>
> 1. **Total Capacity:**  Unlike images or text, EEG signals exhibit extreme heterogeneity due to inter-subject variability, diverse electrode impedances, and complex pathological patterns. As discussed in **Appendix D**, we argue that "model scale corresponds to the diversity of functional experts required to decode the complex signal of the brain". A large total parameter count (1.9B) is necessary to store this vast and diverse knowledge base, functioning like the neocortex with specialized experts for rare events.
>
> 2. **Sparse Activation:**  We do not activate all 1.9 billion parameters for every input. By leveraging the Mixture-of-Experts (MoE) architecture, we achieve "sparse coding" efficiency similar to biological brains. As shown in **Table 5 (Appendix D)**, while $Uni-NTFM_{large}$ has a total capacity of 1.9B parameters, its active inference parameters are only 307M. This means we get the learning capability of a billion-parameter model with the inference latency of a much smaller model.
>
> 3. **Validation of Efficiency:** In **Table 14 (Appendix J)**, we compared our sparse model against dense baselines. The results show that $Uni-NTFM_{small}$ matches the total capacity of a Dense-Large model (421M) with 17% active parameters (74M) and significantly lower 15.87GFLOPs. These results prove that the large parameter count is used for storing diverse experts rather than increasing computational burden, making it a necessary and efficient design for the current scale of EEG data.
>
> **Table 14: Efficiency Comparison between Uni-NTFM$_{small}$ (MoE) and Dense Baselines.**
>
> | Model Variant | Architecture | Total Parameters | Active Parameters | Total GFLOPs | Inference GFLOPs |
> | :---: | :---: | :---: | :---: | :---: | :---: |
> | Dense-Small (Same-FLOPs) | Dense Transformer | 74 M | 74 M | 16.08 | 16.08 |
> | Dense-Large (Same-Params) | Dense Transformer | 421 M | 421 M | 93.15 | 93.15 |
> | Uni-NTFM$_{small}$ (Ours) | Sparse MoE | 427 M | 74 M | 100.40 | 15.87 |
> *Note: Active Params for MoE assumes Top-2 gating with 16 experts.*

---

> ### Author Response · Authors · 2025-11-27
> **Response for question 3.**
>
> Dear reviewer BGmc, for question 3, this is an excellent suggestion. In the revised manuscript, we added **Figure 3 (Page 10)** to provide visually statistical evidence of the ablation experiments.

---

### Author Response · Authors · 2025-11-29
**[2/2]To the Area Chair, Senior Area Chair, Program Committee, and Reviewers**

**We added additional ablation studies:**

- Following **Reviewer 2**, we added an ablation on **data augmentation** to quantify its impact on robustness **(Appendix K, Table 16, Page 28)**. *This addresses the question regarding how performance would change without EEG data augmentation **(Reviewer 2, Question 1)**.*
- Following **Reviewer 2**, we added a detailed ablation of the topological embedding components ($E_{region}, E_{intra}, E_{abs}$) to validate the hierarchical design **(Appendix L, Table 17, Page 28)**. *This addresses the request for specific ablations on the different position embedding component combinations **(Reviewer 2, Question 2)**.*

**We conducted additional robustness and scaling evaluations:**

- Following **Reviewer 3**, we added **Missing Channel Stress Tests**, demonstrating the model's resilience to sensor malfunctions **(Appendix L, Table 18, Page 29)**. *This addresses the weakness regarding the lack of direct tests on systematically missing channels **(Reviewer 3, Weakness 1)**.*
- Following **Reviewer 3**, we added **Cross-Montage Transfer** experiments, proving the model can generalize from high-density (62-ch) to clinical (19-ch) montages without retraining **(Appendix L, Table 19, Page 29)**. *This addresses the suggestion to consider cross-montage transfer to validate topological unification **(Reviewer 3, Weakness 1)**.*
- Following **Reviewer 2 and 3**, we added **Label Efficiency Analysis**, showing that Uni-NTFM retains ~92% performance with only 10% of labeled data **(Appendix N, Table 21, Page 30)**. *This addresses the suggestion to analyze downstream accuracy versus the labeled fraction used in fine-tuning **(Reviewer 3, Question 1)**.*
- Following **Reviewer 1, 3, and 4**, we demonstrated explicit **Scaling Laws** visualizations for both model size and data size **(Appendix H, Figure 4, Page 24)**. *This addresses the requests to create performance scaling plots versus the number of parameters **(Reviewer 1, Question 1)** and to make scaling analysis more explicit **(Reviewer 3, Weakness 3)**.*

We would like to reiterate our deep appreciation for the reviewers' constructive comments and the time dedicated to evaluating our work. These insights have been invaluable in strengthening the manuscript. We hope that the revisions and our point-by-point responses effectively clarify the issues raised. We are ready to provide additional information to ensure all concerns are fully resolved.

Sincerely,

The Authors

---

### Author Response · Authors · 2025-11-29
**[1/2]To the Area Chair, Senior Area Chair, Program Committee, and Reviewers**

To the AC, SAC, PC, and Reviewers:

We would like to express our sincere gratitude for the time and effort dedicated to reviewing our manuscript. We are deeply encouraged by the positive feedback and the recognition of our work's contribution to the field. We notably appreciate the reviewers' acknowledgment of:

- **Originality & Soundness:** The reviewers highlighted the originality of our neuroscience-inspired architecture (**R3**), characterizing the decoupled time-frequency design and topological embeddings as "thoughtful" (**R1**) and "well-motivated" (**R4**). This validates our approach of aligning deep learning architectures with biological principles.
- **Superior Performance:** There is a consensus that Uni-NTFM demonstrates "superior performance" (**R1**) and achieves "state-of-the-art results" (**R2**) across a diverse range of downstream tasks, ranging from clinical diagnostics to cognitive state classification.
- **Scale & Significance:** The implementation of a 1.9B parameter model and the utilization of over 28,000 hours of pre-training data were recognized as establishing a "meaningful benchmark" (**R1**) with "notable significance" (**R3**) for the community.
- **Presentation Quality:** We are grateful for the commendations regarding the "outstanding presentation" (**R1**) and "excellent" clarity (**R2**, **R4**) of the manuscript.

We have carefully considered all comments and conducted extensive additional experiments to address the concerns raised. **All major revisions in the updated manuscript are highlighted in red.** Below is a summary of the key improvements:

**We added comprehensive efficiency analyses and dense model comparisons:**

- Following the suggestions of **Reviewer 1, 2, and 3**, we added a detailed efficiency analysis comparing our sparse MoE architecture against dense baselines matched for both inference compute and total capacity **(Appendix J, Page 27; Appendix D, Table 5, Page 18)**. We demonstrated that $Uni-NTFM_{small}$ matches the capacity of dense models while requiring **83% fewer active parameters** and **~6x lower inference GFLOPs**. *This directly addresses the concern regarding the necessity of the 1.9B parameter scale **(Reviewer 1, Question 2)**, reports training cost and inference speed **(Reviewer 3, Weakness 3)**, and substantiates the claim of efficient scaling by providing the requested matched-compute comparison **(Reviewer 2, Weakness 2)**.*

**We added additional methodological rationales and validations:**

- Following **Reviewer 2 and 3**, we corrected the terminology from "zero-shot" to **"linear probing"** throughout the manuscript. *This addresses the concern that the term "zero-shot transfer" was inaccurate for the linear probing evaluation setup **(Reviewer 2, Weakness 1)**.* And we rephrased the **Abstract (Page 1)**, **Introduction** **(Page 1-3)**, and **Methodology** **(Page 3-5)** sections. *These refinements aim to better state the neuroscience-inspired motivation of our architecture and improve the rigor and readability of the manuscript.*
- Following **Reviewer 3**, we added a visual analysis of **MoE specialization**, utilizing Brain Region-Expert Affinity Matrices to prove that experts specialize in processing signals from distinct functional brain regions **(Appendix O, Figures 6-8, Page 31-33)**. *This addresses the critique that the work did not initially show that different experts specialize on distinct EEG patterns **(Reviewer 3, Weakness 2).***
- Following **Reviewer 3 and 4**, we implemented a computation-matched **STFT-MoE baseline** to rigorously justify the necessity of our decoupled multi-stream architecture (HFPM) over single-stream early fusion **(Appendix M, Table 20, Page 30)**. *This addresses the question of whether three separate streams are truly needed compared to a single well-designed stream **(Reviewer 3, Question 2)** and clarifies the architectural novelty against joint encoding methods **(Reviewer 4, Weakness 1)**.*
- Following **Reviewer 1 and 4**, we strengthened the statistical rigor by reporting **standard deviations** calculated over five independent trials across main experiments **(Tables 1 on Page 8; Tables 2, 3 on Page 9; Tables 4 on Page 10)**. And a Wilcoxon signed-rank test was conducted to compare our model with all other baselines, the comparisons yield *p < 0.01*, indicating our model statistically significant improvements over the second-best methods. *This addresses the concern that reported improvements were marginal **(Reviewer 1, Weakness 2)** and provides the requested statistical evidence to prove significance **(Reviewer 4, Weakness 2)**.*
- Following **Reviewer 1**, we demonstrated the ablation study to include the complex **TUEV dataset** to rule out saturation effects on simpler tasks **(Table 4, Page 10)**. *This addresses the concern that the ablation study was originally conducted on a near-saturation task like TUAB **(Reviewer 1, Weakness 1)**.*

---

### Meta-Review · Area_Chair_QTfN · 2026-01-02

**Summary:**

The paper presents a unified EEG foundation model with clear methodological contributions and strong empirical results across multiple tasks. The authors made a comprehensive rebuttal that satisfactorily addresses the reviewers concerns regarding evaluation clarity, efficiency comparisons, and ablation completeness to support the technical rigour and demonstrated impact. Overall, the revised version of this work is clear for acceptance.

**Reviewer Concerns:**

The rebuttal clarifies evaluation protocols (e.g., linear probing vs. zero-shot), strengthens experimental rigour with additional ablations and statistical analyses, and provides clearer justification for the model design choices and efficiency comparisons. Some minor concerns about the level of conceptual novelty beyond the evaluated datasets remain only partially resolved.

**Reviewer Scores:**

The borderline reviewers are likely to increase their scores to positive, as the key methodological and evaluation concerns were addressed in the rebuttal.

---

### Decision · Program_Chairs · 2026-01-26

Accept (Poster)